# An Improved Empirical Fisher Approximation for Natural Gradient Descent

**Xiaodong Wu**[1*]    **Wenyi Yu**[2*]    **Chao Zhang**[2]    **Philip Woodland**[1]

[1]Dept. of Engineering, University of Cambridge    [2]Dept. of Electronic Engineering, Tsinghua University

`{xw338,pw117}@cam.ac.uk`   `{ywy22@mails,cz277@mail}.tsinghua.edu.cn`

## Abstract

Approximate Natural Gradient Descent (NGD) methods are an important family of optimisers for deep learning models, which use approximate Fisher information matrices to pre-condition gradients during training. The empirical Fisher (EF) method approximates the Fisher information matrix empirically by reusing the per-sample gradients collected during back-propagation. Despite its ease of implementation, the EF approximation has its theoretical and practical limitations. This paper investigates the *inversely-scaled projection* issue of EF, which is shown to be a major cause of its poor empirical approximation quality. An improved empirical Fisher (iEF) method is proposed to address this issue, which is motivated as a generalised NGD method from a loss reduction perspective, meanwhile retaining the practical convenience of EF. The exact iEF and EF methods are experimentally evaluated using practical deep learning setups, including widely-used setups for parameter-efficient fine-tuning of pre-trained models (T5-base with LoRA and Prompt-Tuning on GLUE tasks, and ViT with LoRA for CIFAR100). Optimisation experiments show that applying exact iEF directly as an optimiser provides strong convergence and generalisation. It achieves the best test performance and the lowest training loss for the majority of the tasks, even when compared to well-tuned AdamW/Adafactor baselines. Additionally, under a novel empirical evaluation framework, the proposed iEF method shows consistently better approximation quality to exact Natural Gradient updates than both the EF and the more expensive sampled Fisher methods, meanwhile demonstrating the superior property of being robust to the choice of damping across tasks and training stages. Improving existing approximate NGD optimisers with iEF is expected to lead to better convergence and robustness. Furthermore, the iEF method also serves as a better approximation method to the Fisher information matrix itself, which enables the improvement of a variety of Fisher-based methods, not limited to the scope of optimisation.

## 1 Introduction

Parameter optimisation is a crucial research area in the field of deep learning, where stochastic optimisers are commonly used which update the model parameters iteratively to minimise a target loss function. Approximate Natural Gradient Descent (NGD) [1] methods are an important family of approximate second-order optimisers, which pre-condition the gradient with the (approximate) Fisher information matrix (also called the Fisher matrix) to accelerate training or improve generalisation.

Although there are many successful optimisers based on approximate NGD, many of them in fact use the empirical Fisher (EF) as a pre-conditioner. These methods are referred to as approximate empirical NGD methods [52]. The EF method constructs an approximation to the exact Fisher matrix directly from the gradients of training samples, which are usually readily computed during the

---

*These authors contributed equally to this work

38th Conference on Neural Information Processing Systems (NeurIPS 2024).

training process [20]. In contrast, the exact Fisher matrix needs to be either sampled from the model output distribution [26], or requires repeated evaluation of the matrix-vector product with the Fisher matrix [25], which are both expensive operations. As a result, due to the ease of implementation brought by EF, empirical NGD is used in many approximate NGD optimisers as the default choice [33, 54, 43, 10, 11, 52, 51].

Despite the prevalence of empirical NGD optimisers, it is known that EF is in general a questionable approximation to the exact Fisher matrix [26, 20, 47]. The poor approximation quality of EF-based updates has been experimentally verified for small-scale experimental setups by [20, 47]. However, traditional evaluation methods are used in [20, 47] where the exact NG update and Fisher matrix need to be explicitly computed, making their findings impossible to verify for large deep learning setups for practical tasks. Hence, a more generally applicable evaluation framework is needed. There is also a need for an improved approximation to the exact Fisher matrix (as a pre-conditioner) than EF, while being as efficient to implement. This paper aims to fill these gaps.

**Our Contributions:** In this paper, an improved EF (iEF) approximation for NGD is proposed, which provides a better approximation to the exact Natural Gradient (NG) updates, while maintaining the practical convenience of EF. This method allows for a straightforward upgrade for all existing approximate empirical NGD optimisers. To achieve this, a theoretical investigation into the behaviour of EF update is first carried out, where the impact of the EF update on each involved sample is analysed (the "involved samples" refers to training samples whose gradient is used to construct the EF update). This leads to finding the *inversely-scaled projection* issue of EF (see Sec. 4). Accordingly, the iEF method is proposed to overcome this issue by introducing a diagonal scaling matrix to the standard formulation of the EF pre-conditioner. It is motivated as an approximate Gauss-Newton algorithm from a loss reduction perspective, with global convergence guarantees under mild assumptions (see Sec. 5). A novel empirical evaluation framework for approximate NGD methods is then proposed to enable accurate comparison of approximate Fisher pre-conditioners (*e.g.* EF and iEF) in large-scale optimisation setups (see Sec. 6). We conducted experiments that compare the exact EF and iEF methods in a range of practical deep learning setups including computer vision and fine-tuning large language models. Under our evaluation framework, iEF demonstrates better approximation quality to exact NG updates than both EF and the more expensive Monte-Carlo sampled Fisher method (SF, see Appendix E), meanwhile being significantly more robust to the choice of damping across tasks and training stages. Direct application of iEF as optimiser also shows consistently strong generalisation and convergence, even when compared to well-tuned AdamW/Adafactor baselines (see Sec. 7).

## 2 Related Work

**Approximate (Empirical) NGD:** There are many existing approximate (empirical) NGD methods, most of which use EF despite its theoretical limitations. Some prior work, *e.g.* [43, 40], uses the Woodbury identities [36] to exactly compute EF updates. Recent block-diagonal methods (based on K-FAC [27]) have gained popularity due to their efficiency, which includes work that modify the K-FAC approximation [54, 41, 10, 4, 52, 3] or distributively apply K-FAC as optimisers [34, 49]. Sometimes Adagrad-based methods [9, 13, 17] are also regarded as empirical NGD methods. However, the connection is questionable [20] as these methods use the *square-root* of the EF matrix, instead of the EF matrix itself, as a pre-conditioner.

**Limitations of EF Approximation:** The limitations of EF as an approximate Fisher matrix have been discussed and demonstrated in several papers [26, 20, 47], among which [20] provided a thorough review and analysis. However, as far as we are aware, there has been no prior work that analysed the exact EF method in larger deep-learning setups, and most of the observations are limited to small-scale problems for theoretical machine-learning studies. It is known, however, that practical EF-based optimisers usually require a sophisticated damping scheme to work well [33, 31]. It has even been suggested that an infinitely large damping should be used with the gradient covariance term [4, 35]. These observations can be tied to the theoretical limitations of EF.

**Empirical Evaluation of Approximate NGD Quality:** An accurate evaluation of the approximation quality to exact NG updates is of great importance for approximate NGD methods. Usually, the performance of the method of interest is evaluated on machine learning benchmarks [54, 27, 11], which provide crucial information from the optimisation perspective. However, limited information about the approximation quality to exact NGD can be drawn from these experiments. Therefore,

additional small-scale experiments are usually performed to compare against the exact Fisher matrices [47, 27, 11], or the exact NG updates [20, 41, 3], which are extremely difficult to do for commonplace large-scale models. This limits our understanding of these methods in the context of large-scale tasks.

## 3 Preliminaries

**Supervised Learning for Classification Model with Softmax Activation:** This paper considers supervised learning of categorical classification, where a probabilistic model is trained to predict outputs $y \in \{c | c = 1, 2, \ldots C\}$ of $C$ categories from inputs $\boldsymbol{x} \in \mathbb{X}$. The target model $\boldsymbol{z} = f_{\boldsymbol{\theta}}(\boldsymbol{x})$ has $\boldsymbol{\theta} \in \mathbb{R}^P$ as the model parameters, which outputs the logits $\boldsymbol{z} \in \mathbb{R}^C$. Assume a softmax activation is used on the logits, the model can be expressed as a conditional probability of $p_{\boldsymbol{\theta}}(y | \boldsymbol{x})$. Given $N$ *i.i.d.* training samples $(\boldsymbol{x}_n, y_n)_{n=1}^N$ (assuming $N \ll P$), the following accumulated loss is minimised

$$\mathcal{L}(\boldsymbol{\theta}) = \sum_n - \log p_{\boldsymbol{\theta}}(y = y_n | \boldsymbol{x}_n) = \sum_n l_n, \tag{1}$$

where $l_n = -\log p_{\boldsymbol{\theta}}(y = y_n | \boldsymbol{x}_n)$ is the categorical cross-entropy loss for the $n$-th training sample. For brevity, we denote $p_{\boldsymbol{\theta}}(y = c | \boldsymbol{x}_n) = p_n(c)$.

A vectorised representation of loss $\boldsymbol{l} \in \mathbb{R}^N$ is used where $\boldsymbol{l} = \begin{bmatrix} l_1, l_2, \cdots, l_N \end{bmatrix}^\top$. The accumulated loss then becomes $\mathcal{L}(\boldsymbol{\theta}) = \sum_n l_n = \boldsymbol{l}^\top \mathbf{1}$ where $\mathbf{1}$ is an all 1 column vector of matching dimension, and the accumulated gradient can be re-written as $\nabla_{\boldsymbol{\theta}} \mathcal{L}(\boldsymbol{\theta}) = \nabla_{\boldsymbol{\theta}} \boldsymbol{l}^\top \mathbf{1}$ where $\nabla_{\boldsymbol{\theta}} \boldsymbol{l} \in \mathbb{R}^{N \times P}$ is the Jacobian of per-sample losses *w.r.t.* model parameters.

**NGD and Empirical NGD** In a first-order optimisation method, say SGD [42], the update direction on the model parameter is the estimate of the accumulated gradient $\nabla_{\boldsymbol{\theta}} \mathcal{L}(\boldsymbol{\theta})$. In the NGD method [26], the gradient is pre-conditioned by the Fisher matrix $\mathbf{F}$ (*i.e.* $\mathbf{F}^{-1} \nabla_{\boldsymbol{\theta}} \mathcal{L}(\boldsymbol{\theta})$) to accelerate convergence. The exact Fisher matrix can be computed from the model output distribution using available training samples as follows

$$\mathbf{F} := \sum_n \sum_c p_n(c) \left[ \nabla_{\boldsymbol{\theta}} \log p_n(c) \nabla_{\boldsymbol{\theta}} \log p_n(c)^\top \right]. \tag{2}$$

The Fisher matrix can be estimated with Monte-Carlo (MC) sampling [27]. This approximation method is usually used with one MC sample per training sample, which is termed SF in this paper (see Appendix E). Alternatively, when the model is well trained and $p_n(y_n) \to 1$ for all $N$ samples, it is possible to approximate the exact Fisher with EF using the empirical gradient as follows

$$\tilde{\mathbf{F}} := \sum_n \left[ \nabla_{\boldsymbol{\theta}} \log p_n(y_n) \nabla_{\boldsymbol{\theta}} \log p_n(y_n)^\top \right] = \nabla_{\boldsymbol{\theta}} \boldsymbol{l}^\top \nabla_{\boldsymbol{\theta}} \boldsymbol{l}. \tag{3}$$

Pre-conditioning the gradient with the EF matrix (*i.e.* $\tilde{\mathbf{F}}^{-1} \nabla_{\boldsymbol{\theta}} \mathcal{L}(\boldsymbol{\theta})$) yields the empirical NGD method. Although empirical NGD is prevalent due to the convenience of computing the EF matrix in practice, the approximation quality of EF to the exact Fisher matrix is worth questioning [20, 47].

## 4 Inversely-Scaled Projection Issue of Empirical Fisher

Despite the practical prevalence of the EF method, it is generally believed to be a poor approximation of the exact NGD method [20]. To better understand the cause of the limited approximation quality of the EF method, an analysis of the impact of the EF update on each of the involved samples is presented below. This leads to finding the "*inversely-scaled projection* issue" of the EF method, which provides a focus for the improvement of the EF method.

### 4.1 Formal Definition

Recall the definition of EF in Eqn. (3). The empirical NG update (or just the EF update) can be defined as follows

$$\Delta \boldsymbol{\theta}_{\text{EF}} = -\eta \tilde{\mathbf{F}}^{-1} \nabla_{\boldsymbol{\theta}} \mathcal{L}(\boldsymbol{\theta}) = -\eta \left( \nabla_{\boldsymbol{\theta}} \boldsymbol{l}^\top \nabla_{\boldsymbol{\theta}} \boldsymbol{l} + \lambda \mathbf{I} \right)^{-1} (\nabla_{\boldsymbol{\theta}} \boldsymbol{l}^\top \mathbf{1}) \tag{4}$$

where $\lambda \in \mathbb{R}^+$ is a small damping factor to facilitate inversion (gradient covariance matrix $\nabla_{\boldsymbol{\theta}} \boldsymbol{l}^\top \nabla_{\boldsymbol{\theta}} \boldsymbol{l} \in \mathbb{R}^{P \times P}$ cannot be directly inverted for over-parameterised models). Using the Woodbury identity [36], the EF update can be re-expressed as follows:

$$\Delta \boldsymbol{\theta}_{\text{EF}} = -\eta \nabla_{\boldsymbol{\theta}} \boldsymbol{l}^\top (\nabla_{\boldsymbol{\theta}} \boldsymbol{l} \nabla_{\boldsymbol{\theta}} \boldsymbol{l}^\top + \lambda \mathbf{I})^{-1} \mathbf{1}. \tag{5}$$

The loss change induced on each sample (denoted as $\Delta l_{\text{EF}}$) when applying the EF update to the model can be estimated using the Jacobian $\nabla_{\boldsymbol{\theta}} l$ as follows:

$$\Delta l_{\text{EF}} = -\nabla_{\boldsymbol{\theta}} l \Delta \boldsymbol{\theta}_{\text{EF}} = -\eta \, \nabla_{\boldsymbol{\theta}} l \nabla_{\boldsymbol{\theta}} l^{\top} (\nabla_{\boldsymbol{\theta}} l \nabla_{\boldsymbol{\theta}} l^{\top} + \lambda \mathbf{I})^{-1} \mathbf{1} \approx -\eta \mathbf{1},$$

This result means that EF updates have the property of inducing an equal loss reduction on every involved sample. For the $n$-th sample, the projection of the EF update onto gradient direction $\nabla_{\boldsymbol{\theta}} l_n$ (denoted as $(\kappa_n)_{\text{EF}}$) can be computed as follows

$$(\kappa_n)_{\text{EF}} = \Delta \boldsymbol{\theta}_{\text{EF}}^{\top} \frac{\nabla_{\boldsymbol{\theta}} l_n}{\|\nabla_{\boldsymbol{\theta}} l_n\|_2} = -\frac{\eta}{\|\nabla_{\boldsymbol{\theta}} l_n\|_2}, \tag{6}$$

where $\|\nabla_{\boldsymbol{\theta}} l_n\|_2$ denotes the $l_2$ norm of the $n$-th per-sample gradient. This means that the projection of EF update onto every sample gradient is inversely proportional to the gradient norm of each sample. Note that a smaller $\|\nabla_{\boldsymbol{\theta}} l_n\|_2$ generally indicates the sample is better trained (or more converged, or closer to its minimum). The EF update is therefore easily biased towards well-trained samples, and tends to have a larger norm as training progresses ($\|\nabla_{\boldsymbol{\theta}} l_n\|_2$ decreases) [20]. We term this the *inversely-scaled projection* issue of the EF update, which is further illustrated in the following section.

## 4.2 Visual Illustration

The detrimental impact of the *inversely-scaled projection* issue of EF updates is illustrated in a 2-parameter 2-datum linear least-square regression problem in Fig. 1 (third plot). It is shown that EF updates are "attracted" to the minimum of each training sample (the dashed lines), leading to a distorted update vector field and inefficient training trajectories. Also, EF updates have a much larger norm when either training sample is nearly converged, suggesting the necessity of a complicated step-size scheduler. Please refer to Appendix B for a detailed description and discussion, which also includes an additional visualisation for a logistic regression setup in Fig. 4 which leads to similar observations. These effects of the *inversely-scaled projection* issue are further validated in experiments (E1) and (E2) in large-scale deep learning setups in Sec. 7.

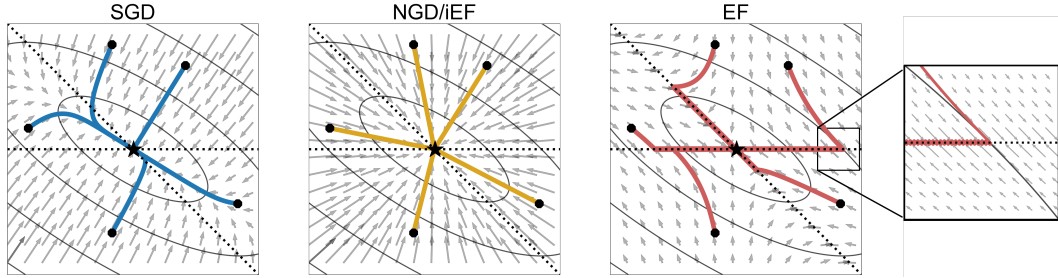

Figure 1: A visual comparison of Fisher, iEF and EF as pre-conditioners for a 2-parameter 2-datum linear least-squares regression problem inspired by [20] (see Appendix B for details). All three plots are loss landscapes with the $x$-axis and $y$-axis representing $\theta_0$ and $\theta_1$ respectively. The first plot shows the gradient vector field of the loss function and 5 sampled training trajectories for SGD updates. Similarly, the second plot is for NGD/iEF updates and the third plot is for EF updates (with a zoomed view). The global minimum $(0, 0)$ is marked with a star where visible. The two dashed lines on all plots represent the optimal parameter sets for each training sample. It can be seen that the EF method has a highly distorted update vector field while the iEF and NGD methods adapt to the curvature of the problem successfully.

## 5 Improved Empirical Fisher

The EF method is a widely used approximate NGD method, mainly because it can be implemented conveniently by constructing the EF matrix with the per-sample gradients that are readily computed during backpropagation. In this section, we propose the improved EF (iEF) method which preserves the implementational convenience of the EF method, meanwhile alleviating the *inversely-scaled projection* issue. The iEF method can be justified as an approximate (generalised) NGD method from a loss reduction perspective. Continuous-time convergence analyses also show that the iEF method guarantees sub-linear/linear convergence to the global minimum under mild assumptions.

## 5.1 Update Formulation

The nature of the *inversely-scaled projection* issue is that the EF update enforces a constant loss reduction regardless of the convergence level of each sample (see Sec. 4.1). To address this issue, the iEF update is designed to induce a per-sample loss reduction that takes into account the convergence level. The loss reduction induced by the iEF update for the $n$-th sample is designed to be

$$\Delta(l_n)_{\text{iEF}} = \nabla_{\boldsymbol{\theta}} l_n^\top \Delta \boldsymbol{\theta}_{\text{iEF}} \approx -\eta \, \|\nabla_{\boldsymbol{z}_n} l_n\|_2^2. \tag{7}$$

where $\|\nabla_{\boldsymbol{z}_n} l_n\|_2$ is the gradient norm at the model output logits-level. Note that $\|\nabla_{\boldsymbol{z}_n} l_n\|_2$ in general decreases as the $n$-th sample gets better trained because of the positive convexity of the objective of interest (cross-entropy with softmax activation). Therefore, this update formulation allows the induced per-sample loss reduction by the iEF update to be closely related to how well a sample has converged, which then greatly alleviates the *inversely-scaled projection* issue of the EF method.

A viable formulation of the iEF update $\Delta \boldsymbol{\theta}_{\text{iEF}}$ that both satisfies Eqn. (7) and relies only on the per-sample gradients is proposed as follows

$$\Delta \boldsymbol{\theta}_{\text{iEF}} = -\eta \, \nabla_{\boldsymbol{\theta}} \boldsymbol{l}^\top (\nabla_{\boldsymbol{\theta}} \boldsymbol{l} \nabla_{\boldsymbol{\theta}} \boldsymbol{l}^\top + \lambda \mathbf{I})^{-1} \boldsymbol{s}_{\text{iEF}}, \tag{8}$$

where $\boldsymbol{s}_{\text{iEF}} \in \mathbb{R}^N$ is a scaling vector defined as

$$\boldsymbol{s}_{\text{iEF}} = \begin{bmatrix} \|\nabla_{\boldsymbol{z}_1} l_1\|_2^2 & \|\nabla_{\boldsymbol{z}_2} l_2\|_2^2 & \cdots & \|\nabla_{\boldsymbol{z}_N} l_N\|_2^2 \end{bmatrix}^\top.$$

which can be obtained along with back-propagation (*e.g.* in Pytorch [32]) with negligible overhead. This improved formulation for EF is shown to be effective. In the toy examples in Fig. 1 and 4, switching from EF to iEF completely removes the distortion in the EF update vector fields. Results in Sec. 7 also validate that iEF achieves consistently better approximation quality to NG updates than both EF and SF methods in practical deep learning setups (experiment (E1)), meanwhile being robust to the choice of damping $\lambda$ across tasks and training stages (experiment (E3)).

## 5.2 Theoretical Connection to Generalised NGD

The choice of scaling vector $\boldsymbol{s}_{\text{iEF}}$ is motivated by the Gauss-Newton (GN) algorithm, which is a type of generalised NGD method [30]. The update for the GN algorithm is defined as

$$\Delta \boldsymbol{\theta}_{\text{GN}} = -\eta \, \hat{\mathbf{G}}^{-1} \nabla_{\boldsymbol{\theta}} \mathcal{L}(\boldsymbol{\theta}), \tag{9}$$

where $\hat{\mathbf{G}} = \sum_n \nabla_{\boldsymbol{\theta}} \boldsymbol{z}_n^\top \nabla_{\boldsymbol{\theta}} \boldsymbol{z}_n$ is the GN matrix. The GN algorithm can be effectively viewed as a gradient descent method on the model output logits space ($\boldsymbol{z}_n$-space) and the loss reduction induced for the $n$-th sample by the GN update is approximately

$$\Delta(l_n)_{\text{GN}} \approx -\eta \, \|\nabla_{\boldsymbol{z}_n} l_n\|_2^2, \tag{10}$$

which takes exactly the same form as the per-sample loss reduction induced by the iEF update (see Eqn. (7)). Therefore, the iEF method can be regarded as an efficient approximation to the GN algorithm in terms of its loss-reduction behaviour. In particular, it can be shown that the iEF method is equivalent to the GN algorithm for all supervised learning problems with a regression model and the exact NGD method for the least-squares regression problem (see Appendix A).

## 5.3 Convergence Analysis

In this section, two continuous time convergence analyses are provided for the non-stochastic version of the iEF method, which shows its sub-linear or linear global convergence guarantee for different types of objective functions (see Appendix C for proofs). The analysis can be considered as extensions of proofs provided in [53] to setups using non-regression models and cross-entropy objectives. The two base assumptions used by the two convergence analysis are as follows:

**Assumption 5.1.** At time $t$, the full-batch, un-damped iEF update to model parameters $\boldsymbol{\theta}(t)$ is

$$\frac{\mathrm{d}\boldsymbol{\theta}(t)}{\mathrm{d}t} = -\nabla_{\boldsymbol{\theta}(t)} \boldsymbol{l}(t)^\top [\nabla_{\boldsymbol{\theta}(t)} \boldsymbol{l}(t) \nabla_{\boldsymbol{\theta}(t)} \boldsymbol{l}(t)^\top]^{-1} \boldsymbol{s}_{\text{iEF}}(t), \tag{11}$$

**Assumption 5.2.** $\forall t > 0$, the gradient covariance matrix (or Gram matrix) $[\nabla_{\boldsymbol{\theta}(t)} \boldsymbol{l}(t)][\nabla_{\boldsymbol{\theta}(t)} \boldsymbol{l}(t)]^\top$ is always full rank.

The two main conclusions of the analysis are described below.

**Sub-linear Global Convergence for Softmax + Cross-Entropy Objective**    When the target model uses softmax output and cross-entropy loss (as described in Sec. 3), the Theorem 5.3 can be proved.

**Theorem 5.3.** *Suppose Assumption 5.2 holds, $\forall n \in \{1, \ldots, N\}$, the target probability $\hat{p}_n(t) := p_{\boldsymbol{\theta}(t)}(y = y_n | \boldsymbol{x}_n)$ for the $n$-th training sample is bounded as follows*

$$\hat{p}_n(t) > 1 - \frac{2}{t + C_0 + 1}, \tag{12}$$

*where $C_0 = \frac{1}{1 - \hat{p}_n(0)} + \log \frac{\hat{p}_n(0)}{1 - \hat{p}_n(0)}$ and $t > max\{-1 - C_0, 0\}$.*

**Linear Global Convergence for Strongly Convex Objective**    When the target model uses an $m$-strongly convex objective function [2] (see Assumption C.2, note that cross-entropy loss does not satisfy this assumption), the Theorem 5.4 can be proved.

**Theorem 5.4.** *Suppose Assumption 5.2 and C.2 holds, $\forall n \in \{1, \ldots, N\}$, the per-sample loss $l_n(t)$ for the $n$-th training sample is bounded as follows*

$$l_n(t) - l_n^\star \leq e^{-2mt}(l_n(0) - l_n^\star), \tag{13}$$

*where $l_n^\star$ is the minimum loss for the $n$-th sample.*

**Remark:** Theorem 5.4 only assumes a strongly-convex target objective *w.r.t* model output (Assumption C.2). The target loss landscape *w.r.t* model parameters can still be arbitrarily non-convex depending on the target model structure.

## 5.4    Applications of IEF

As an approximate NGD method, the exact iEF method can be used directly as an optimiser (see Algorithm 1) for models with a small parameter size. Its performance is evaluated in experiment (E2) in Sec. 7, which demonstrates competitive convergence and generalisation when compared to well-tuned baselines. Refer to Appendix. D.1 for discussions on the implementation and complexity.

More importantly, the iEF method provides an improved approximation method to the exact Fisher matrix. The iEF approximated Fisher matrix (iEF matrix) $\tilde{\mathbf{F}}^\star \in \mathbb{R}^{P \times P}$ takes the following form

$$\tilde{\mathbf{F}}^\star = \nabla_{\boldsymbol{\theta}} \boldsymbol{l}^\top \text{diag}(\boldsymbol{s}_{\text{iEF}})^{-1} \nabla_{\boldsymbol{\theta}} \boldsymbol{l}, \tag{14}$$

which can be derived from Eqn. (8) (see Appendix D.2.1). $\tilde{\mathbf{F}}^\star$ by design takes a highly similar form to the EF matrix (see Eqn 3), making them equally convenient to compute. Also, results in Sec. 7 show that updates preconditioned with the iEF matrix achieve consistently better approximation quality to NG updates than both EF and SF updates, meanwhile obviating the need for damping tuning. Consequently, the iEF matrix can be considered as a cheap yet better approximation method for the Fisher matrix than both the EF and SF methods, which opens up the possibility of improving a wide range of Fisher-based methods (not limited to optimisation methods). An example is provided in Appendix D.2.2 to demonstrate that iEF can be easily integrated into the popular empirical K-FAC optimiser [37]. Preliminary experimental results show that the integration leads to consistent improvements of the approximation quality to exact NG updates. Another example is provided in Appendix D.2.3 to demonstrate that iEF can be directly applied to improve the EF approximated Hessian used in the WoodFisher algorithms for model compression [46].

## 6    Empirical Evaluation Framework for Approximate NGD Methods

Traditional evaluation methods for quality of approximate NGD methods have high memory and time complexity, which is infeasible for large setups (see discussion in Sec. 2). In order to accurately evaluate the quality of approximate NGD methods (EF, iEF, SF *etc.*) in practical deep-learning setups, we introduce an efficient empirical evaluation framework which enables a quantitative comparison of different approximate NGD methods under large-scale setups. For a given approximate NGD method that generates an update $\Delta\boldsymbol{\theta}$, our proposed evaluation framework satisfies the following requirements: **1)** provides a quantitative evaluator $\gamma(\Delta\boldsymbol{\theta})$ that measures the (direction-wise) approximation quality to the exact NG update; **2)** the evaluation process is efficient in modern auto-grad frameworks, and it poses no constraints on the size or structure of the target model. The implementation and theoretical motivations of this empirical evaluation framework are discussed in the following sections.

## 6.1 Efficient Indicator of Approximation Quality

The proposed evaluation framework revolves around the indicator $\gamma(\Delta\boldsymbol{\theta})$ which is designed to accurately reflect the quality of an approximate NG update, while being efficient to compute. For an update $\Delta\boldsymbol{\theta}$ of interest, the proposed indicator $\gamma(\Delta\boldsymbol{\theta}) \in \mathbb{R}^+$ is defined as

$$\gamma(\Delta\boldsymbol{\theta}) = \frac{(\Delta\boldsymbol{\theta}^\top \mathbf{F}\Delta\boldsymbol{\theta})^{\frac{1}{2}}}{|\Delta\boldsymbol{\theta}^\top \nabla_{\boldsymbol{\theta}}\mathcal{L}(\boldsymbol{\theta})|}, \tag{15}$$

and the smaller the value of $\gamma(\Delta\boldsymbol{\theta})$, the better the approximation quality of $\Delta\boldsymbol{\theta}$ to the exact NG update. This indicator mainly requires computing a matrix-vector product with the exact Fisher matrix (*i.e.* $\mathbf{F}\Delta\boldsymbol{\theta}$), which can be efficiently done in modern auto-grad frameworks [32]. This allows for the application of this framework to large-scale models in practical setups. Refer to Appendix F.1 for implementation details, algorithm complexity and a comparison with traditional methods.

## 6.2 Theoretical Motivation

In this section, the proposed indicator $\gamma(\cdot)$ is justified as a theoretically appropriate evaluator of the quality of an approximate NG update. An alternative definition for the NGD is first proposed, which formulates the NG update direction with an unconstrained optimisation problem as

$$\zeta' \mathbf{F}^{-1}\nabla_{\boldsymbol{\theta}}\mathcal{L}(\boldsymbol{\theta}) = \underset{\Delta\boldsymbol{\theta}}{\arg\min}\, \gamma(\Delta\boldsymbol{\theta})^2, \tag{16}$$

where $\zeta' \in \mathbb{R}$ is an arbitrary non-zero scalar. It is shown that this alternative definition for NGD is implicitly used in the Hessian-free method [25] and the linear conjugate gradient (CG) algorithm used in Hessian-free to solve for the exact NG update is a locally optimal minimiser for $\gamma(\Delta\boldsymbol{\theta})^2$ (see Appendix F.2 for proof). Under this definition, any approximate NG update with a smaller $\gamma(\Delta\boldsymbol{\theta})^2$ is a "strictly better approximation" to the exact NG update (which is the minimiser for $\gamma(\Delta\boldsymbol{\theta})^2$).

Furthermore, $\gamma(\Delta\boldsymbol{\theta})$ can also be justified from a second-order optimisation perspective. $\frac{1}{2\gamma(\Delta\boldsymbol{\theta})^2}$ is shown to quantify the maximum achievable loss reduction for a given update direction under a local quadratic approximation of the loss function (see Appendix. F.3 for proof). Consequently, the proposed indicator can be used to accurately predict the convergence ability of a target update generation method (see experiment (E2) in Sec. 7).

# 7 Experiments

Experimental results are presented in this section. The main goal of the experiments is to verify that the behaviour of *exact* EF and iEF methods align with our theories in practical deep learning setups. Mainly three approximation methods are compared: EF, iEF and SF (an unbiased yet more expensive Fisher approximation method, see Appendix E). The exact updates of each method are generated based on Eqn. (5), (8), (49) respectively. Fifteen different setups are used to evaluate the optimisation performance and the approximation quality of these methods, including widely used parameter-efficient fine-tuning (PEFT) for pre-trained models. These include T5-base with LoRA and Prompt-Tuning on GLUE tasks [7], and ViT with LoRA for CIFAR100 [15]. PEFT of pre-trained models is investigated because it involves large-scale practical models, while having a small trainable parameter size (the implementation of *exact* EF, iEF and SF methods are memory intensive, see Appendix D.1). Please refer to Appendix H.1 for detailed experimental setups. The following three findings are demonstrated with our experiments.

(E1) The approximation quality (to exact NG updates) of EF, iEF, SF and SGD was evaluated and compared using the proposed evaluation framework in Sec. 6 on all setups. It is shown that iEF consistently improves on SGD updates and is superior to both EF and SF methods for the majority of the training stages for all setups.

(E2) The optimisation performance of EF, iEF, SF and SGD was evaluated on all setups. For each task, an additional well-tuned baseline optimiser (Adafactor/AdamW) was also compared. It is shown that iEF consistently achieves comparable or better performance than the corresponding baseline, while EF and SF suffer from unstable training to different extents.

(E3) The impact of damping on the approximation quality of EF, iEF and SF was analysed under the proposed evaluation framework. It is shown that the quality of traditional EF and SF methods relies

heavily on careful damping tuning, unlike iEF which works well with any near-zero damping across tasks and training stages.

Finally, results for an additional experiment considering a 10M parameter Multi-layer Perceptron (MLP) on the CIFAR10 [19] dataset are provided in Appendix H.7. This additional experiment further validates the aforementioned findings for a train-from-scratch setup with a much larger ($10\times$) trainable parameter size.

**E1: Approximation Quality to NG Updates**  The behaviour of updates generated with EF, iEF, SF and SGD methods were compared using the proposed empirical evaluation framework in terms of their approximation quality to exact NG updates. The updates for EF, iEF and SF were generated according to Eqns. (5), (8), and (49) respectively, and the evaluation framework follows Algorithm 4. The "un-damped" behaviour of these methods is analysed and a near-zero damping factor is used for update generation. The checkpoints at the end of each epoch generated by the baseline optimisation methods (AdamW/Adafactor) for each task were used for evaluation. In each evaluation. For each checkpoint $\boldsymbol{\theta}(t)$, indicators were computed from 100 batches of randomly picked training samples of the target task of batch size $M = 160$. The averaged indicator for each update were then evaluated $(\bar{\gamma}(\Delta\boldsymbol{\theta}_{\text{SGD}}(t)), \bar{\gamma}(\Delta\boldsymbol{\theta}_{\text{EF}}(t)), \bar{\gamma}(\Delta\boldsymbol{\theta}_{\text{iEF}}(t)), \bar{\gamma}(\Delta\boldsymbol{\theta}_{\text{SF}}(t))$, which are denoted as $\gamma_{\text{SGD}}, \gamma_{\text{iEF}}, \gamma_{\text{EF}}, \gamma_{\text{SF}}$ for simplicity). The relationship among these indicators across epochs and tasks is shown in Fig. 2. Note that results are presented for only 3 representative setups due to space limit (indicator plots for all tasks are shown in Appendix H.5.1). Three findings can be concluded from these figures: **1)** EF achieves poorer approximation quality even than SGD updates for most training stages and tasks. This is aligned with the finding in prior work that EF is a questionable approximation to NGD. **2)** The fourth plot shows that the gradient norm imbalance gets larger as training progresses. This correlates well with both the EF and SF curves, while impacting iEF less. This means that the *inversely-scaled projection* issue indeed plays a significant role in reducing the approximation quality of the EF (and SF) approximation. **3)** Comparing the first three plots, it can be seen that, for the majority of the training stages, the approximation quality follows iEF > SF > EF. IEF gives a consistently better approximation, and EF and SF are only able to beat iEF at the start of training (where a good approximation to the NG update has less impact).

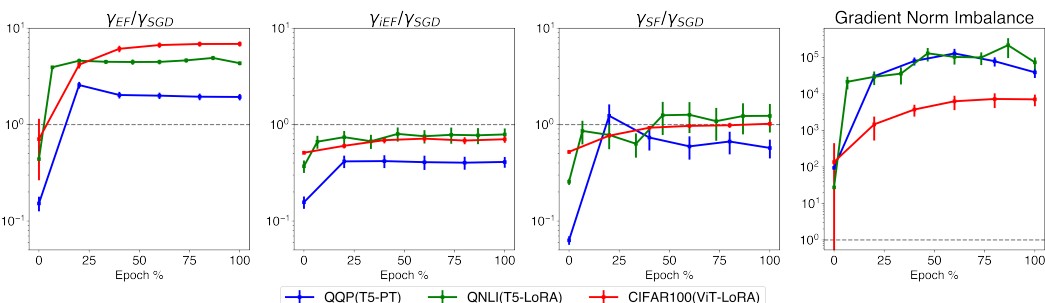

Figure 2: Four (log-scaled) ratios computed for checkpoints at various stages of training (sampled at the interval of one epoch) for 3 of the all 15 tasks. The $x$-axes represent the training stages of the model. $0\%$ means the initialised model and $100\%$ means model at the end of the last epoch. Each data point is averaged across 100 evaluations, and the error bars represent the standard deviation (1-sigma). The first plot shows $\gamma_{\text{EF}}/\gamma_{\text{SGD}}$, which denotes the relative approximation quality improvement of EF updates *w.r.t.* SGD updates (the lower the better). The second plot shows $\gamma_{\text{iEF}}/\gamma_{\text{SGD}}$, and the third plot shows $\gamma_{\text{SF}}/\gamma_{\text{EF}}$. The last plot depicts the *imbalance of gradient norms*, which is the average ratio between the maximum and minimum gradient norm for each evaluated batch (a larger value indicates more imbalanced per-sample gradient norms, which should lead to a more significant *inversely-scaled projection* issue). Overall, the approximation quality follows iEF > SF > EF.

**E2: Optimisation Performance**  The exact iEF, EF and SF methods were implemented as stochastic optimisers (following Algorithms 1, 2, 3 respectively). The same near-zero damping factor was used as in (E1). The averaged test metrics for GLUE and CIFAR100 for each optimiser are shown in Table 1 (see full test results in Table 7, validation result in Table 6, final training loss in Table 5 and training curves in Fig. 12 and 13). The following three observations can be made:
**1)** From the final training loss reported in Table 5, the ranking of final training loss generally follows iEF < AdamW/Adafactor < SGD < SF < EF (the lower the better). This ranking of training

convergence follows the ranking of indicators in (E1) closely, demonstrating the effectiveness of the empirical evaluation framework in predicting the training behaviour of optimisers. **2)** For most of the tasks, EF always suffer from unstable training (see training curves in Fig. 12 and 13), while iEF consistently reaches the lowest training loss at the end of training (even when compared with well-tuned Adafactor/AdamW baselines). This further confirms the *inversely-scaled projection* issue of EF, and demonstrates the strong convergence ability of the proposed iEF method. **3)** From test results in Table 1, it can be seen that iEF achieves the best generalisation for Prompt Tuning tasks (outperformed Adafactor in 6 out of 7 tasks). For LoRA tasks, iEF remains competitive to AdamW with each of them outperformed the other in 4 out of 8 tasks. This is likely because LoRA setups (which on average have 50 times more trainable parameters than Prompt Tuning) have a stronger reliance on regularisation and momentum, which have not been properly extended to use together with the exact iEF optimiser yet. Overall, iEF achieves the best generalisation for the majority of tasks (10 out of 15), indicating its potential as a strong optimiser for PEFT for pre-trained models.

Table 1: Average test performance of different optimisers for GLUE and CIFAR100. For GLUE tasks, the average metric results for the 7 tasks are used as the final test score. For tasks with two metrics, these metrics are averaged first [50]. For all tasks, the test result is computed for the best validation accuracy checkpoint. Refer to Table 7 for a more complete test performance report and detailed explanations on metrics.

|  | AdamW | Adafactor | SGD | EF | SF | iEF |
|---|---|---|---|---|---|---|
| **GLUE + T5 + Prompt Tuning** | - | 77.1 | 67.4 | 48.1 | 69.7 | **79.3** |
| **GLUE + T5 + LoRA** | **80.1** | - | 77.3 | 63.1 | 76.5 | 79.3 |
| **CIFAR100 + ViT + LoRA** | 93.9 | - | 91.3 | 31.0 | 92.8 | **94.3** |

**E3: Impact of Damping** As is discussed in Sec. 2, practical approximate NGD optimisers rely heavily on a good damping schedule, which is typically chosen based on empirical experience [27]. Using the proposed evaluation framework, it is straightforward to analyse the impact of damping on the approximation quality of EF, SF and iEF. For a target task, the indicator $\gamma$ *w.r.t.* damping $\lambda$ curve is computed at the start, mid-way and end of the training. Graph for an example task is shown in Fig. 3 (graphs for other tasks are provided in Appendix H.5.2). Two observations can be made:
**1)** A well-chosen damping factor significantly improves the approximation quality of EF and SF, which aligns well with observations in prior work on approximate NGD optimisers [27, 33, 34]. However, the optimal damping factor changes greatly for different tasks and training stages, which makes the damping schedule for SF or EF based optimisers necessary yet hard-to-design in practice.
**2)** Across all tasks and training stages, iEF robustly achieves great approximation quality with a near-zero damping factor. More importantly, its approximation quality is consistently better than EF method, and is comparable to the optimally-damped SF method (which is much more expensive, particularly when the cost of damping tuning is considered). Overall, iEF can be considered a cheaper, higher-quality and more robust alternative to both the EF and SF approximation methods.

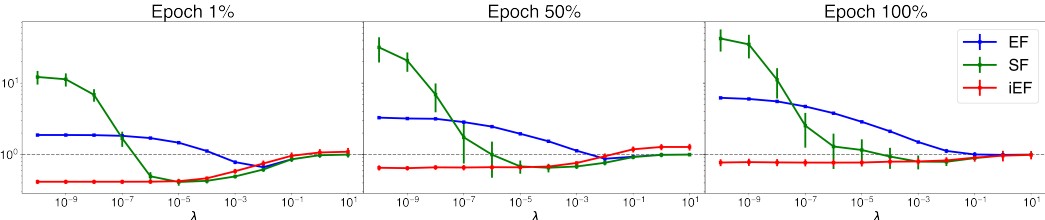

Figure 3: Approximation quality (relative to SGD) of EF, SF and iEF methods *w.r.t.* damping factor $\lambda$ at different training stages of task CoLA+T5+LoRA. $x$-axes show the value of the damping factor, $y$-axes depict the relative approximation quality improvement of the target update method *w.r.t.* SGD (the lower the better). Each data point is averaged across 100 evaluations, and the error-bars represent the standard deviation (1-sigma). The first plot is for checkpoint saved at the end of the first training epoch, the second plot for the mid-way epoch and the third plot for the final epoch. It can be observed that iEF achieves the best approximation quality robustly for any near-zero $\lambda$. In contrast, $\lambda$ has a non-linear impact on both SF and EF. When optimally tuned, an EF update can achieve better approximation quality than SGD, and an SF update can achieve comparable quality to iEF. However, the optimal damping factor for EF and SF changes greatly with training stages (and tasks).

# 8 Conclusions and Future Work

This paper presents the iEF method, which addresses the *inversely-scaled projection* issue of the EF approximation for NGD, meanwhile maintaining the implementational convenience. A novel empirical evaluation framework for the quality of general approximate NGD update is also proposed, which enables quantified comparison of approximate NGD methods in large deep learning setups[1]. Based on the experiments with practical PEFT of pre-trained models for NLP and CV classification tasks, the exact iEF optimiser shows superior convergence and generalisation for majority of the tasks, supporting the applicability of iEF directly as an optimiser. Further evaluation on approximation quality concludes that iEF achieves consistently better approximation quality than both EF and SF. The iEF method also demonstrates the superior property of being robust to the choice of damping factor across different tasks and training stages.

As is discussed in Sec. 5.4, the iEF method can be viewed not only as an improved approximate NGD optimiser, but also as an improved approximation method for the exact Fisher matrix in general. This opens up many opportunities of future work to improve a wide range of Fisher-based methods (not limited to optimisation methods). Some example applications include improving the empirical K-FAC optimiser [37, 33] (which has shown promising results in preliminary experiments) and improving the WoodFisher algorithm for model compression [46].

## Acknowledgements

Xiaodong Wu is in part funded by the Cambridge Trust, a donation from Meta Systems and Christ's College, Cambridge. This work has in part been performed using resources provided by the Cambridge Tier-2 system operated by the University of Cambridge Research Computing Service (www.hpc.cam.ac.uk) funded by EPSRC Tier 2 capital grant EP/T022159/1.

---

[1]Codebase will be publicly released.

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

# A    Relation between iEF, GN and NGD in Different Machine Learning Setups

The connection between iEF, GN and NGD methods is discussed for different common machine-learning scenarios. It is explained that for a scalar output (regression) model, the iEF method and the GN algorithm is equivalent. Furthermore, for a least-squares problem, the iEF, GN and NGD methods are all equivalent.

## A.1    Regression Model Problem

In this section, a supervised learning problem for a target model with only a scalar output (*i.e.* regression model) is discussed. The definition of the setup is as follows.

Consider a target regression model $f_{\boldsymbol{\theta}}(\cdot) \in \mathbb{R}^P \to \mathbb{R}$, where $\boldsymbol{\theta} \in \mathbb{R}^P$ is the trainable parameters of size $P$. Given $N$ *i.i.d.* training samples $(\boldsymbol{x}_n, y_n)_{n=1}^N$ (where $\boldsymbol{x}_n$ is the input feature vector and $y_n \in \mathbb{R}$ is the scalar label), the output of the model for the $n$-th sample is $z_n = f_{\boldsymbol{\theta}}(\boldsymbol{x}_n)$ where $z_n \in \mathbb{R}$ is a scalar. For the $n$-th sample, the per-sample loss $l_n$ is defined as

$$l_n = \mathcal{F}^{\mathrm{obj}}(z_n, y_n), \tag{17}$$

where $\mathcal{F}^{\mathrm{obj}}(\cdot) \in \mathbb{R} \to \mathbb{R}$ is the per-sample objective function, and the accumulated loss $\sum_n l_n$ is to be minimised. Some examples of this problem setup include least-squares regression problems where a mean-square error is used as the loss function, and binary classification problems where the Sigmoid-activation plus binary cross-entropy is used as the loss function.

Recall the definition of the iEF matrix in Eqn. (14). In this problem setup, the iEF matrix can be computed as follows

$$\tilde{\mathbf{F}}^{\star} = \sum_n \|\frac{\mathrm{d}l_n}{\mathrm{d}z_n}\|_2^{-2} (\frac{\mathrm{d}l_n}{\mathrm{d}z_n} \nabla_{\boldsymbol{\theta}} z_n)(\frac{\mathrm{d}l_n}{\mathrm{d}z_n} \nabla_{\boldsymbol{\theta}} z_n)^{\top} = \sum_n \nabla_{\boldsymbol{\theta}} z_n \nabla_{\boldsymbol{\theta}} z_n^{\top} = \hat{\mathbf{G}}, \tag{18}$$

which takes the same form as the GN matrix. Therefore, the iEF and GN methods are equivalent to supervised learning problems of regression models.

## A.2    Least-Squares Problem

The least-squares problem is a representative example of supervised learning problems of regression models, which includes the toy example shown in Fig. 1. The definition of this type of problem follows most of the definitions in Appendix A.1, apart from the per-sample objective function, which is elaborated as follows. For the least-squares problem, the objective function is defined as follows

$$\mathcal{F}^{\mathrm{obj}}(z_n, y_n) = \frac{1}{2}(z_n - y_n)^2. \tag{19}$$

The Fisher matrix for the least-squares problem is defined as follows [47]

$$\mathbf{F} = \sum_n \nabla_{\boldsymbol{\theta}} z_n \nabla_{\boldsymbol{\theta}} z_n^{\top} = \tilde{\mathbf{F}}^{\star} = \hat{\mathbf{G}}, \tag{20}$$

which coincides with the iEF and GN matrix in this problem setup. Consequently, the iEF, GN and NGD methods are equivalent for least-squares problems. This is verified in the second plot of Fig. 1 where NGD and iEF share the same update vector field.

# B    Visualisation for Linear Least-Squares Problem with Two Training Samples

**Setup Description**    In this section, the Fig. 1 referenced in Sec. 4.2 is explained in detail. The vector field graph is based on a simple linear least-squares regression problem, with 2 training samples and 2 trainable parameters.

The linear least-squares problem is chosen for visualisation because it is not only a highly representative machine learning setup [47, 20], but also there has been a precedent of using this problem to visualise the distortion in EF (see Figure 1 in [20]). The trainable parameter size of $P = 2$ is chosen

to facilitate the 2D visualisation, and the training sample size of $N = 2$ (*i.e.* $N \leq P$) is chosen to better match the practical deep learning scenarios where the model is over-parameterised.

The target linear model is formulated as $f_{\boldsymbol{\theta}}(x_n) = \boldsymbol{\theta}^\top [x_n, 1]^\top = \theta_0 + \theta_1 x_n$ with $x_n \in \mathbb{R}$ and $\boldsymbol{\theta} \in \mathbb{R}^2$. The two training samples are $(x_n, y_n)_{n=1}^N = \{(0,0), (1,0)\}$ (with $N = 2$) and the target loss to be minimised is $\mathcal{L}(\boldsymbol{\theta}) = \sum_{n=1}^2 \frac{1}{2}[y_n - f_{\boldsymbol{\theta}}(x_n)]^2$. It is obvious that the optimal parameter (at global minimum) is $\boldsymbol{\theta}^\star = [0, 0]^\top$.

The update vector fields are generated using the following definition:

$$\begin{cases} \Delta\boldsymbol{\theta}_{\text{SGD}} & = \nabla_{\boldsymbol{\theta}}\mathcal{L}(\boldsymbol{\theta}) \\ \Delta\boldsymbol{\theta}_{\text{NGD}} & = (\mathbf{F} + \lambda_{\text{NGD}}\mathbf{I})^{-1}\nabla_{\boldsymbol{\theta}}\mathcal{L}(\boldsymbol{\theta}) \\ \Delta\boldsymbol{\theta}_{\text{EF}} & = (\tilde{\mathbf{F}} + \lambda_{\text{EF}}\mathbf{I})^{-1}\nabla_{\boldsymbol{\theta}}\mathcal{L}(\boldsymbol{\theta}) \\ \Delta\boldsymbol{\theta}_{\text{iEF}} & = (\tilde{\mathbf{F}}^\star + \lambda_{\text{iEF}}\mathbf{I})^{-1}\nabla_{\boldsymbol{\theta}}\mathcal{L}(\boldsymbol{\theta}). \end{cases} \quad (21)$$

The EF matrix $\tilde{\mathbf{F}}$ follows the definition in Eqn. (3). The iEF matrix $\tilde{\mathbf{F}}^\star$ and Fisher matrix $\mathbf{F}$ shares the same definition as shown in Eqn. (20) (see Appendix A.2). For this toy problem, the damping factors $\lambda_{\text{iEF}}$ and $\lambda_{\text{NGD}}$ are set to zero. The damping factor $\lambda_{\text{EF}}$ is set to zero everywhere, apart from when one of the per-sample gradient gives a norm of 0, a damping factor of $1 \times 10^{-4} \max(\text{diag}(\tilde{\mathbf{F}}))$ is used to facilitate inversion. Finally, in the third plot of Fig. 1, the EF update vectors are normalised for better visualisation, because EF updates have hugely different scales across the contour plots. In the zoomed view next to the third plot, the EF update vectors are not normalised, which should give a better demonstration of the inverse scaling issue of EF updates.

The same set of 5 initial parameters (shown as black solid octagons) are used to generate 5 training trajectories (coloured curves) on each plot in Fig. 1. Each follows the corresponding update vector fields (each step follows the direction of the update vector field, and is normalised to $1 \times 10^{-2}$). To demonstrate how each sample affects the EF updates, the optimal parameter set for each training sample (*i.e.* when $\frac{1}{2}[y_n - f_{\boldsymbol{\theta}}(x_n)]^2 = 0$) is added on all plots in Fig. 1 as dashed lines. For training sample 1: $(0, 0)$, the optimal parameter set is $\theta_0 = 0$, which is shown as horizontal dashed lines. For training sample 2: $(1, 0)$, the optimal parameter set is $\theta_0 + \theta_1 = 0$, which is shown as diagonal dashed lines.

**Observations**   By observing the behaviour of the vector fields for different updates on the loss landscape, along with the 5 sampled trajectories in Fig. 1, it can be seen that all methods successfully reached the global minimum, with the NGD/iEF updates having the most efficient training trajectory. However, the EF method has highly distorted update vector field and training trajectories due to the *inversely-scaled projection* issue discussed in Sec. 4.1. The following conclusions can be drawn for EF updates:

1. **Directional bias towards converged samples:**   The EF update vector field is easily "attracted" to the dashed lines because the EF update is strongly affected by the better-converged sample (due to inverse scaling). This leads to a distorted update vector field and ineffective training trajectories. In particular, when per-sample gradients have a highly imbalanced norm near the dashed lines, the EF update vector field becomes almost parallel to the loss contour. This causes them to deviate significantly from both the gradient descent direction and the optimal direction (NG update direction), and oscillations can be observed along the dashed line in the zoomed view.

2. **Inversely-scaled update norm:**   EF updates have larger norms when either of the training samples is nearly converged (see the zoomed plot). In fact, the EF training trajectories managed to converge because we normalised every update to a small norm of $1 \times 10^{-2}$, which is not needed for SGD, NGD and iEF methods. Consequently, it is likely that a sophisticated learning rate scheduler is necessary when training with the EF method in practical deep learning setups (as is mentioned in [20]).

**Motivation**   As is stated in Sec. 3, the paper mainly focuses on classification setups, but a visualisation of a regression setup in Fig. 1 is used in the main paper for the following reasons: **1)** The least-squares regression problem is commonly used when analysing NGD and EF in the literature [20, 47]. Particularly, our visualisation follows a similar setup to [20], which is an important related work regarding the limitations of EF. Overall, the toy regression example allows for consistency with

the literature. **2)** The 2-datum least-squares regression problem have several nice properties. There exists a unique global minimum; the NG update can find the global minimum in one step, and the advantage over all other updates is self-evident; the iEF update and NG update has the same form; the distortion of the EF update is extreme. All of these properties make the visualisation in Fig. 1 much more straightforward to understand than a visualisation for a classification setup.

**Additional Visualisation for Logistic Regression Problem** Given that the paper focuses on classification setups, it is important to also include a visualisation for a toy classification setup. Consequently, an additional visualisation, in a similar fashion, that compares Fisher, iEF and EF as pre-conditioners in a toy classification setup is provided in Fig. 4. This figure considers a 2-datum logistic regression setup and a more detailed description is provided in the caption. This new visualisation demonstrates consistent results to that of Fig. 1. Particularly, it can be observed that EF updates deviate from the optimal decision boundary because it is biased toward further optimising the better classified datum (*i.e.* the datum with a lower CE loss). Also, EF updates become larger when either of the datum achieves a small loss. Meanwhile, SGD, NGD and iEF update approaches the optimal decision boundary consistently, and arguably NGD and iEF reach the optimal decision boundary more effectively than SGD.

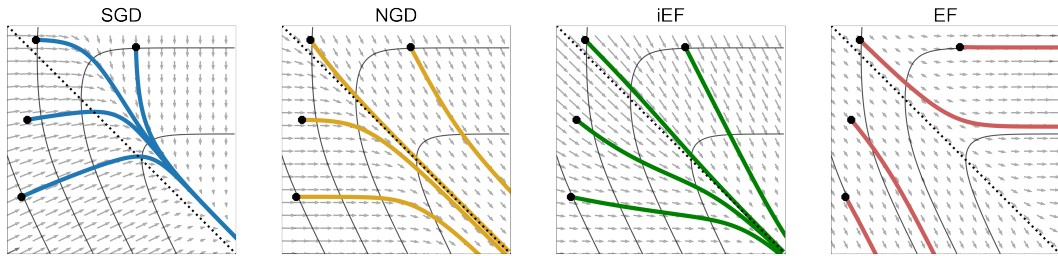

Figure 4: A visual comparison of Fisher, iEF and EF as pre-conditioners for a logistic regression problem (classifying two 1D datum $x_0 = 0, x_1 = 2$ into two classes). The target model is $p_{\boldsymbol{\theta}}(x_n) = \sigma(\theta_0 + \theta_1 x_n)$, where $\sigma(\cdot)$ is the Sigmoid function and CE loss is used, which follows the problem setup description in Sec. 3. This figure is different from Fig. 1 in 3 aspects: **1)** iEF and NG updates are no longer identical, and are presented in separate plots; **2)** There is no global minima, but the model achieves lower loss when moving further down the bottom-right corner; **3)** The dashed line now represents the optimal parameter set for a decision boundary of $x = 1$. The training trajectory of EF is still ill-behaved, meanwhile both NG and iEF updates move towards the optimal decision boundary smoothly.

## C Convergence Analysis

The proofs for Theorem. 5.3 and Theorem. 5.3 are provided in this section. The proofs follow and extend the continuous-time analysis framework in [53] and [2].

**Justification of Full-rank Gram Matrix Assumption** Recall Assumption 5.2 which assumes $[\nabla_{\boldsymbol{\theta}(t)} \boldsymbol{l}(t)][\nabla_{\boldsymbol{\theta}(t)} \boldsymbol{l}(t)]^\top$ to be full-rank throughout the training process. This is equivalent to assuming $\nabla_{\boldsymbol{\theta}(t)} \boldsymbol{l}(t) \in \mathbb{R}^{N \times P}$ always have full row rank. This is considered a mild assumption for practical deep learning because:

1. It is generally true that for each individual per-sample gradient $\nabla_{\boldsymbol{\theta}(t)} l_n(t)$, the norm of the gradient will be zero if and only if $\nabla_{\boldsymbol{z}_n(t)} l_n(t) = \mathbf{0}$ (*i.e.* at the minimum of that sample). Therefore, it is unlikely that there exists zero norm per-sample gradients in $\nabla_{\boldsymbol{\theta}(t)} \boldsymbol{l}(t)$ during training.

2. The target model is usually highly over-parameterised with $P \gg N$ and the model is highly complex. It is in general unlikely that $\nabla_{\boldsymbol{\theta}(t)} \boldsymbol{l}(t)$ is rank-deficient, as long as there are no duplicate training samples.

**Ideal Per-sample Loss Change for IEF** Given Assumption 5.1 and Assumption 5.2, the following Lemma can be derived

**Lemma C.1.** *Suppose Assumption 5.2 holds, $\forall n \in \{1, \ldots, N\}$, the loss reduction induced on the n-th training sample by the iEF update follows*

$$\frac{\mathrm{d}l_n(t)}{\mathrm{d}t} = -\|\nabla_{\boldsymbol{z}_n(t)} l_n(t)\|_2^2. \tag{22}$$

*Proof of Lemma. C.1:* $\frac{\mathrm{d}\boldsymbol{l}(t)}{\mathrm{d}t}$ can be computed as follows

$$\frac{\mathrm{d}\boldsymbol{l}(t)}{\mathrm{d}t} = \nabla_{\boldsymbol{\theta}(t)}\boldsymbol{l}(t)\frac{\mathrm{d}\boldsymbol{\theta}(t)}{\mathrm{d}t} = -\nabla_{\boldsymbol{\theta}(t)}\boldsymbol{l}(t)\nabla_{\boldsymbol{\theta}(t)}\boldsymbol{l}(t)^\top [\nabla_{\boldsymbol{\theta}(t)}\boldsymbol{l}(t)\nabla_{\boldsymbol{\theta}(t)}\boldsymbol{l}(t)^\top]^{-1}\boldsymbol{s}_{\mathrm{iEF}}(t) = -\boldsymbol{s}_{\mathrm{iEF}}(t). \tag{23}$$

Therefore, $\frac{\mathrm{d}l_n(t)}{\mathrm{d}t} = [\frac{\mathrm{d}\boldsymbol{l}(t)}{\mathrm{d}t}]_n = -\|\nabla_{\boldsymbol{z}_n(t)} l_n(t)\|_2^2$.

**Global Sub-linear Convergence for Softmax + CE Objective Function**  The proof for Theorem 5.3 is provided here. In this analysis, the supervised learning setup with softmax + CE objective function is considered (as described in Sec. 3). This is a setup commonly used in practical deep learning, making this convergence analysis practically relevant.

As stated in Sec. 3, the target model is assumed to use a softmax activation function. For the $n$-th sample at time step $t$, $\boldsymbol{z}_n(t) = f_{\boldsymbol{\theta}(t)}(\boldsymbol{x}_n)$, where $\boldsymbol{z}_n(t) \in \mathbb{R}^C$ is the output logits. The output probability for class $c$ can be computed from the logits using the softmax activation: $[p_n(c)](t) = [\boldsymbol{\sigma}_{\mathrm{SM}}(\boldsymbol{z}_n(t))]_c$ where $\boldsymbol{\sigma}_{\mathrm{SM}}(\cdot) : \mathbb{R}^C \to \mathbb{R}^C$ is the softmax activation function.

The per-sample loss in this setup is therefore defined as

$$l_n(t) := -\log[p_n(y_n)](t) := -\log \hat{p}_n(t) = -\log \left[\boldsymbol{\sigma}_{\mathrm{SM}}(\boldsymbol{z}_n(t))\right]_{y_n}, \tag{24}$$

where $\left[\boldsymbol{\sigma}_{\mathrm{SM}}(\boldsymbol{z}_n)\right]_{y_n}$ represents the output probability for the target class $y_n$. It can be seen that the lowest loss for sample $n$ tends to 0, which is achieved when $\hat{p}_n(t) \to 1$.

The gradient of $l_n(t)$ w.r.t. $\boldsymbol{z}_n(t)$ can be computed as

$$\left[\nabla_{\boldsymbol{z}_n(t)} l_n(t)\right]_c = \begin{cases} -[p_n(c)](t) & c \neq y_n, \\ 1 - \hat{p}_n(t) & c = y_n. \end{cases} \tag{25}$$

The norm of the gradient satisfies the following inequality

$$\|\nabla_{\boldsymbol{z}_n(t)} l_n(t)\|_2^2 \geq [1 - \hat{p}_n(t)]^2. \tag{26}$$

Combining Lemma. C.1 with the definition in Eqn. (24), the following equation can be obtained

$$\frac{\mathrm{d}l_n(t)}{\mathrm{d}t} = -\frac{\mathrm{d}\log \hat{p}_n(t)}{\mathrm{d}t} = -\frac{1}{\hat{p}_n(t)}\frac{\mathrm{d}\hat{p}_n(t)}{\mathrm{d}t} = -\|\nabla_{\boldsymbol{z}_n(t)} l_n(t)\|_2^2. \tag{27}$$

Combining Eqn. (26)(27), the following inequality can be obtained

$$\frac{1}{[\hat{p}_n(t)][1 - \hat{p}_n(t)]^2}\mathrm{d}\hat{p}_n(t) \geq \mathrm{d}t. \tag{28}$$

Integrating on both sides gives

$$\frac{1}{1 - \hat{p}_n(t)} + \log \frac{\hat{p}_n(t)}{1 - \hat{p}_n(t)} - C_0 \geq t. \tag{29}$$

where $C_0 = \frac{1}{1-\hat{p}_n(0)} + \log \frac{\hat{p}_n(0)}{1-\hat{p}_n(0)}$. It is known that $x > \log x$ for $x > 0$, therefore the inequality can be further relaxed to

$$\frac{1}{1 - \hat{p}_n(t)} + \frac{\hat{p}_n(t)}{1 - \hat{p}_n(t)} > t + C_0, \tag{30}$$

which is equivalent to

$$\frac{2}{1 - \hat{p}_n(t)} > t + C_0 + 1. \tag{31}$$

Now consider a large $t$, s.t. $t + C_0 + 1 > 0$, a bound can be provided for $\hat{p}_n(t)$ as follows

$$\hat{p}_n(t) > 1 - \frac{2}{t + C_0 + 1}. \tag{32}$$

This shows that iEF guarantees global sub-linear convergence for target probability $\hat{p}_n \to 1$ for every training sample. It equivalently implies that iEF guarantees sub-linear convergence to the global minimum of the accumulated cross-entropy loss (achieved when $\forall n = \{1, \ldots N\}, \hat{p}_n = 1$), given the per-sample objective function is the softmax + cross-entropy combination.

### C.1 Global Linear Convergence for Strongly Convex Objective Functions

The proof for Theorem 5.4 is provided here. Unlike for Theorem 5.3, where the model is assumed to use a softmax + cross-entropy loss function, this theorem is applicable to a group of common loss functions: $m$-strongly convex objective functions.

The details of the setup used here mostly follow that in Sec. 3. The differences are described as follows. Given $N$ *i.i.d.* training samples $(\boldsymbol{x}_n, y_n)_{n=1}^N$ (where $\boldsymbol{x}_n$ is the input feature vector and $y_n$ is a label of arbitrary form), the output of the model for the $n$-th sample is $\boldsymbol{z}_n = f_{\boldsymbol{\theta}}(\boldsymbol{x}_n)$ where $z_n \in \mathbb{R}^C$ is a general model output vector (no longer logits vector). A $m$-strongly convex objective loss function $\mathcal{F}_{\text{obj}}(\cdot)$ is used to compute the per-sample loss for the $n$-th sample $l_n \in \mathbb{R}$ as follows

$$l_n = \mathcal{F}_{\text{obj}}(\boldsymbol{z}_n, y_n) := \mathcal{F}_{\text{obj}}(\boldsymbol{z}_n), \tag{33}$$

where the label $y_n$ is omitted for simplicity. Finally, the following accumulated loss is to be minimised

$$\mathcal{L}(\boldsymbol{\theta}) = \sum_n l_n. \tag{34}$$

In addition to Assumption 5.1 and 5.2, an additional assumption is made for the objective loss function as follows

**Assumption C.2.** For the $n$-th sample, the objective loss function $\mathcal{F}_{\text{obj}}(\boldsymbol{z}_n)$ is assumed to be $m$-strongly convex on the model output space $\boldsymbol{z}_n$, which then satisfies the following Polyak-Lojasiewicz inequality [5]

$$\mathcal{F}_{\text{obj}}(\boldsymbol{z}_n) - \mathcal{F}_{\text{obj}}(\boldsymbol{z}_n^\star) \leq \frac{1}{2m} \|\nabla_{\boldsymbol{z}_n} \mathcal{F}_{\text{obj}}(\boldsymbol{z}_n)\|^2, \tag{35}$$

where $\mathcal{F}_{\text{obj}}(\boldsymbol{z}_n^\star)$ is the global minimum of the loss for the $n$-th sample. For simplicity, the notation $l_n^\star$ is used in place of $\mathcal{F}_{\text{obj}}(\boldsymbol{z}_n^\star)$. The inequality is therefore rewritten as follows

$$l_n - l_n^\star \leq \frac{1}{2m} \|\nabla_{\boldsymbol{z}_n} l_n\|^2. \tag{36}$$

Assumption. C.2 is quoted from [2], which covers a wide range of loss functions including mean-square-error. Note that under this assumption, both per-sample losses and accumulated loss can still have an arbitrarily non-convex landscape on the parameter space.

Based on Lemma C.1, when training with iEF updates, for the $n$-th sample, the loss change is

$$\frac{\mathrm{d}l_n(t)}{\mathrm{d}t} = -\|\nabla_{\boldsymbol{z}_n(t)} l_n(t)\|_2^2 \tag{37}$$

Using the Polyak-Lojasiewicz inequality in Assumption. C.2, the following inequality can be obtained

$$\frac{\mathrm{d}(l_n(t) - l_n^\star)}{\mathrm{d}t} \leq -2m(l_n(t) - l_n^\star) \Rightarrow \frac{1}{(l_n(t) - l_n^\star)} \frac{\mathrm{d}(l_n(t) - l_n^\star)}{\mathrm{d}t} \leq -2m. \tag{38}$$

Integrating on both sides gives the following

$$l_n(t) - l_n^\star \leq e^{-2mt}(l_n(0) - l_n^\star), \tag{39}$$

which shows that iEF pre-conditioned gradient flow linearly converges to the global minimum for every sample. This then implies that iEF has a global linear convergence guarantee for an accumulated loss $\mathcal{L}(\boldsymbol{\theta})$.

## D Discussion on Practical Applications of IEF

### D.1 Stochastic IEF/EF Optimiser

As is mentioned in Sec. 5.4, the stochastic version of the exact iEF method can be directly used as an optimiser. It is described in Algorithm 1. In this section, the implementation of this exact iEF optimiser is discussed, along with its computation and memory complexity. Note that the stochastic exact EF optimiser can be constructed in a similar form in Algorithm 2. The following discussions should also apply to the exact EF optimiser.

**Algorithm 1** Stochastic Optimisation with Exact IEF
***
**Require:** All $N$ training samples, initial model parameters $\boldsymbol{\theta}(0)$
**for** $t = 0$ **to** $T - 1$ **do**
    Sample Mini-batch $\mathcal{M}(t)$ with size $M$
    Perform forward pass on batch $\mathcal{M}(t)$ to obtain the per-sample loss vector $\boldsymbol{l}(t)$
    Perform back-propagation to obtain Jacobian $\nabla_{\boldsymbol{\theta}(t)}\boldsymbol{l}(t)$ and logits gradient norm vector $\boldsymbol{s}_{\text{iEF}}(t)$
    Compute iEF update $\Delta\boldsymbol{\theta}_{\text{iEF}}(t) = \nabla_{\boldsymbol{\theta}(t)}\boldsymbol{l}(t)^\top (\nabla_{\boldsymbol{\theta}(t)}\boldsymbol{l}(t)\nabla_{\boldsymbol{\theta}(t)}\boldsymbol{l}(t)^\top + \lambda\mathbf{I})^{-1}\boldsymbol{s}_{\text{iEF}}(t)$: Eqn. (8)
    Update model $\boldsymbol{\theta}(t + 1) = \boldsymbol{\theta}(t) - \eta\Delta\boldsymbol{\theta}_{\text{iEF}}(t)$
**end for**
***

**Algorithm 2** Stochastic Optimisation with Exact EF
***
**Require:** All $N$ training samples, initial model parameters $\boldsymbol{\theta}(0)$
**for** $t = 0$ **to** $T - 1$ **do**
    Sample Mini-batch $\mathcal{M}(t)$ with size $M$
    Perform forward pass on batch $\mathcal{M}(t)$ to obtain the per-sample loss vector $\boldsymbol{l}(t)$
    Perform back-propagation to obtain Jacobian $\nabla_{\boldsymbol{\theta}(t)}\boldsymbol{l}(t)$
    Compute EF update $\Delta\boldsymbol{\theta}_{\text{EF}}(t) = \nabla_{\boldsymbol{\theta}(t)}\boldsymbol{l}(t)^\top (\nabla_{\boldsymbol{\theta}(t)}\boldsymbol{l}(t)\nabla_{\boldsymbol{\theta}(t)}\boldsymbol{l}(t)^\top + \lambda\mathbf{I})^{-1}\mathbf{1}$: Eqn. (5)
    Update model $\boldsymbol{\theta}(t + 1) = \boldsymbol{\theta}(t) - \eta\Delta\boldsymbol{\theta}_{\text{EF}}(t)$
**end for**
***

**Implementation Details**   There are two main aspects of the implementation of this optimiser that are non-trivial. These are discussed respectively as follows:

1. **Jacobian matrix $\nabla_{\boldsymbol{\theta}}\boldsymbol{l}$:**

   The computation of $\nabla_{\boldsymbol{\theta}}\boldsymbol{l}$ is effectively collecting the per-sample gradients for a batch during the back-propagation process. In Pytorch [32], the per-sample gradients are readily computed during back-propagation, but are usually accumulated along the batch dimension to compute the total gradient, and are not available for collection directly. Therefore, additional backward hooks need to be attached to trainable modules to store these per-sample gradients during the back-propagation process. This is a standard procedure used in most approximate NGD optimisers [27, 4, 43]. Our implementation is partially based on this K-FAC implementation. Note that this additional procedure of computing per-sample gradients is negligible *w.r.t.* forward and backward of large pre-trained models in PEFT setups, making iEF/EF have comparable speed as standard AdamW/Adafactor/SGD optimisers.

2. **Logits gradient norm vector $\boldsymbol{s}_{\text{iEF}}$:**

   During back-propagation in Pytorch [32], the gradient of logits $\nabla_{\boldsymbol{z}_n}l_n$ is already computed, but is not stored because logits vector $\boldsymbol{z}_n$ is not a leaf-node. This can be easily changed by calling the ".retain_grad()" method on the logits vector. Although this operation is non-standard to common approximate NGD optimisers, it adds negligible cost to standard back-propagation (in general deep learning setup, not limited to PEFT) and its effect on speed can be ignored.

**Time and Memory Complexity**   Given a batch size of $M$, and trainable parameters of size $P$, assume the per-sample gradients have already been computed through backpropagation, the time complexity of computing each update is $O(M^3 + M^2P)$, and the memory complexity is $O(M^2 + MP)$. Due to model over-parameterisation, we have $P \gg M$. Then, the time complexity becomes $O(M^2P)$ and memory complexity becomes $O(MP)$. In practical deep learning frameworks such as Pytorch [32], the limiting factor for the applicability of such an exact EF-like method is the memory complexity $O(MP)$, which is essentially the storage of the $M$ per-sample gradients involved in the computation of exact iEF updates. It is therefore only possible to apply exact EF-like methods to models with small trainable parameter sizes (either full tuning of small models or parameter-efficient fine-tuning of large models) or small batch size $M$. This is the reason why exact EF is never directly used to optimise modern deep learning models, and additional approximation is always necessary (e.g. K-FAC [27] or SVD-based pruning [43]). Nevertheless, given the rise of large pre-trained models [48, 6, 39, 38], parameter-efficient fine-tuning has gained traction [15, 16, 7] and

direct application of exact iEF may still be beneficial (as the trainable parameter size is usually $< 1\%$ of the pre-trained model size).

## D.2 Improving Existing Fisher-based Methods with IEF

It is mentioned in Appendix D.1 that exact iEF as an optimiser is limited by its memory complexity. Given the descent training performance of exact iEF optimiser, it would be interesting to observe the performance of iEF on more general setups. As is mentioned in Sec. 5.4, it can be achieved by incorporating iEF into existing EF-based optimisers such as empirical K-FAC [27]. In this section, an improvement to the EF-based K-FAC optimiser with iEF is proposed, which can act as a starting point for future work in improving other approximate empirical NGD methods.

### D.2.1 Expression of IEF Matrix

The iEF matrix can be computed according to Eqn. (14): $\mathbf{F}^\star = \nabla_{\boldsymbol{\theta}} l^\top \mathrm{diag}(\boldsymbol{s}_{\mathrm{iEF}})^{-1} \nabla_{\boldsymbol{\theta}} l$. This can be derived from Eqn. (8) using the identity $(\mathbf{UV} + \lambda\mathbf{I})^{-1}\mathbf{U} = \mathbf{U}(\mathbf{VU} + \lambda\mathbf{I})^{-1}$. For simplicity of derivation, we denote $\mathbf{J} = \nabla_{\boldsymbol{\theta}} l$ and assume $\mathbf{J}$ is a square matrix and is full-rank. Also, we denote $\mathbf{S} = \mathrm{diag}(\boldsymbol{s}_{\mathrm{iEF}})^{\frac{1}{2}}$ and assume the diagonal matrix $\mathbf{S}$ to be full-rank. Therefore, the proposed iEF matrix can be expressed as $\mathbf{F}^\star = \mathbf{J}^\top \mathbf{S}^{-2} \mathbf{J}$. The following derivation can be made

$$
\begin{aligned}
(\mathbf{F}^\star)^{-1}(\nabla_{\boldsymbol{\theta}} l^\top \mathbf{1}) &= (\mathbf{J}^\top \mathbf{S}^{-2}\mathbf{J})^{-1}\mathbf{J}^\top\mathbf{1} = [(\mathbf{J}^\top\mathbf{S}^{-1})(\mathbf{S}^{-1}\mathbf{J})]^{-1}(\mathbf{J}^\top\mathbf{S}^{-1})(\mathbf{S1}) \\
&= (\mathbf{J}^\top\mathbf{S}^{-1})[(\mathbf{S}^{-1}\mathbf{J})(\mathbf{J}^\top\mathbf{S}^{-1})]^{-1}\mathbf{S1} = \mathbf{J}^\top\mathbf{S}^{-1}\mathbf{S}[\mathbf{JJ}^\top]^{-1}\mathbf{SS1} \qquad (40) \\
&= \mathbf{J}^\top[\mathbf{JJ}^\top]^{-1}\mathbf{S}^2\mathbf{1} = \mathbf{J}^\top[\mathbf{JJ}^\top]^{-1}\boldsymbol{s}_{\mathrm{iEF}}.
\end{aligned}
$$

This shows that the update pre-conditioned by $\mathbf{F}^\star$ indeed is the same as the iEF update described in Eqn. (8).

The expression for the iEF matrix in Eqn. (14) can be alternatively rewritten as follows

$$
\mathbf{F}^\star = \nabla_{\boldsymbol{\theta}} l^\top \mathrm{diag}(\boldsymbol{s}_{\mathrm{iEF}})^{-1}\nabla_{\boldsymbol{\theta}} l = \sum_n (\|\nabla_{\boldsymbol{z}_n} l_n\|_2^{-1}\nabla_{\boldsymbol{\theta}} l_n)(\|\nabla_{\boldsymbol{z}_n} l_n\|_2^{-1}\nabla_{\boldsymbol{\theta}} l_n)^\top, \qquad (41)
$$

which differs from the EF matrix in Eqn. (3) in that each per-sample gradient is re-scaled with $\|\nabla_{\boldsymbol{z}_n} l_n\|^{-1}$. Such simple scaling is easy to implement in most approximate empirical NGD optimisers.

### D.2.2 Improving Empirical K-FAC with IEF

The empirical K-FAC [37] is a widely used version [33, 34, 49, 31] of the original K-FAC method [27]. Its formulation can be described as follows.

Assume there are $n$ samples in the target batch. For one fully connected layer in a target model, the weight matrix is denoted by matrix $\mathbf{W} \in \mathbb{R}^{m \times k}$, the (batched) input is denoted by $\boldsymbol{a} \in \mathbb{R}^{n \times m}$ and the (batched) output is denoted by row vector $\boldsymbol{c} \in \mathbb{R}^{n \times k}$. They satisfy $\boldsymbol{c} = \boldsymbol{a}\mathbf{W}$. The gradient $\nabla_{\boldsymbol{c}} l \in \mathbb{R}^{n \times k}$ is denoted by $\boldsymbol{g}$. The block-diagonal portion of the EF matrix corresponding to this layer (denoted as $\tilde{\mathbf{F}}_\mathbf{W}$) can be estimated using K-FAC approximation as follows [27]

$$
\tilde{\mathbf{F}}_\mathbf{W} = \mathbb{E}[(\boldsymbol{g}^\top\boldsymbol{g}) \otimes (\boldsymbol{a}^\top\boldsymbol{a})] \approx \mathbb{E}[\boldsymbol{g}^\top\boldsymbol{g}] \otimes \mathbb{E}[\boldsymbol{a}^\top\boldsymbol{a}]. \qquad (42)
$$

This is based on the gradient expression for weight matrix $\mathbf{W}$: $\nabla_\mathbf{W} l = \boldsymbol{g}^\top\boldsymbol{a}$. By rescaling this gradient with vector $\boldsymbol{s}_{\mathrm{iEF}}$, the expression for $\tilde{\mathbf{F}}_\mathbf{W}$ becomes:

$$
\tilde{\mathbf{F}}_\mathbf{W}^\star = \mathbb{E}[\boldsymbol{g}^\top\mathrm{diag}(\boldsymbol{s}_{\mathrm{iEF}})^{-1}\boldsymbol{g}] \otimes \mathbb{E}[\boldsymbol{a}^\top\boldsymbol{a}], \qquad (43)
$$

where the $\boldsymbol{s}_{\mathrm{iEF}}$ vector is easy to compute in Pytorch [32] with negligible extra cost (as shown in Appendix D.1), and such diagonal rescaling is straightforward to implement. In conclusion, the idea of iEF can be easily integrated into existing approximate empirical NGD optimisers, and it is interesting to observe the improvements due to such integration.

Preliminary evaluation of the approximation quality to the exact NG update of the block-diagonal version of the iEF method is conducted using the evaluation framework proposed in Sec. 6 (following the style of experiment (E1)). The approximation quality to exact NG updates (relative to SGD updates) are reported for iEF, KFAC (block-diagonal version of SF method [12]), eKFAC (block-diagonal version of EF method [37]) and ieKFAC (block-diagonal version of iEF method), for different training stages of selected tasks in Fig. 5. All updates are generated using a near-zero damping.

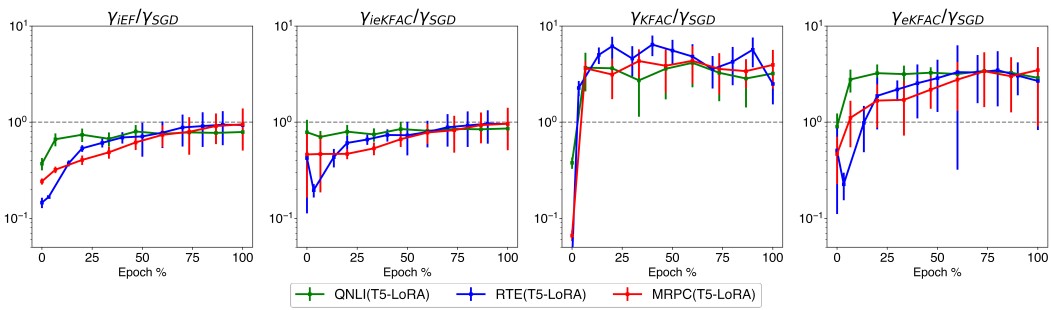

Figure 5: Approximation quality (relative to SGD) of "un-damped" iEF, ieKFAC, KFAC and eKFAC for 3 selected PEFT tasks (QNLI+LoRA, RTE+LoRA, MRPC+LoRA) across training stages. The style of the visualisation follows that for the first 3 plots of Fig. 2. This evaluation shows that, ieKFAC update has a similar approximation quality to the exact iEF method, and a much better approximation quality than both KFAC and eKFAC in most training stages. This demonstrates the effectiveness of using ieKFAC to approximate iEF and its potential of further improving the approximation quality of existing KFAC-based methods.

### D.2.3 Improving Empirical WoodFisher Model Compression

The EF matrix is used in the WoodFisher model compression algorithm [46] as an approximation to the Hessian matrix. It is therefore natural to consider using the iEF matrix in place of the EF matrix to improve the approximation quality. The WoodFisher algorithm relies on a recursion-base formulation of the EF matrix as follows

$$\tilde{\mathbf{F}}_{n+1} = \tilde{\mathbf{F}}_n + \frac{1}{N}\nabla_{\boldsymbol{\theta}} l_n \nabla_{\boldsymbol{\theta}} l_n^\top. \tag{44}$$

This can be easily switched to the iEF matrix using Eqn. 41 as follows

$$\mathbf{F}^\star_{n+1} = \mathbf{F}^\star_n + \frac{1}{N}(\|\nabla_{\boldsymbol{z}_n} l_n\|_2^{-1}\nabla_{\boldsymbol{\theta}} l_n)(\|\nabla_{\boldsymbol{z}_n} l_n\|_2^{-1}\nabla_{\boldsymbol{\theta}} l_n)^\top, \tag{45}$$

## E Monte-Carlo Sampled Fisher Matrix

The SF method used in the experimental setups (see Sec. 7) is introduced and analysed in details in this section. A commonly used method to estimate the exact Fisher matrix without bias is through Monte-Carlo sampling, which is used in the original K-FAC method [27]. This method is referred to as sampled Fisher in this paper. Particularly, when only one Monte-Carlo sample is generated for each training sample, the corresponding method is termed SF for brevity. Recall the definition of the Fisher matrix in Eqn. (2), it can be rewritten using expectations as follows

$$\mathbf{F} = \sum_n \mathbb{E}_{c \sim p_{\boldsymbol{\theta}}(y|\boldsymbol{x}_n)}[\nabla_{\boldsymbol{\theta}} \log p_n(c)\nabla_{\boldsymbol{\theta}} \log p_n(c)^\top]. \tag{46}$$

This means, by generating enough labels (with Monte-Carlo sampling) from the output distribution $p_{\boldsymbol{\theta}}(y|x_n)$, and computing the gradient *w.r.t.* these labels $\nabla_{\boldsymbol{\theta}} \log p_n(c)$, the Fisher matrix can be estimated without bias.

Assume $K$ labels are generated for each training sample, the exact expression for sampled Fisher with $K$ samples (denoted as $\hat{\mathbf{F}}(K)$) is as follows

$$\hat{\mathbf{F}}(K) = \frac{1}{K}\sum_{n=1}^{N}\sum_{k=1}^{K}[\nabla_{\boldsymbol{\theta}} \log p_n(\hat{y}_n^{(k)})\nabla_{\boldsymbol{\theta}} \log p_n(\hat{y}_n^{(k)})^\top] = \frac{1}{K}\hat{\mathbf{J}}(K)^\top\hat{\mathbf{J}}(K), \tag{47}$$

where $\hat{y}_n^{(k)} \sim p_{\boldsymbol{\theta}}(y|\boldsymbol{x}_n)$ is the $k$-th generated label for the $n$-th sample, Jacobian $\hat{\mathbf{J}}(K) \in \mathbb{R}^{(NK)\times P}$ denotes the stacked sampled gradients $[\nabla_{\boldsymbol{\theta}} \log p_1(\hat{y}_1^{(1)}), \ldots, \nabla_{\boldsymbol{\theta}} \log p_n(\hat{y}_n^{(k)}), \ldots, \nabla_{\boldsymbol{\theta}} \log p_N(\hat{y}_N^{(K)})]^\top$.

$\hat{\mathbf{F}}(1)$, *i.e.* sampled Fisher with 1 sampling for each training sample, is used as a baseline approximate NGD method in Sec. 7 (termed as SF in this paper) and is compared against EF and iEF. This has two reasons:

1. For $\hat{\mathbf{F}}(K)$, $K = 1$ is more commonly chosen in practice [13, 27] than $K > 1$. This is mainly because $\hat{\mathbf{F}}(K)$ requires $K$ additional back-propagations through the target batch, which becomes very expensive for a large $K$.

2. $\hat{\mathbf{F}}(1)$ has a maximum rank of $N$, which is the same rank as the EF matrix ($\tilde{\mathbf{F}}$) and the iEF matrix ($\tilde{\mathbf{F}}^\star$).

Note that even for $\hat{\mathbf{F}}(1)$, as compared to EF and iEF methods, requires an additional back-propagation through target batches. This makes $\hat{\mathbf{F}}(1)$ hard to implement in practice, and becomes nearly twice as expensive as EF/iEF. It is the leading reason that EF is commonly used in favour of $\hat{\mathbf{F}}(1)$ in practice [33, 34, 49, 31].

**Exact Pre-conditioning with $\hat{\mathbf{F}}(1)$**   To properly evaluate the pre-conditioner $\hat{\mathbf{F}}(1)$ either through optimisation or our evaluation framework (see Sec. 6), it is necessary to exactly compute the pre-conditioned gradient. The Sherman-Morrison-Woodbury (SMW) identity [40] can be used to achieve this, which states that $(\mathbf{U}^\top\mathbf{U} + \lambda\mathbf{I})^{-1} = \frac{1}{\lambda}[\mathbf{I} - \mathbf{U}^\top(\mathbf{U}\mathbf{U}^\top + \lambda\mathbf{I})^{-1}\mathbf{U}]$.

The update pre-conditioned by $\hat{\mathbf{F}}(1)$ (denoted as $\Delta\boldsymbol{\theta}_{\text{SF}}$) takes the following form

$$\Delta\boldsymbol{\theta}_{\text{SF}} = (\hat{\mathbf{F}}(1) + \lambda\mathbf{I})^{-1}\nabla_{\boldsymbol{\theta}}\mathcal{L}(\boldsymbol{\theta}). \tag{48}$$

By plugging in $\hat{\mathbf{F}}(1) = \hat{\mathbf{J}}(1)^\top\hat{\mathbf{J}}(1)$ and using the SMW identity, the update can then be computed as follows

$$\Delta\boldsymbol{\theta}_{\text{SF}} = \frac{1}{\lambda}[\mathbf{I} - \hat{\mathbf{J}}(1)^\top(\hat{\mathbf{J}}(1)\hat{\mathbf{J}}(1)^\top + \lambda\mathbf{I})^{-1}\hat{\mathbf{J}}(1)]\nabla_{\boldsymbol{\theta}}\mathcal{L}(\boldsymbol{\theta}), \tag{49}$$

which is easy to compute once the Jacobian ($\hat{\mathbf{J}}(1)$ for sampled gradients is collected (using the per-sample gradient collection method described in Appendix D.1). Similar to EF and iEF, an exact stochastic optimiser can also be constructed for SF, as is described in Algorithm 3. Note that as compared to Algorithm 2,1, SF requires an additional back-propagation.

---

**Algorithm 3** Stochastic Optimisation with Exact SF

---

**Require:** All $N$ training samples, initial model parameters $\boldsymbol{\theta}(0)$
**for** $t = 0$ **to** $T - 1$ **do**
    Sample Mini-batch $\mathcal{M}(t)$ with size $M$
    Perform forward pass on batch $\mathcal{M}(t)$ to obtain output distribution $\boldsymbol{p}(t)$
    Perform back-propagation to obtain accumulated loss $\nabla_{\boldsymbol{\theta}(t)}\mathcal{L}(\boldsymbol{\theta}(t))$
    Sample output labels $\hat{\boldsymbol{y}}(t) \sim \boldsymbol{p}(t)$
    Compute cross-entropy loss between $\boldsymbol{p}(t)$ and sampled label $\hat{\boldsymbol{y}}(t)$ to obtain the pseudo per-sample loss vector $\hat{l}(t)$
    Perform **additional back-propagation** to obtain Jacobian $\mathbf{A}(t) = \nabla_{\boldsymbol{\theta}(t)}\hat{l}(t)$
    Compute SF update $\Delta\boldsymbol{\theta}_{\text{SF}}(t) = \frac{1}{\lambda}[\mathbf{I} - \mathbf{A}(t)^\top(\mathbf{A}(t)\mathbf{A}(t)^\top + \lambda\mathbf{I})^{-1}\mathbf{A}(t)]\nabla_{\boldsymbol{\theta}(t)}\mathcal{L}(\boldsymbol{\theta}(t))$: Eqn. (49)

    Update model $\boldsymbol{\theta}(t + 1) = \boldsymbol{\theta}(t) - \eta\Delta\boldsymbol{\theta}_{\text{SF}}(t)$
**end for**

---

**Impact of Inversely-Scaled Projection on SF Update**   Although the same analysis in Sec. 4 cannot be applied to SF due to the different formulation of the update, the *inversely-scaled projection* issue is still expected to impact SF. As the model gets better trained, it becomes likely for the sampled Fisher matrix $\hat{\mathbf{F}}(1)$ of SF to sample empirical per-sample gradients. Pre-conditioning with the inverse of this matrix would then cause the inverse-scaling issue just like EF. Consequently, a poorer approximation quality of SF to the exact NG update is still probable.

## F   Empirical Evaluation Framework for Approximate NGD Methods

### F.1   Implementation Details

As introduced in Sec. 6, the proposed empirical evaluation framework is built around the proposed indicator $\gamma(\cdot)$. Consider a given model checkpoint $\boldsymbol{\theta}(t)$, $M$ pairs of training samples $(\boldsymbol{x}_n, y_n)_{n=1}^M$

from a target dataset and $K$ approximate NGD methods $\boldsymbol{g}^{(k)}(\cdot)$ of interest. Each update generation method can generate an update $\Delta\boldsymbol{\theta}(t)^{(k)}$ for the provided samples and a given model checkpoint following $\Delta\boldsymbol{\theta}(t)^{(k)} = \boldsymbol{g}^{(k)}(\boldsymbol{\theta}(t), (\boldsymbol{x}_n, y_n)_{n=1}^M)$. The framework is implemented such that $K$ indicators $\gamma(\Delta\boldsymbol{\theta}(t)^{(k)})$ are computed, which allows for the quantitative comparison of these update generation methods *w.r.t.* their approximation quality to the exact NGD method. The detailed evaluation process is described in Algorithm 4. In practice, for the $k$-th update generation method, multiple batches are used to compute a series of $\gamma(\Delta\boldsymbol{\theta}^{(k)})$, and their average is used to provide a more accurate indicator value. Note that this evaluation process can be repeated for model checkpoints at different training stages $\boldsymbol{\theta}_{t'}$ to provide a more thorough comparison.

It is worth mentioning that the ratio $\gamma(\Delta\boldsymbol{\theta}(t)^{(k)})/\gamma(\nabla_{\boldsymbol{\theta}(t)}\mathcal{L}(\boldsymbol{\theta}(t)))$ is an important quantity. This ratio describes the relative approximation quality of the target update generation method *w.r.t.* standard SGD method, and should be smaller than 1 for a good approximate NGD method. If this ratio is consistently larger than 1, then the target approximate NGD method $\boldsymbol{g}^{(k)}(\cdot)$ has a worse approximation quality than the SGD method and is likely to perform poorly in practice.

The four update methods investigated in the main paper (see Sec. 7) are SGD, EF, iEF and SF (sampled Fisher with 1 Monte-Carlo sample for each training sample). The SGD update is simply the gradient $\nabla_{\boldsymbol{\theta}(t)}\mathcal{L}(\boldsymbol{\theta}(t))$; the iEF update is computed according to Eqn. (8); the EF update is computed according to Eqn. (5); the SF update is computed according to Eqn. (49).

Finally, the computationally most expensive operation when computing indicators is finding the matrix-vector product between the exact Fisher and a given update direction, *i.e.* $\mathbf{F}\Delta\boldsymbol{\theta}$. A detailed discussion is provided on its implementation and algorithm complexity in the following section.

---

**Algorithm 4** Empirical evaluation framework for approximate NGD Methods

---

**Input:** A batch of $M$ training data pairs $(\boldsymbol{x}_n, y_n)_{n=1}^M$, model checkpoint $\boldsymbol{\theta}(t)$, loss function $\mathcal{L}(\boldsymbol{\theta}(t))$, $K$ update generation methods $\boldsymbol{g}^{(k)}(\boldsymbol{\theta}(t), (\boldsymbol{x}_n, y_n)_{n=1}^M), k \in \{1, \cdots, K\}$.
**Execute:**
Compute accumulated gradient $\nabla_{\boldsymbol{\theta}}\mathcal{L}(\boldsymbol{\theta}(t))$ on $M$ training samples
**for** $k = 1$ **to** $K$ **do**
    Compute update $\Delta\boldsymbol{\theta}(t)^{(k)} = \boldsymbol{g}^{(k)}(\boldsymbol{\theta}(t), (\boldsymbol{x}_n, y_n)_{n=1}^M)$
    Compute indicator $\gamma(\Delta\boldsymbol{\theta}(t)^{(k)}) = \frac{[(\Delta\boldsymbol{\theta}(t)^{(k)})^\top \mathbf{F}\Delta\boldsymbol{\theta}(t)^{(k)}]^{\frac{1}{2}}}{|(\Delta\boldsymbol{\theta}(t)^{(k)})^\top \nabla_{\boldsymbol{\theta}(t)}\mathcal{L}(\boldsymbol{\theta}(t))|}$
**end for**
**Output:** Return $K$ indicators $\left[\gamma(\Delta\boldsymbol{\theta}(t)^{(k)})\right]_{k=1}^K$. Updates with smaller indicator values are considered better approximate NG updates.

---

### F.1.1 Matrix-Vector Product and Algorithm Complexity

The most important operation in Algorithm 4 when computing the proposed indicator is the computation of the matrix-vector product $\mathbf{F}(\cdot)$. It is the most expensive operation that dominates both the time and space complexity of this algorithm. Fortunately, this can be efficiently computed in a modern auto-grad framework using the double differentiation trick [44]. The key to computing the matrix-vector product with the exact Fisher matrix is identifying the equivalence of the Fisher matrix and the Generalised Gauss-Newton matrix (GGN matrix, denoted as $\mathbf{G}$). It can be shown that for the target model which uses softmax activation and cross-entropy loss function, the following equality holds

$$\mathbf{F} = \sum_n \nabla_{\boldsymbol{\theta}}\boldsymbol{z}_n^\top (\nabla_{\boldsymbol{z}_n}^2 l_n)\nabla_{\boldsymbol{\theta}}\boldsymbol{z}_n = \mathbf{G}. \tag{50}$$

Therefore, the matrix-vector product can be broken down into the computation of three separate matrix-vector products. The double differentiation trick is useful when computing the first two matrix-vector products: $\nabla_{\boldsymbol{\theta}}\boldsymbol{z}_n(\cdot), \nabla_{\boldsymbol{z}_n}^2 l_n(\cdot)$. The final matrix-vector product $\nabla_{\boldsymbol{\theta}}\boldsymbol{z}_n^\top(\cdot)$ can be computed using the standard back-propagation pipeline. Please refer to [44] for implementation details.

Overall, with the double differentiation trick, the time and memory complexity of the matrix-vector product with the Fisher matrix is comparable to standard training with batch size $M$. In our implementation in Pytorch, the matrix-vector product requires 2 forward passes and 2 backward passes

through the model on the $M$ samples. This makes our proposed empirical evaluation framework efficient in practice.

Finally, for the reader's information, if we adopt the method used in [25], it is possible to further reduce the cost of the matrix-vector product down to only 1 additional forward pass and backward pass. However, this may not be a preferable choice because it requires customised forward-propagation code, making our evaluation framework less generally applicable.

### F.1.2 Comparison to Traditional Evaluation Methods

As is summarised in Sec. 2, traditional evaluation of the approximation quality to the NGD method requires the explicit storage and inversion of the exact Fisher matrix. This is usually done using the definition in Eqn. (2). For a general model, computing the Fisher matrix in Eqn. (2) requires $C$ separate backwards and forward passes through the model ($C$ being the output category number). This can range from 10 (e.g. MNIST[21]) to >10,000 (for our LLM finetune setup [7]). From a time complexity perspective, the traditional method can be arbitrarily more expensive than our evaluation framework depending on the output category size $C$. From a memory complexity perspective, the storage of the Fisher matrix of $\mathbb{R}^{P \times P}$ is infeasible for a large-scale model. Both of these concerns limit the traditional evaluation of approximate NGD methods in large practical setups. On the contrary, our proposed evaluation framework resolves these limitations of the traditional evaluation method, meanwhile having a strong theoretical motivation. This greatly facilitates the evaluation of approximate NGD methods in large-scale deep-learning setups for methods not limited to EF, iEF and SF.

### F.2 Hessian-free Implicitly Minimises $\gamma(\cdot)$ Indicator

In Sec. 6.2, it is stated that the indicator $\gamma(\cdot)$ is implicitly minimised in the linear CG algorithm to iteratively approximate the exact NG update in Hessian-free method [25]. In this section, this statement is formally introduced and justified.

The Hessian-free method is an important approximate NGD method. Unlike most other approximate NGD methods, which approximate the Fisher matrix $\mathbf{F}$ directly, the Hessian-free method does not explicitly express the Fisher matrix. Instead, it uses an iterative method to directly solve for $\Delta\boldsymbol{\theta}_{\mathrm{HF}}$ in the following equation

$$\mathbf{F}\Delta\boldsymbol{\theta}_{\mathrm{HF}} = \nabla_{\boldsymbol{\theta}}\mathcal{L}(\boldsymbol{\theta}). \tag{51}$$

The iterative method used in Hessian-free is the linear CG algorithm [45], which is a classical method that iteratively solves for $\Delta\boldsymbol{\theta}_{\mathrm{HF}}$ in Eqn. (51) with only the access to the matrix-vector product function $\mathbf{F}(\cdot)$ (introduced in Sec. 6). Since the Fisher is not required to be explicitly stored, the method enjoys a memory complexity of $O(P)$ where $P$ is the number of trainable parameters in a target model.

The linear CG algorithm (without pre-conditioning) used in HF is shown in Algorithm 5.

It is known that the CG algorithm is an efficient solver for Eqn. (51) and is usually regarded as a locally optimal minimiser (each step uses an exact line search) for the equivalent pseudo loss function [25, 45]

$$L'_{\mathrm{CG}}(\Delta\boldsymbol{\theta}_{\mathrm{HF}}) = -\Delta\boldsymbol{\theta}_{\mathrm{HF}}^{\top}\nabla_{\boldsymbol{\theta}}\mathcal{L}(\boldsymbol{\theta}) + \frac{1}{2}\Delta\boldsymbol{\theta}_{\mathrm{HF}}^{\top}\mathbf{F}\Delta\boldsymbol{\theta}_{\mathrm{HF}}. \tag{52}$$

However, it is also possible to interpret this method as a locally optimal minimiser [18] for the generalised Rayleigh Quotient of positive semi-definite matrices $\mathbf{F}$ and $\nabla_{\boldsymbol{\theta}}\mathcal{L}(\boldsymbol{\theta})\nabla_{\boldsymbol{\theta}}\mathcal{L}(\boldsymbol{\theta})^{\top}$

$$\gamma(\Delta\boldsymbol{\theta}_{\mathrm{HF}})^2 = \frac{(\Delta\boldsymbol{\theta}_{\mathrm{HF}}^{\top}\mathbf{F}\Delta\boldsymbol{\theta}_{\mathrm{HF}})}{\Delta\boldsymbol{\theta}_{\mathrm{HF}}^{\top}[\nabla_{\boldsymbol{\theta}}\mathcal{L}(\boldsymbol{\theta})\nabla_{\boldsymbol{\theta}}\mathcal{L}(\boldsymbol{\theta})^{\top}]\Delta\boldsymbol{\theta}_{\mathrm{HF}}} \tag{53}$$

where $\gamma(\Delta\boldsymbol{\theta}_{\mathrm{HF}})$ happens to be the proposed indicator of this paper. The proof of this statement is provided in the following section.

### F.2.1 Proof that Linear CG is a Locally Optimal Minimiser for Indicator $\gamma(\cdot)$

Note that in every iteration in Algorithm 5, an update $\alpha_m \boldsymbol{v}_m$ is accumulated to the final solution $\boldsymbol{x}_{M_{\mathrm{CG}}}$. This update $\alpha_m \boldsymbol{v}_m$ can be shown to achieve the local maximum reduction in $\gamma(\boldsymbol{x})^2$ in every

**Algorithm 5** The Linear CG Algorithm in Hessian-free Method

---

**Input:** The maximum CG execution iterations $M_{\text{CG}}$
The gradient vector $\boldsymbol{b} = \nabla_{\boldsymbol{\theta}} \mathcal{L}(\boldsymbol{\theta})$
The matrix-vector product function for the Fisher $\mathbf{F}$
**Execute:**
Set $\boldsymbol{r}_0 \leftarrow \boldsymbol{b}$
Set $\boldsymbol{v}_0 \leftarrow \boldsymbol{r}_0$
Set $m \leftarrow 0$
**while** $m < M_{\text{CG}}$ **do**
    Compute $\|\boldsymbol{r}_m\|^2 = \boldsymbol{r}_m^\top \boldsymbol{r}_m$
    Set $\alpha_m \leftarrow \|\boldsymbol{r}_m\|^2 / (\boldsymbol{v}_m^\top \mathbf{F} \boldsymbol{v}_m)$
    Update $\boldsymbol{x}_{m+1} \leftarrow \boldsymbol{x}_m + \alpha_m \boldsymbol{v}_m$
    Update $\boldsymbol{r}_{m+1} \leftarrow \boldsymbol{r}_m - \alpha_m \mathbf{F} \boldsymbol{v}_m$
    Compute $\|\boldsymbol{r}_{m+1}\|^2 = \boldsymbol{r}_{m+1}^\top \boldsymbol{r}_{m+1}$
    Set $\beta_{m+1} \leftarrow \|\boldsymbol{r}_{m+1}\|^2 / \|\boldsymbol{r}_m\|^2$
    Update $\boldsymbol{v}_{m+1} \leftarrow \boldsymbol{r}_{m+1} + \beta_{m+1} \boldsymbol{v}_m$
    Update $m \leftarrow m + 1$
**end while**
**Output:** Return $\boldsymbol{x}_{M_{\text{CG}}}$ as an approximate solution for $\Delta \boldsymbol{\theta}_{\text{EF}}$

---

iteration. Formally, it is to be shown that, for the current partial solution $\boldsymbol{x}_m$ and the given search direction $\boldsymbol{v}_m$, the scaling factor $\alpha_m$ achieves the maximum reduction in $\gamma(\boldsymbol{x}_{M_{\text{CG}}})^2$:

$$\alpha_m = \arg \min_{\alpha_m} \gamma(\boldsymbol{x}_{M_{\text{CG}}})^2. \tag{54}$$

Now, assume the actual minimiser $\alpha'_m$ is different from the $\alpha_m$ in the CG iterations. The true minimiser $\alpha'_m$ can be acquired through a Ritz-Rayleigh analysis [18]. By setting $\mathbf{B}' = \nabla_{\boldsymbol{\theta}} \mathcal{L}(\boldsymbol{\theta}) \nabla_{\boldsymbol{\theta}} \mathcal{L}(\boldsymbol{\theta})^\top$, and plugging in the true minimiser $\alpha'_m$, Eqn. (54) becomes:

$$\alpha'_m = \arg \min_{\alpha'_m} \frac{[(\boldsymbol{x}_m + \alpha'_m \boldsymbol{v})^\top \mathbf{F} (\boldsymbol{x}_m + \alpha'_m \boldsymbol{v})]}{[(\boldsymbol{x}_m + \alpha'_m \boldsymbol{v})^\top \mathbf{B}' (\boldsymbol{x}_m + \alpha'_m \boldsymbol{v})]}. \tag{55}$$

The Ritz-Rayleigh analysis then gives the optimal scaling:

$$\alpha'_m = \frac{\boldsymbol{b}^\top \boldsymbol{v}_m}{\boldsymbol{v}_m^\top \mathbf{F} \boldsymbol{v}_m}. \tag{56}$$

Recall that $\alpha_m = \frac{\boldsymbol{r}_m^\top \boldsymbol{r}_m}{\boldsymbol{v}_m^\top \mathbf{F} \boldsymbol{v}_m}$, in order to prove $\alpha_m = \alpha'_m$, the following need be proved $\boldsymbol{b}^\top \boldsymbol{v}_m = \boldsymbol{r}_m^\top \boldsymbol{r}_m$. This can be done through recursion.

For $m = 0$, it is obvious that $\boldsymbol{b}^\top \boldsymbol{v}_0 = \boldsymbol{b}^\top \boldsymbol{b} = \boldsymbol{r}_0^\top \boldsymbol{r}_0$.

For $m > 1$, it is known that $\boldsymbol{b}^\top \boldsymbol{v}_{m-1} = \boldsymbol{r}_{m-1}^\top \boldsymbol{r}_{m-1}$ is true, then:

$$\begin{aligned}
\boldsymbol{b}^\top \boldsymbol{v}_m &= \boldsymbol{b}^\top (\boldsymbol{r}_m + \beta_m \boldsymbol{v}_{m-1}) \\
&= \boldsymbol{b}^\top \boldsymbol{r}_m + \frac{\boldsymbol{r}_m^\top \boldsymbol{r}_m^\top}{\boldsymbol{r}_{m-1}^\top \boldsymbol{r}_{m-1}} \boldsymbol{b}^\top \boldsymbol{v}_{m-1} \\
&= \boldsymbol{r}_0^\top \boldsymbol{r}_m + \frac{\boldsymbol{r}_m^\top \boldsymbol{r}_m^\top}{\boldsymbol{r}_{m-1}^\top \boldsymbol{r}_{m-1}} \boldsymbol{r}_{m-1}^\top \boldsymbol{r}_{m-1} \\
&= 0 + \boldsymbol{r}_m^\top \boldsymbol{r}_m \quad (\because \forall i \neq j, \boldsymbol{r}_i^\top \boldsymbol{r}_j = 0) \\
&= \boldsymbol{r}_m^\top \boldsymbol{r}_m
\end{aligned} \tag{57}$$

In conclusion, $\alpha_m$ is indeed the true minimiser for $\frac{[(\boldsymbol{x}_m + \alpha'_m \boldsymbol{v})^\top \mathbf{F} (\boldsymbol{x}_m + \alpha'_m \boldsymbol{v})]}{[(\boldsymbol{x}_m + \alpha'_m \boldsymbol{v})^\top \mathbf{B}' (\boldsymbol{x}_m + \alpha'_m \boldsymbol{v})]}$ in every CG iterations, making CG a locally optimal optimiser for $\gamma(\boldsymbol{x})^2$.

It is known that Hessian-free iteratively approaches the exact NG update [25], and a larger iteration number $M_{\text{CG}}$ leads to a better approximation. Now that it has been shown that the indicator $\gamma(\boldsymbol{x})$ decreases (optimally) for every iteration of Hessian-free, it verifies that indicator $\gamma(\boldsymbol{x})$ directly describes the approximation level of the intermediate solution $\boldsymbol{x}_m$ to the exact NG update.

### F.3 $\gamma(\cdot)$ Quantifies Loss Reduction Effectiveness under the Local Quadratic Approximation

The indicator $\gamma(\Delta\boldsymbol{\theta})$ can also describe the loss-reduction effectiveness of a target update direction under the Local Quadratic Approximation. The loss change induced by a target update can be estimated using a Taylor expansion up to the second order as follows:

$$\mathcal{L}(\boldsymbol{\theta}(t+1)) \approx \mathcal{L}(\boldsymbol{\theta}(t)) + \Delta\boldsymbol{\theta}(t)^\top \nabla_{\boldsymbol{\theta}}\mathcal{L}(\boldsymbol{\theta}(t)) + \frac{1}{2}\Delta\boldsymbol{\theta}(t)^\top \mathbf{B}_t \Delta\boldsymbol{\theta}(t) \tag{58}$$

where $\mathbf{B} \in \mathbb{R}^{P \times P}$ is the choice of curvature matrix. Such an approximation to the loss function is regarded as a Local Quadratic Approximation [55].

Consider a fixed-direction target update $\Delta\boldsymbol{\theta}$ with an unknown step size $\eta$, the maximum achievable loss reduction based on Local Quadratic Approximation can be obtained as follows

$$-\frac{1}{2\gamma(\Delta\boldsymbol{\theta})^2} = \min_{\eta} \eta\Delta\boldsymbol{\theta}^\top \nabla_{\boldsymbol{\theta}}\mathcal{L}(\boldsymbol{\theta}) + \frac{\eta^2}{2}\Delta\boldsymbol{\theta}^\top \mathbf{F}\Delta\boldsymbol{\theta}, \tag{59}$$

where $\mathbf{F}$ is used in place of the curvature matrix $\mathbf{B}$. This implies that the maximum loss reduction along update direction $\Delta\boldsymbol{\theta}$ is $-1/2\gamma(\Delta\boldsymbol{\theta})^2$, which is inversely proportional to the square of the proposed indicator. For an update with a smaller $\gamma(\Delta\boldsymbol{\theta})$, it is expected to induce a larger loss reduction, with the exact NG update achieving the maximum loss reduction. Consequently, the proposed indicator has a strong correlation to practical convergence ability of the target update generation method.

## G  Limitations

**Focus on Theoretical Approximation for NGD Method**   The focus of the paper is on the theoretical approximation methods for NGD, including the exact EF method, the proposed iEF method, and also the SF method. Practical approximate (empirical) NGD optimisers are not the main focus of this paper (*e.g.* K-FAC [27], EK-FAC [11], TNT [41] *etc.*), and no experiments are conducted for them. Despite the importance of these practical optimisers in the scene of NGD-based optimisation, they can all be considered a further (structure-aware) approximation to the EF or SF methods. However, it is an important future work to apply iEF to improve existing approximate NGD optimisers (an example of using iEF to improve empirical K-FAC is provided in Appendix D.2.2).

**Limitations of Exact Update Generation**   The updates generated by EF, iEF and SF methods (defined in Eqn. (5), (8), (49) respectively) investigated in this paper are "*exact*", in order to provide a more theoretically rigorous comparison and analysis. In our experiments, they are generated based only on the current provided batch $\mathcal{M}(t)$ as is described in Algorithms 2, 1, 3 respectively. The implementation of exact EF, iEF and SF updates requires storage of the $M$ per-sample gradients, which is memory-demanding in practice. This limits the scope of application of the exact iEF (and EF/SF) optimisers to setups where trainable parameter size is small, such as PEFT for pre-trained models. However, given the rise of PEFT setups, and consider the competitive optimisation performance of the exact iEF optimiser, such limitations may be out-weighted by the gain in practice. Additionally, the exact update formulation of iEF means momentum and weight decay cannot be directly applied with the resultant optimiser. This affects the optimisation performance on certain setups, and further work is required to integrate these key techniques with exact iEF optimiser. However, none of these limitations would affect the future work where iEF is incorporated into practical approximate NGD optimisers, where memory constraints and momentum/weight decay integration is already resolved in these practical NGD optimisers.

# H  Detailed Experiment Information

## H.1  Overall Experimental Setup

**Textual Classification of GLUE**    We have investigated 7 selected GLUE benchmark tasks [50] including CoLA, SST-2, MRPC, QQP, MNLI, QNLI, and RTE, which together cover a range of NLP classification tasks such as sentiment classification, semantic equivalence checking, grammar checking, entailment prediction etc. The same selection of GLUE tasks is used in [7]. One key aspect of GLUE tasks is that they do not have test labels, and for test evaluations, they have to be submitted to their official website (with a maximum allowed frequency of 3 submits per day). The textual label tokens are to be directly predicted by the target model. Also, an instruction is prepended to the input of each training sample. See details in Table. 2. For each task, a pre-trained T5-base model [39] is trained with two parameter-efficient finetuning methods: Prompt Tuning [22] and LoRA [16].

Table 2: Training details for the 7 GLUE tasks. The input format column describes how the inputs are structured before being sent to the model. {S1}, {S2} represents the default input sentence(s) of the corresponding tasks, some tasks have only one input sentence and some have two. Labels are the tokens to be predicted by the model for each task.

| Task | Input Format | Labels |
|------|--------------|--------|
| **CoLA** | Classify if the following sentence's grammar is acceptable or unacceptable: {S1} | *acceptable*, *unacceptable* |
| **SST-2** | Classify if the following sentence's sentiment is positive or negative: {S1} | *positive*, *negative* |
| **QQP** | Classify if the following Q1 and Q2 are semantically equivalent, answer yes or no: Q1: {S1} Q2: {S2} | *yes*, *no* |
| **MRPC** | Classify if the following S1 and S2 are semantically equivalent, answer yes or no: S1: {S1} S2: {S2} | *yes*, *no* |
| **MNLI** | Predict whether the premise entails the hypothesis, contradicts the hypothesis,or neither, answer yes, no or maybe: premise: {S1} hypothesis: {S2} | *yes*, *no*, *maybe* |
| **QNLI** | Determine whether the context sentence S contains the answer to the question Q, answer yes or no: Q: {S1} S: {S2} | *yes*, *no* |
| **RTE** | Classify if S1 entailment S2 or not, answer yes or no: S1: {S1} S2: {S2} | *yes*, *no* |

**Computer Vision Classification with CIFAR100**    The well-known CIFAR100 dataset [19] is used to finetune the pretrained ViT model [8]. The model is finetuned with LoRA [16] following the setup in [15].

## H.2  Optimisation Experimental Setup

For Prompt Tuning, the Adafactor [23] optimiser is used as baseline (as is done in [23, 7]). For all other training setups, the AdamW [24] optimiser was used as the baseline.

Different tasks were trained with a different number of epochs. The validation accuracy (on the dev set of each task) of the best checkpoint is reported and sent for test evaluation (for GLUE, the checkpoints are submitted to GLUE website [50]). Details are shown in Table 3.

For all the GLUE tasks and for each finetuning method (Prompt Tuning or LoRA), a single set of hyper-parameters were searched for each optimiser and is used across runs on all 7 GLUE tasks. Three runs with different seeds were conducted to generate the error bar. For each run, the checkpoint with the highest validation accuracy was saved for later evaluation (to get test performance or to be evaluated in the proposed evaluation framework). All optimisers were trained on a batch size of 32.

**Prompt Tuning + T5-base on GLUE:** 20 prompts are used for Prompt Tuning. The trainable parameter size is $15,360$, taking up $0.0069\%$ of the total parameter size (222,918,912) of the T5-base model. Constant scheduling was used for Adafactor, SGD and iEF runs. For iEF, EF and SF, a damping of $\lambda = 1 \times 10^{-12}$ was used. For the Adafactor baseline optimiser, the hyper-parameter provided in [7] was used (which comes from [23]): weight-decay $1 \times 10^{-5}$, $\beta_2 = 0.8$, learning rate $\eta = 0.3$ and no parameter scaling. For the SGD method, the learning rate is $\eta = 100$, which was

Table 3: Optimisation details all involved tasks. Train epochs represent the number of epochs for which the model is trained. Evaluation frequency describes the number of update steps between each validation evaluation on the development set.

| Task | Train Epochs | Evaluation Frequency |
|---|---|---|
| **CoLA** | 20 | 100 |
| **SST-2** | 20 | 1000 |
| **QQP** | 5 | 1000 |
| **MRPC** | 30 | 50 |
| **MNLI** | 5 | 1000 |
| **QNLI** | 15 | 1000 |
| **RTE** | 40 | 30 |
| **CIFAR100** | 5 | 1000 |

searched from $\{0.1, 1, 10, 20, 50, 100\}$. For the iEF method, the learning rate was $\eta = 50$, which was searched from $\{1, 10, 50, 100\}$. For the EF method, a different scheduling of learning rate was used to guarantee convergence, due to the inverse scaling of EF updates. The chosen strategy was a linearly decaying normalised update, with the first update being normalised to 1 ($\{1 \times 10^{-3}, 5 \times 10^{-3}, 1 \times 10^{-2}, 1 \times 10^{-1}, 1, 10\}$) and linearly decaying to 0. The SF method was trained using the same method as EF with the same set of hyperparameters.

**LoRA + T5-base on GLUE:** The LoRA was set to have rank 8 with dropout 0.1 [15]. The trainable parameter size is $884,736$, taking up $0.40\%$ of the total parameter size ($222,918,912$) of the T5-base model. Constant scheduling was used for the AdamW, SGD and iEF runs. For iEF, EF and SF, a damping of $\lambda = 1 \times 10^{-7}$ was used (it is 5 orders of magnitude larger than that used in Prompt Tuning because the diagonal of the gradient covariance matrix has a 5 order of magnitude larger norm in LoRA than Prompt Tuning). For the AdamW baseline optimiser, the hyper-parameters are: weight-decay $1 \times 10^{-2}$, and the learning rate of $1 \times 10^{-3}$ was searched from $\{1 \times 10^{-3}, 5 \times 10^{-4}, 1 \times 10^{-4}\}$. For the SGD method, the learning rate was $\eta = 0.1$, which was searched from $\{0.1, 1, 10, 20, 50, 100\}$. For the iEF method, the learning rate was $\eta = 100$, which was searched from $\{1, 10, 50, 100\}$. For the EF method, a normalised update with linear scheduling was used similar to Prompt Tuning, and a starting learning rate of 0.01 was searched from $\{1 \times 10^{-3}, 5 \times 10^{-3}, 1 \times 10^{-2}, 1 \times 10^{-1}, 1\}$. The SF method was trained with the same strategy and hyperparameters.

**LoRA + ViT on CIFAR100** The setup of using LoRA to finetune CIFAR100 in [15] was used. The trainable parameter size is 313,344, taking up $0.36\%$ of the total parameter size (86,862,568) of the ViT model (vit-base-patch16-224). The same hyperparameters for LoRA + T5 was used for these experiments.

## H.3 Experiments Compute Resources

All the optimisation experiments and evaluation experiments are run on a cloud linux machine with 8 A100 GPUs with 80GB GRAM. For all the optimisation experiments, optimisation time with different optimisers are similar, apart from the SF optimiser where the additional back-propagation leads to an additional 60% runtime. For standard SGD/AdamW/Adafactor/EF/iEF optimisers, on average each complete run takes 10 hours. The slowest task (QQP + LoRA) takes 20 hours and the quickest task (RTE + Prompt Tuning) takes 0.3 hours. In total 420 GPU hours are run for all optimisation experiments. For evaluation runs, each evaluation is done on 100 batches of 160 training samples. For each checkpoint and a choice of damping, the evaluation takes on average 30 minutes. For all the evaluation runs (damping evaluated 5 tasks x 3 ckpts x 10 damping = 150 evaluations, standard evaluated 15 tasks x 7 ckpts = 115 evaluated, total 265 evaluation), 133 GPU hours are done. Considering hyper-parameter tuning, and various preliminary runs to make exact EF/iEF/SF optimisers and the evaluation framework to run properly, in total 2 times of additional compute time is required for all experiments.

## H.4 Licenses for All Used Assets

The license, citation and link to various asset used in this paper is provided in Table 4.

Table 4: The license, citation and link to various asset used in this paper.

| Asset Name | Citation | URL | License |
|---|---|---|---|
| T5-base | [39] | Model checkpoint | Apache-2.0 |
| ViT-B/16 | [8] | Model checkpoint | Apache-2.0 |
| GLUE | [50] | Website | Multiple |
| CIFAR10/100 | [19] | Website | Unknown |
| Pytorch | [32] | Website | Multiple |

## H.5 Further Evaluation Plots

Due to space limit and presentation concerns, the evaluation results presented in the main text in experiments (E1) and (E3) are only partial. In this section, additional evaluation results are presented.

### H.5.1 Approximation Quality across Tasks and Training Stages

The same indicator evaluation in Fig. 2 is done for all experimented tasks. A full view of the indicator plots are provided in Fig. 6 and 7. Similar trend can be found across tasks.

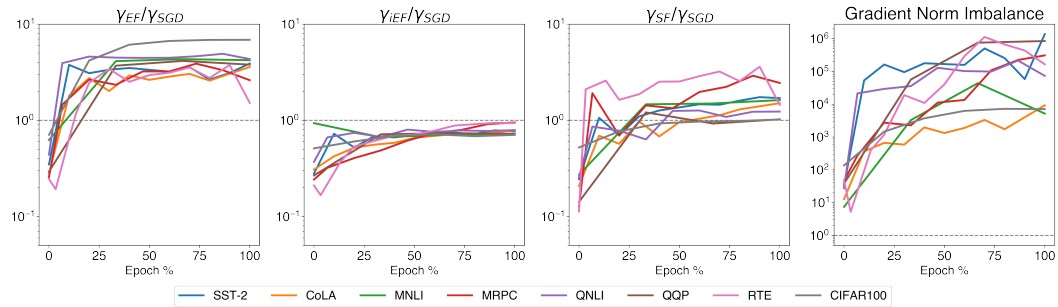

Figure 6: Four (log-scaled) ratios computed for checkpoints at various stages of training (sampled at the interval of one epoch) for all LoRA tasks, including T5 + 7 GLUE tasks and ViT + 1 CIFAR100. The figure is drawn in the same fashion as Fig. 2. Note that the error bar is not presented to improve presentation clarity.

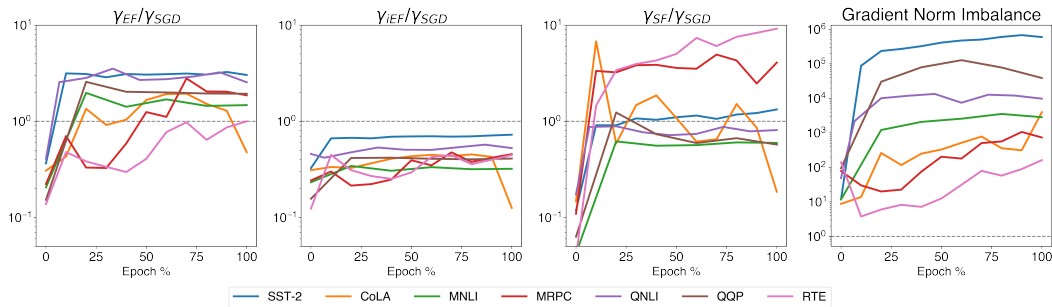

Figure 7: Four (log-scaled) ratios computed for checkpoints at various stages of training (sampled at the interval of one epoch) for 7 Prompt Tuning tasks for T5 + GLUE. The figure is drawn in the same fashion as Fig. 2. Note that the error bar is not presented to improve presentation clarity.

### H.5.2 Impact of Damping on Approximation Quality

The same damping analysis in Fig. 3 is applied to several other tasks (SST2+T5+Prompt Tuning, MRPC+T5+Prompt Tuning, CIFAR100+ViT+LoRA, RTE+T5+LoRA,as shown in Fig. 8, 9, 10, 11 respectively). Similar trend can be found across tasks.

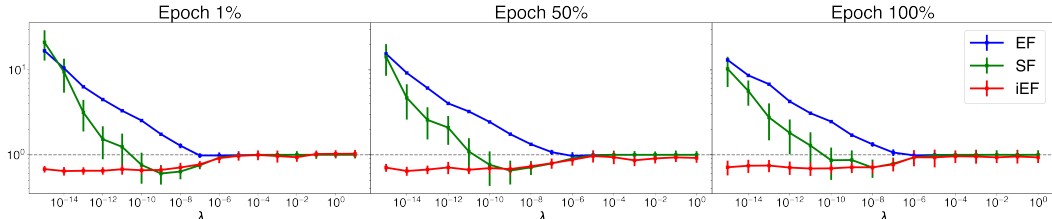

Figure 8: Approximation quality (relative to SGD) of EF, SF and iEF methods *w.r.t.* damping factor $\lambda$ at different training stages of task SST2+T5+Prompt Tuning. The figure is drawn in the same fashion as Fig. 3.

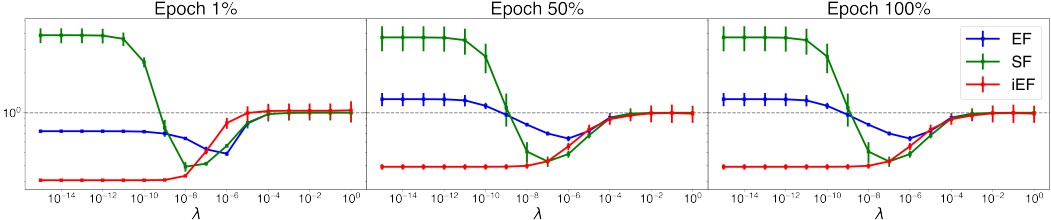

Figure 9: Approximation quality (relative to SGD) of EF, SF and iEF methods *w.r.t.* damping factor $\lambda$ at different training stages of task MRPC+T5+Prompt Tuning. The figure is drawn in the same fashion as Fig. 3.

## H.6 Full Optimisation Results

Due to space limit, only a partial test performance table (Table 1) is presented for the experiment (E2) in the main text. In this section, the full training, validation and test results for all combinations of structure, tasks and optimisers are reported for the reader's information.

Training curves and validation accuracy curves of different optimisers for the described 15 setups are presented in Fig. 12 and 13. The final training loss for every task, structure and optimiser combination is presented in Table 5. The validation performance for every task, structure and optimiser combination is presented in Table 6. The test performance for all tasks is presented in Table 7.

Table 5: Final train loss for all the task, structure and optimiser combinations. The average train loss of the final 100 steps of 3 random seed runs is reported, along with the standard deviation. IEF achieves the lowest final training loss for 12 out of 15 tasks.

| | CoLA | SST-2 | MRPC | QQP | MNLI | QNLI | RTE | CIFAR100 |
|---|---|---|---|---|---|---|---|---|
| | | | | Prompt Tuning | | | | |
| **Adafactor** | $0.07_{\pm 0.08}$ | $0.07_{\pm 0.02}$ | $0.16_{\pm 0.07}$ | $\mathbf{0.02}_{\pm 0.02}$ | $0.19_{\pm 0.02}$ | $0.10_{\pm 0.05}$ | $0.29_{\pm 0.03}$ | - |
| **SGD** | $0.33_{\pm 0.06}$ | $0.04_{\pm 0.02}$ | $0.36_{\pm 0.11}$ | $0.15_{\pm 0.13}$ | $0.31_{\pm 0.05}$ | $0.15_{\pm 0.06}$ | $0.35_{\pm 0.03}$ | - |
| **EF** | $2.28_{\pm 1.11}$ | $0.99_{\pm 0.35}$ | $2.80_{\pm 1.19}$ | $4.29_{\pm 1.04}$ | $6.11_{\pm 1.10}$ | $5.12_{\pm 0.88}$ | $1.79_{\pm 0.16}$ | - |
| **SF** | $0.16_{\pm 0.10}$ | $0.06_{\pm 0.11}$ | $0.43_{\pm 0.09}$ | $0.15_{\pm 0.02}$ | $0.28_{\pm 0.02}$ | $0.27_{\pm 0.14}$ | $0.36_{\pm 0.03}$ | - |
| **iEF** | $\mathbf{0.01}_{\pm 0.01}$ | $\mathbf{0.02}_{\pm 0.01}$ | $\mathbf{0.04}_{\pm 0.01}$ | $\mathbf{0.02}_{\pm 0.01}$ | $\mathbf{0.17}_{\pm 0.07}$ | $\mathbf{0.09}_{\pm 0.06}$ | $\mathbf{0.02}_{\pm 0.01}$ | - |
| | | | | LoRA | | | | |
| **AdamW** | $\mathbf{0.10}_{\pm 0.03}$ | $\mathbf{0.02}_{\pm 0.01}$ | $0.07_{\pm 0.06}$ | $\mathbf{0.07}_{\pm 0.01}$ | $0.22_{\pm 0.06}$ | $0.10_{\pm 0.03}$ | $\mathbf{0.02}_{\pm 0.03}$ | $0.60_{\pm 0.21}$ |
| **SGD** | $0.24_{\pm 0.02}$ | $0.03_{\pm 0.09}$ | $0.12_{\pm 0.01}$ | $0.10_{\pm 0.05}$ | $0.22_{\pm 0.12}$ | $0.11_{\pm 0.07}$ | $0.18_{\pm 0.07}$ | $0.94_{\pm 0.34}$ |
| **EF** | $1.92_{\pm 0.49}$ | $0.90_{\pm 0.03}$ | $2.541_{\pm 0.78}$ | $1.03_{\pm 0.51}$ | $3.42_{\pm 0.33}$ | $1.31_{\pm 0.89}$ | $3.08_{\pm 0.40}$ | $3.10_{\pm 0.11}$ |
| **SF** | $0.26_{\pm 0.03}$ | $0.14_{\pm 0.22}$ | $0.35_{\pm 0.11}$ | $0.20_{\pm 0.12}$ | $0.24_{\pm 0.09}$ | $0.30_{\pm 0.21}$ | $0.39_{\pm 0.02}$ | $0.63_{\pm 0.12}$ |
| **iEF** | $0.16_{\pm 0.04}$ | $\mathbf{0.02}_{\pm 0.02}$ | $\mathbf{0.06}_{\pm 0.04}$ | $\mathbf{0.07}_{\pm 0.05}$ | $0.36_{\pm 0.05}$ | $\mathbf{0.08}_{\pm 0.06}$ | $0.05_{\pm 0.02}$ | $\mathbf{0.59}_{\pm 0.30}$ |

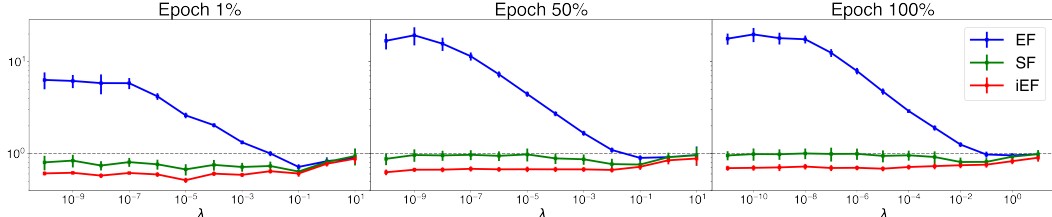

Figure 10: Approximation quality (relative to SGD) of EF, SF and iEF methods *w.r.t.* damping factor $\lambda$ at different training stages of task CIFAR100+ViT+LoRA. The figure is drawn in the same fashion as Fig. 3.

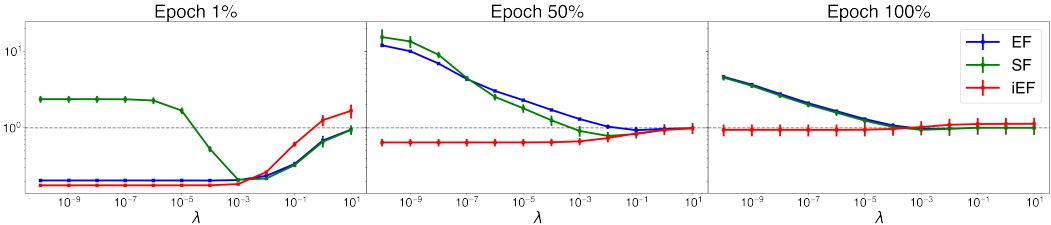

Figure 11: Approximation quality (relative to SGD) of EF, SF and iEF methods *w.r.t.* damping factor $\lambda$ at different training stages of task RTE+T5+LoRA. The figure is drawn in the same fashion as Fig. 3.

Table 6: Validation metrics (multiplied by 100) (on development set) for all the task, structure and optimiser combinations. The average of the highest validation accuracy of 3 random seed runs is reported, along with the standard deviation. Task-specific metrics are reported in this table. For SST-2, QNLI, RTE and CIFAR100, accuracy is reported. For CoLA, both accuracy and Matthew's Corr is reported. For MRPC and QQP, F1-score and Accuracy (in order) are reported. For MNLI, average of accuracy for entire development set is reported.

| | CoLA | SST-2 | MRPC | QQP | MNLI | QNLI | RTE | CIFAR100 |
|---|---|---|---|---|---|---|---|---|
| | | | | Prompt Tuning | | | | |
| Adafactor | $\mathbf{82.0}_{\pm 0.44}$ $53.8_{\pm 1.01}$ | $94.2_{\pm 0.25}$ | $84.8_{\pm 0.35}$ $88.0_{\pm 0.73}$ | $\mathbf{90.7}_{\pm 0.01}$ $\mathbf{87.7}_{\pm 0.01}$ | $82.4_{\pm 0.25}$ | $\mathbf{91.9}_{\pm 0.09}$ | $64.7_{\pm 0.34}$ | - |
| SGD | $69.1_{\pm 0.09}$ $-0.1_{\pm 2.76}$ | $93.3_{\pm 0.14}$ | $70.0_{\pm 0.31}$ $\mathbf{80.8}_{\pm 0.14}$ | $85.0_{\pm 4.11}$ $80.6_{\pm 5.03}$ | $74.5_{\pm 1.17}$ | $88.7_{\pm 0.20}$ | $54.8_{\pm 0.34}$ | - |
| EF | $69.0_{\pm 0.01}$ $-0.55_{\pm 0.58}$ | $90.3_{\pm 0.23}$ | $68.4_{\pm 0.01}$ $81.2_{\pm 0.01}$ | $62.2_{\pm 0.23}$ $0.11_{\pm 0.07}$ | $32.9_{\pm 0.09}$ | $50.9_{\pm 0.59}$ | $55.6_{\pm 1.30}$ | - |
| SF | $76.1_{\pm 2.27}$ $39.4_{\pm 7.71}$ | $93.5_{\pm 0.37}$ | $70.3_{\pm 1.23}$ $81.8_{\pm 0.65}$ | $90.3_{\pm 0.03}$ $87.2_{\pm 0.08}$ | $77.1_{\pm 0.93}$ | $64.2_{\pm 0.65}$ | $57.2_{\pm 0.75}$ | - |
| iEF | $81.7_{\pm 0.16}$ $50.7_{\pm 1.25}$ | $\mathbf{94.4}_{\pm 0.09}$ | $\mathbf{86.2}_{\pm 1.41}$ $\mathbf{89.5}_{\pm 0.46}$ | $\mathbf{90.7}_{\pm 0.02}$ $\mathbf{87.7}_{\pm 0.05}$ | $\mathbf{83.4}_{\pm 0.09}$ | $\mathbf{91.9}_{\pm 0.03}$ | $\mathbf{74.6}_{\pm 0.85}$ | - |
| | | | | LoRA | | | | |
| AdamW | $83.1_{\pm 0.15}$ $58.7_{\pm 0.55}$ | $94.9_{\pm 0.07}$ | $\mathbf{88.6}_{\pm 0.51}$ $\mathbf{91.9}_{\pm 0.26}$ | $90.0_{\pm 0.16}$ $86.8_{\pm 0.06}$ | $83.2_{\pm 0.03}$ | $92.2_{\pm 0.06}$ | $\mathbf{83.4}_{\pm 1.06}$ | $93.7_{\pm 0.32}$ |
| SGD | $81.3_{\pm 0.36}$ $53.6_{\pm 0.97}$ | $\mathbf{95.0}_{\pm 0.18}$ | $87.3_{\pm 0.28}$ $90.1_{\pm 0.41}$ | $89.9_{\pm 0.69}$ $86.7_{\pm 0.88}$ | $\mathbf{83.3}_{\pm 0.05}$ | $\mathbf{92.3}_{\pm 0.09}$ | $80.9_{\pm 0.72}$ | $90.6_{\pm 1.02}$ |
| EF | $69.1_{\pm 0.01}$ $10.5_{\pm 2.10}$ | $91.9_{\pm 0.07}$ | $55.5_{\pm 0.28}$ $71.4_{\pm 0.23}$ | $86.3_{\pm 0.28}$ $82.6_{\pm 0.32}$ | $61.5_{\pm 0.04}$ | $89.4_{\pm 0.21}$ | $52.7_{\pm 0.01}$ | $30.2_{\pm 1.20}$ |
| SF | $79.4_{\pm 0.49}$ $48.3_{\pm 1.45}$ | $94.5_{\pm 0.29}$ | $76.9_{\pm 0.28}$ $85.2_{\pm 0.30}$ | $\mathbf{90.1}_{\pm 0.11}$ $\mathbf{86.9}_{\pm 0.10}$ | $81.9_{\pm 0.17}$ | $91.8_{\pm 0.32}$ | $71.8_{\pm 0.96}$ | $92.7_{\pm 0.72}$ |
| iEF | $\mathbf{83.4}_{\pm 0.24}$ $\mathbf{59.5}_{\pm 0.64}$ | $94.9_{\pm 0.21}$ | $88.5_{\pm 0.88}$ $91.8_{\pm 0.55}$ | $89.9_{\pm 0.09}$ $86.6_{\pm 0.12}$ | $81.2_{\pm 0.02}$ | $92.2_{\pm 0.11}$ | $81.7_{\pm 0.55}$ | $\mathbf{94.1}_{\pm 0.15}$ |

Table 7: Test performance for all task, structure and optimiser combinations. For all tasks, only one test result is reported for the best validation checkpoint across three random seed runs. Task-specific metrics (all multiplied by 100) on the test set are reported in this table. For SST-2, QNLI, RTE and CIFAR100, accuracy is reported. For CoLA, Matthew's Corr is reported. For MRPC and QQP, F1-score and Accuracy (in order) are reported. For MNLI, matched accuracy and unmatched Accuracy (in order) are reported.

| | CoLA | SST-2 | MRPC | QQP | MNLI | QNLI | RTE | CIFAR100 |
|---|---|---|---|---|---|---|---|---|
| | | | Prompt Tuning | | | | | |
| **Adafactor** | 45.1 | 94.3 | 87.1 82.8 | 71.8 88.8 | 82.9 82.7 | **91.5** | 60.7 | - |
| **SGD** | 6.4 | 93.5 | 78.3 66.6 | 71.3 88.9 | 75.7 76.5 | 87.4 | 55.3 | - |
| **EF** | −3.8 | 90.2 | 79.9 66.5 | 0.3 81.6 | 33.4 33.4 | 52.4 | 50.4 | - |
| **SF** | 45.2 | 93.7 | 79.7 67.8 | 71.5 88.5 | 77.8 78.5 | 64.1 | 52.3 | - |
| **iEF** | **50.9** | **94.4** | **88.4** **84.2** | **72.0** **89.2** | **83.5** **83.4** | 91.3 | **68.2** | - |
| | | | LoRA | | | | | |
| **AdamW** | **52.2** | **94.5** | 88.6 85.1 | 71.5 88.8 | **83.6** 83.1 | **92.2** | **71.2** | 93.9 |
| **SGD** | 47.8 | 94.0 | 79.9 66.6 | **71.6** 88.8 | 83.5 **83.8** | 91.9 | 70.1 | 91.3 |
| **EF** | 0.0 | 92.1 | 79.9 66.5 | 65.4 84.4 | 61.4 62.7 | 89.4 | 50.4 | 31.0 |
| **SF** | 42.2 | 94.2 | 84.3 75.8 | 71.1 88.4 | 82.1 82.3 | 91.8 | 64.9 | 92.8 |
| **iEF** | 51.2 | 94.4 | **89.3** **85.9** | 71.5 **88.9** | 81.1 80.8 | **92.2** | 69.1 | **94.3** |

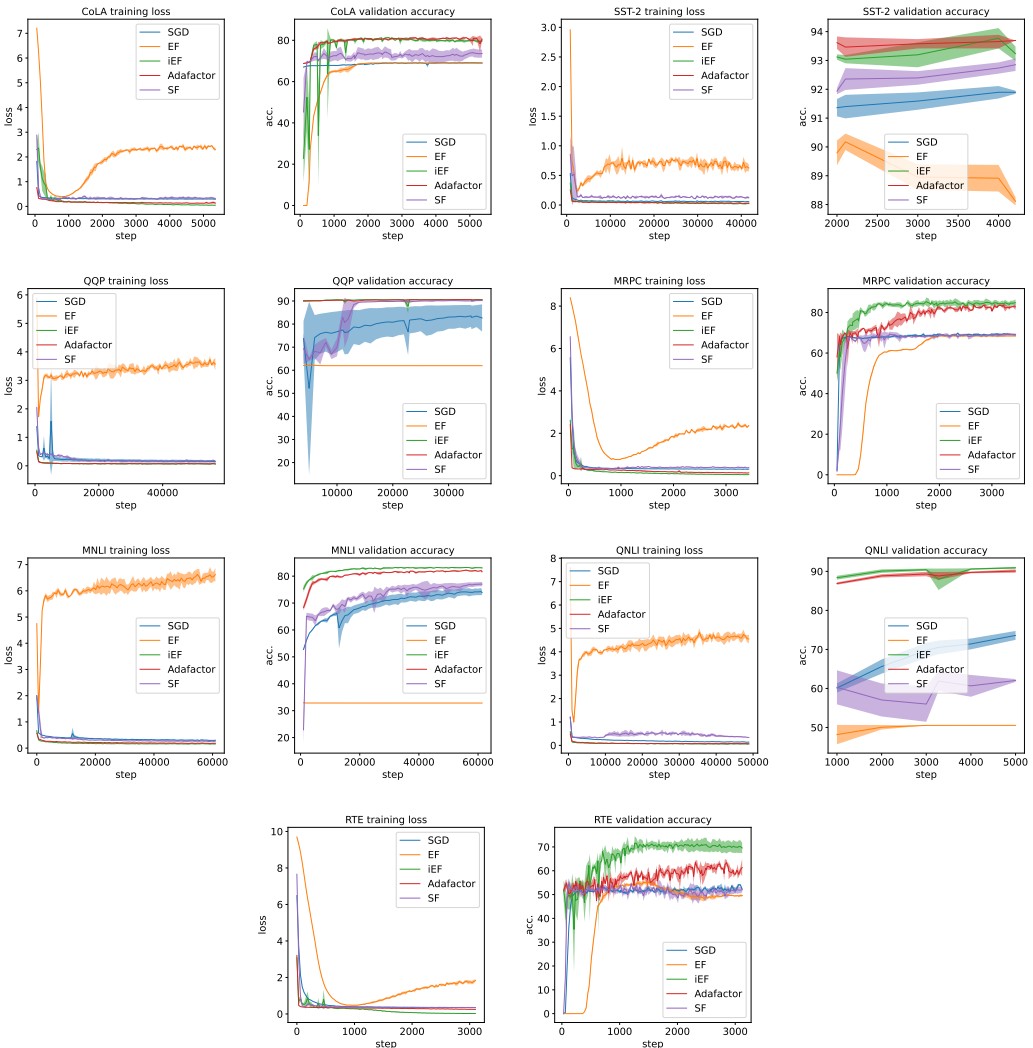

Figure 12: Training loss and validation accuracy of all the Prompt Tuning tasks (7 GLUE tasks). Validation accuracy is reported at the same frequency as is stated by Table 3. For training loss, 100 points are reported across all training stages for every task. Each train loss data point represents the averaged train loss for all train batches between each reported data point. The error bars represent the standard deviation (1-sigma) for 3 random seed runs. The training loss for EF always starts to diverge halfway through training despite the more complicated scheduling. For most tasks, iEF is able to reach a lower training loss than EF, SF and the well-tuned baseline Adafactor.

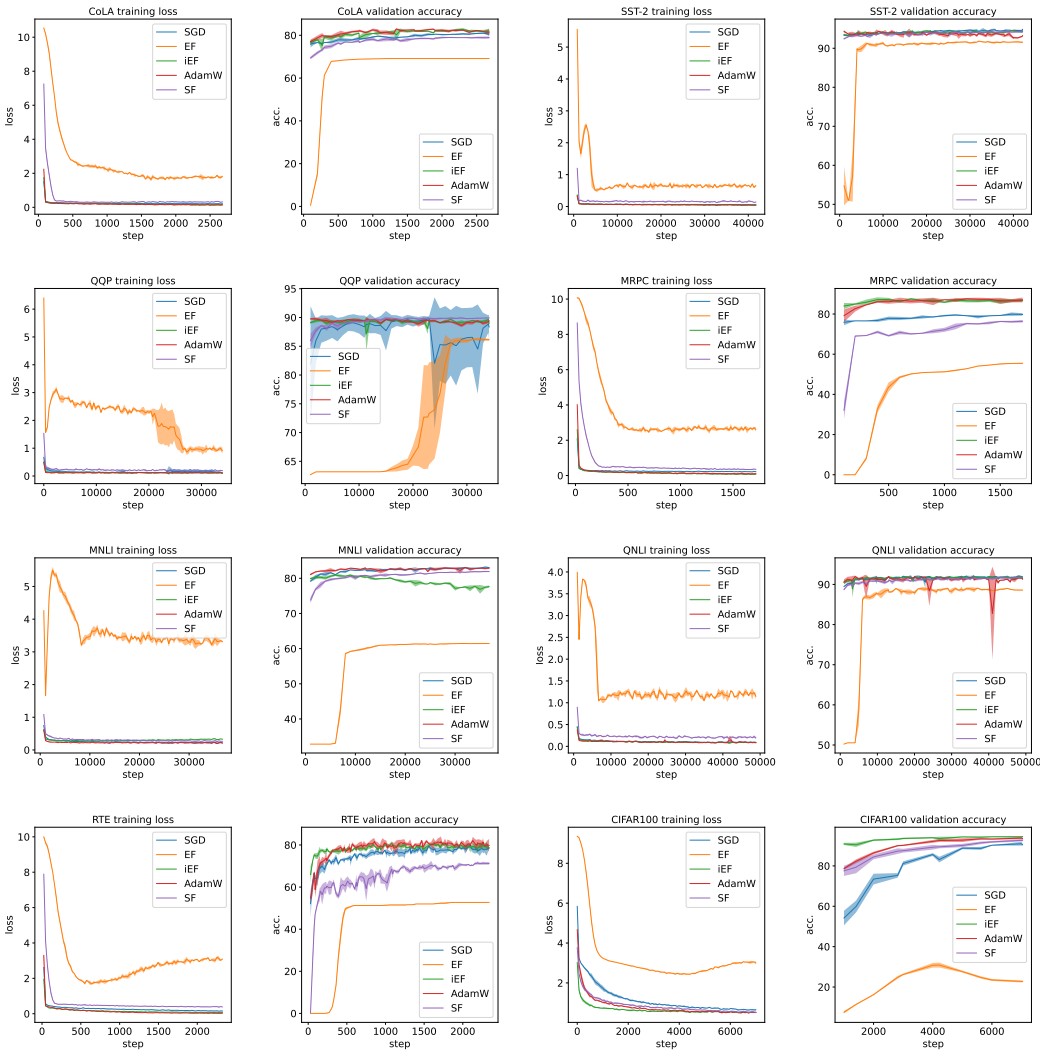

Figure 13: Training loss and validation accuracy of all the LoRA tasks (7 GLUE tasks and 1 CIFAR100). Validation accuracy is reported at the same frequency as is stated by Table 3. For training loss, 100 points are reported across all training stages for every task. Each train loss data point represents the averaged train loss for all train batches between each reported data point. The error bars represent the standard deviation (1-sigma) for 3 random seed runs. The training loss for EF always starts to diverge halfway through training despite the more complicated scheduling. For most tasks, iEF is able to reach a lower training loss than EF, SF and the well-tuned baseline AdamW.

### H.7  Additional Train-from-Scratch Experiment

All of the experiments presented in Sec. 7 are conducted under PEFT setups. While this allows our experimental results to consider up-to-date model structures and tasks, an additional experiment with a larger trainable parameter size (currently less than 1M parameters) and consider non-finetuning setup would be beneficial to further validating the statement of this paper. In this section, a train-from-scratch experiment using a 10M parameter MLP model on the CIFAR10 dataset is conducted and analysed.

**Optimisation Setup**    The used model structure is a 2-hidden-layer ReLU-activated MLP model with a parameter size of 10,510,346 ($\sim$10M), which takes in a flattened 3x32x32 image, and has two 2048 hidden layers. This model structure is an extension to the setups used in [29, 28]. The model is trained for 60 epochs from scratch on the CIFAR10 dataset. During optimisation, no weight decay or dropout is applied. The Adam, SGD, EF, SF and iEF optimisers are used, and for all optimisers, 60 epochs are run with a batch size of 64. The learning rate $1 \times 10^{-4}$ of Adam is searched from $\{5 \times 10^{-5}, 1 \times 10^{-4}, 5 \times 10^{-4}\}$. The learning rate 0.1 of the SGD is searched from 0.01, 0.1, 0.5. The learning rate 50 of iEF is searched from 10, 50, 100. The learning rate 0.1 of SF is searched from $\{0.01, 0.1, 0.5\}$. The learning rate $1 \times 10^{-4}$ of EF is searched from $\{1 \times 10^{-5}, 1 \times 10^{-4}, 1 \times 10^{-3}, 1 \times 10^{-2}, 1 \times 10^{-1}\}$. Normalised update with a linear scheduler is used for EF and SF as in the paper. A constant learning rate is used for iEF. A multi-step scheduler (0.1 decay at fixed epoch number 15 [14]) is used for Adam and SGD. For each optimiser run, 3 seeds are used to generate the error-bars.

**Experimental Results**    Following experiment (E2), the training loss curve and validation accuracy curve is plotted in Fig. 14. The validation and test accuracy is reported in Table 8. Following experiment (E3), the effect of damping on the approximation quality across different stages of training is shown in Fig. 15. Note that we did not conduct a corresponding experiment for (E1) because we believe most of the relevant information is included in Fig. 15. The experimental results show that the conclusions drawn by the PEFT experiment also hold for large train-from-scratch experiments. The exact iEF method demonstrate comparative/stronger model optimisation performance to Adam/SGD baselines, and it remains to be the strongest and most robust method when compared against EF/SF in terms of approximation quality to exact NG updates. Meanwhile, EF shows consistently terrible approximation quality, and struggles to optimise the model, when not carefully damped.

Table 8: Validation and Test accuracy of different optimiser runs for MLP+CIFAR10 setup. For each optimiser run, only one test accuracy is evaluated for the checkpoint with the best validation accuracy.

|  | iEF | Adam | SGD | SF | EF |
|---|---|---|---|---|---|
| **Validation Accuracy (%)** | $\mathbf{58.8}_{\pm 0.87}$ | $56.3_{\pm 0.22}$ | $54.3_{\pm 0.66}$ | $54.8_{\pm 0.08}$ | $28.2_{\pm 1.43}$ |
| **Test Accuracy (%)** | **58.6** | 56.6 | 54.4 | 55.2 | 29.2 |

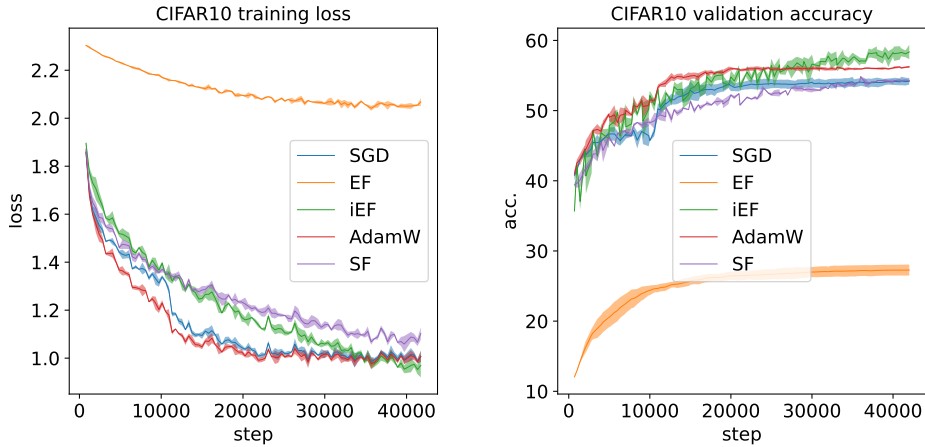

Figure 14: Training loss and validation accuracy curves for the MLP + CIFAR10 train-from-scratch setup. The style of the figure follows that of Fig. 12 and 13. The optimisation performance follows EF < SGD ≈ SF < Adam < iEF, which overall matches the result for the PEFT experiments. Note that, eventually, iEF achieves both the highest validation accuracy and the lowest training loss with a constant learning rate, while Adam and SGD require a multi-step scheduler to perform well.

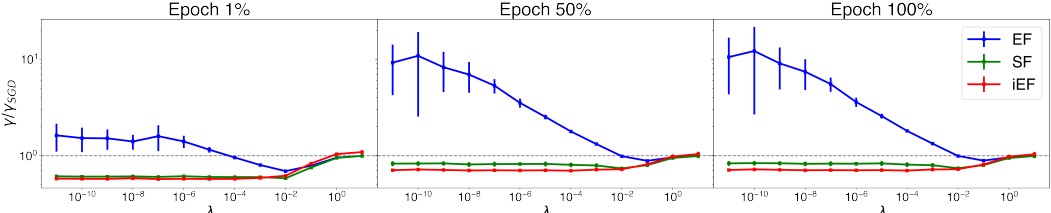

Figure 15: Approximation quality (relative to SGD) of EF, SF and iEF methods *w.r.t.* damping factor $\lambda$ at different training stages of setup task CIFAR10+MLP. The visualisation style and the experimental setup follows that of Fig. 3 (E3). This figure demonstrates that the better approximation quality and the robustness to damping of iEF also hold for a larger train-from-scratch image classification task at different training stages.

