# OpenReview forum: "An Improved Empirical Fisher Approximation for Natural Gradient Descent"
_NeurIPS.cc/2024/Conference — NeurIPS 2024 poster_

### Official Review · Reviewer_Khnn · 2024-06-17

**Soundness:** 3
**Presentation:** 3
**Contribution:** 3
**Rating:** 6
**Confidence:** 3

**Summary:**

The paper proposes a modification of the empirical Fisher matrix that re-scales
a datum's parameter gradient by its logit gradient. This aims to remove a bias
in the empirical Fisher, dubbed 'inversely-scaled projection issue', which
benefits data points that have already been learned. While the modified
empirical Fisher does not introduce any significant cost, its approximate
natural gradient is shown to be closer to the true natural gradient of the
empirical Fisher and a Monte-Carlo approximated Fisher, and is more robust to
the selection of damping.

**Strengths:**

- **Clarity & Evaluation:** The text is clearly written, and the proposed
  modification is theoretically justified. The experiments look thorough to me
  (error bars, modern fine-tuning applications), are clearly analyzed, and
  support the paper's main claims.

- **Effectiveness:** The proposed modification does not add significant
  computational overhead, but is shown to significantly improve the proximity to
  the true natural gradient. It is easy to integrate into existing approximate
  natural gradient optimizers and could therefore serve as promising direction
  for the development of future approximate NGD methods.

**Weaknesses:**

- **Explanation for better approximation of NG update:** The paper demonstrates
  empirically that their EF modification leads to pre-conditioned gradients
  which are closer to the true natural gradient. To me, it is not completely
  clear how addressing the EF's 'inversely-scaled projection issue' causes this
  effect. The authors could further strengthen their paper by providing a
  theoretical argument why removing the empirical Fisher's bias towards learned
  data points results in pre-conditioned gradients with stronger resemblance to
  the true natural gradient.

  One way to do this could be by considering the distributions over which the
  gradient covariances are taken. The empirical Fisher uses the data
  distribution, and the true Fisher uses the likelihood implied by the model. It
  might be interesting to look into whether one can make statements how the
  proposed modification changes the data distribution, and whether it brings it
  closer to the likelihood used in the true Fisher.

- **Optimizer scalability \& experimental scope:** The authors focus on
  evaluating their modification in the 'exact' setting, i.e. without introducing
  additional structural approximations like block diagonals or Kronecker
  factorizations. The implementation needs to hold the per-sample gradients in
  memory, which is costly and renders the optimizer impractical for full
  training of large architectures. I think this is to some extent okay for the
  scope of this paper, because it is relatively obvious how one would apply
  these structural approximations on top of the proposed EF modification.
  However, the current experiments focus heavily on fine-tuning settings and I
  wonder if the findings hold for other settings, too. I would be convinced
  further about the findings if the authors provided an additional experiment
  for 'traditional' image classification (say CIFAR100 or CIFAR10) with a ResNet
  (say ~10M parameters).

  It should be possible to scale the implementation to this setup. I believe a
  good starting point are the hook-based approaches of the BackPACK [1] and ASDL
  [2] frameworks, in combination with the low-rank tricks described in [3, 4],
  which instead of storing the per-sample gradients, compute the pre-conditioner
  matrices from layer inputs and output gradients using less memory.

**References**

[1] Dangel, F., Kunstner, F., & Hennig, P. (2020). BackPACK: Packing more into
  Backprop. ICLR.

[2] Osawa, K., Ishikawa, S., Yokota, R., Li, S., & Hoefler, T. (2023). ASDL: A
  unified interface for gradient preconditioning in PyTorch.

[3] Dangel, F., Tatzel, L., & Hennig, P. (2022). ViViT: curvature access through
  the generalized Gauss-Newton's low-rank structure. TMLR.

[4] Yang, M., Xu, D., Wen, Z., Chen, M., & Xu, P. (2022). Sketch-based empirical
  natural gradient methods for deep learning. Journal of Scientific Computing.

**Questions:**

- Q1: Are there any assumptions on the number of neural network outputs for
  Theorems 5.3, 5.4? You say they are extensions of the proofs in [48], which I
  believe are stated for single-output NNs.

- Q2: How does iEF/EF work in settings with sequence-valued predictions per
  datum and does the computational cost scale in the sequence length?

- Misc: Some typos & editing suggestions
  - L71: 'has gained' -> 'have gained'
  - Fig. 1: Please add somewhere in the caption that $N=2$
  - L153: 'NGD' instead of 'GDN'?
  - L306: 'suffer' -> 'suffers'

**Limitations:**

yes

---

> ### Author Rebuttal · Authors · 2024-08-06
>
> # Author Response to Reviewer Khnn
> Thank you for your review!
>
> >Explanation for better approximation of NG update...To me, it is not completely clear...why removing the empirical Fisher's bias towards learned data points results in pre-conditioned gradients with stronger resemblance to the true natural gradient... One way to do this could be by considering the distributions over which the gradient covariances are taken...
>
> We agree that the statistical framework is commonly used to analyse the approximation quality to NG updates. But such a framework cannot be easily applied to analyse iEF because iEF is primarily motivated from an optimisation/geometric perspective. However, it is important future work to find theoretical support for iEF from a statistical standpoint.
>
> As stated in Sec. 5.2, the iEF update is designed to exactly match the loss reduction behaviour of the GN algorithm, which reduces the loss for each sample according to their logits-level gradient norm. This allows iEF to be "curvature-aware", and perform more agressive updates for untrained samples (with large gradient norm) and conservative updates for converged samples (with small gradient norm). This curvature-awareness helps iEF to better approximate second-order methods, including the NGD method. In contrast, due to the inversely-scaled projection issue, EF updates are essentially "anti-curvature" and perform agressive updates for converged samples and consevative updates for untrained samples. This clearly makes EF a worse second-order method than iEF. It is worth mentioning that, such curvature-awareness may not hold for SGD updates, which may even increase the loss of some samples (depending on the covariance of the per-sample gradients) in a training batch.
>
> >Optimizer scalability & experimental scope: The authors focus on evaluating their modification in the 'exact' setting...I think this is to some extent okay for the scope of this paper...However...I would be convinced further about the findings if the authors provided an additional experiment for 'traditional' image classification (say CIFAR100 or CIFAR10) with a ResNet (say ~10M parameters)...I believe a good starting point are the hook-based approaches...in combination with the low-rank tricks.
>
> We note the reviewer's understanding of our "exact" experimental setup. Also, the reviewer has provided suggestions regarding scaling up to larger setups. We are already using our implementation of hook-based methods to quickly compute per-sample gradients. However, it should be easier if we use the recommended BACKPACK codebase [1] to extend exact iEF to other model structures. Also, we appreciate the approximation techniques recommended by the reviewer, as making iEF an efficient optimiser through approximation is an important future work.
>
> We agree that an additional train-from-scratch image classification setup with a larger model would be beneficial to support the claims in the main paper. We have conducted an experiment for CIFAR10 + large MLP (~10M parameters), which further validates the claims in the main paper. Please refer to ***AE2 of the Global Author Response*** for the detailed experiment setup and results. Note that in this setup we did not need to apply any approximations, as we are able to fit the per-sample gradients of each batch into our GPU RAM.
>
> [1] Dangel, F., Kunstner, F., & Hennig, P. (2020). BackPACK: Packing more into Backprop. ICLR.
>
> >Q1: Are there any assumptions on the number of neural network outputs for Theorems 5.3, 5.4?
>
> There are no assumptions on the number of neural network outputs. This extends the proof in [2] where a regression (scalar-output) model is assumed.
>
> [2] G. Zhang et al (2019). “Fast Convergence of Natural Gradient Descent for Over-Parameterized Neural Networks”.
>
> >Q2: How does iEF/EF work in settings with sequence-valued predictions per datum and does the computational cost scale in the sequence length?
>
> We focus on classification problems where there is only one output distribution per-datum in the main paper. For sequence-valued predictions, there are multiple output distributions per-datum. Assume teacher-forcing is used for training, where the objective function is cross-entropy and a label is provided for every output time-step (assume $T_o$ output time-steps for each datum). There are two ways of applying EF/iEF to this setup.
>
> If the output sequence is treated as a single probability distribution, the EF update can be computed according to Eqn (3) (with the single label $y_n$ now replaced with the target output sequence $(y\_n)\_1, (y\_n)\_2...(y\_n)\_{T\_o}$). Similarly, for iEF, by treating the model output logits as a whole, iEF can still be computed following Eqn (8). Note that now for the iEF scaling factor $\|\nabla_{z_n}l_n\|_2^2$ of the $n$-th datum, output logits $z_n$ becomes the concatenated logits vector for all output time-steps, and the loss $l_n$ becomes the accumulated cross-entropy loss for all output time-steps. In this case, for both EF/iEF, the time complexity is constant per-datum.
>
> Alternatively, every output time-step is treated as an independent classification problem ($T_o$ in total for each datum). Both EF/iEF will now require $T_o$ gradients per-datum (one for each output time-step). In this case, the time complexity becomes $O(T_o)$ per-datum, which scales linearly with sequence length.
>
> The effectiveness of these methods requires further investigation and is left to future work.
>
> > Editing suggestions...
>
> We agree with all the editing suggestions by the reviewer, and will fix them in the final version of the paper.

---

> > ### Comment · Reviewer_Khnn · 2024-08-11
> > **Follow-up comments**
> >
> > Dear authors, thank you for your detailed response.
> >
> > I appreciate your attempt to re-explain why iEF might better approximate the true NG than EF and the additional visualizations.
> >
> > Regarding the from-scratch experiment: I understand the constraints given the limited time. However, the the three-layer MLP on CIFAR10 is weak evidence to me, as the architecture is rather synthetic, achieves only mediocre performance (~60% accuracy), and would therefore not be used by a practitioner to train on this data set.
> >
> > Therefore, I still see the necessity to implement and evaluate iEF on networks with convolution layers that are somewhat representative for what a practitioner would do on CIFAR10 (maybe one of the nets from https://github.com/kuangliu/pytorch-cifar that reach ~90% accuracy).

---

> ### Author Response · Authors · 2024-08-13
>
> # Further Response to Reviewer Khnn
> Thank you for your comments.
>
> > Regarding the from-scratch experiment: I understand the constraints given the limited time. However, the the three-layer MLP on CIFAR10 is weak evidence to me, as the architecture is rather synthetic, achieves only mediocre performance (~60% accuracy), and would therefore not be used by a practitioner to train on this data set.
>
> Thank you for understanding the time constraints that we are experiencing. We note that the reviewer considers our MLP train-from-scratch experiment (***AE2*** of Global Author Response) to be "synthetic" and not "practical", and therefore provides "weak evidence". However, we believe ***AE2*** still provides reasonable evidence for the effectiveness of the iEF method in the train-from-scratch scenario for the following reasons:
> 1. Our MLP architecture is an extension of the MLP setup used in [1]. The main difference in our setup is that we used a hidden layer size 4 times larger than [1] (to reach the recommended 10M parameter size from the reviewers) and we used no dropout in ***AE2***. To better compare our setup with [1], we conducted additional experiments (1-seed) for our setup with a dropout of 0.1 (determined by a grid-search) and the Adam baseline reached a test accuracy of 60.7% and iEF reached 62.2%. Considering the larger capacity of our model, we believe that these results are comparable to the 56.8% of the Adam baseline in [1], indicating that our setup is ***well-tuned***.
> 2. Although the test accuracy is far from the SOTA performance for CIFAR10, we believe ***AE2*** yields useful insights for a large train-from-scratch scenario as long as the setup is well-tuned and the comparison among various optimisers is fair. We believe we have achieved both in ***AE2***
>
> [1] Mahesh Chandra Mukkamala, Matthias Hein. (2017). Variants of RMSProp and Adagrad with Logarithmic Regret Bounds. ICML 2017.
>
> > Therefore, I still see the necessity to implement and evaluate iEF on networks with convolution layers that are somewhat representative for what a practitioner would do on CIFAR10 (maybe one of the nets...that reach ~90% accuracy).
>
> Although we believe that our MLP experiment can provide reasonable evidence for the behaviour of our proposed method in a train-from-scratch setup, we agree with the reviewer that it would be ideal if we could provide experimental results for a better-performing CNN model on CIFAR10.
>
> Therefore, we have been working on implementing a CIFAR10 + ResNet32 [2] (we did not include batch normalisation layers due to time constraints) setup and we have now completed our first experiments with it. In this setup, iEF achieved the highest test accuracy of 85.6%, which is close to the ~90% mentioned by the reviewer and is significantly better than the 58.6% in ***AE2***, making this a much more practically useable setup. The experimental results further validate our claims in the main paper and the details are provided in the section ***AE4: Train-from-scratch Experiment for CNN*** in the Official Comment to the Global Author Response.
>
> [2] Kaiming H, Xiangyu Zhang, Shaoqing Ren, Jian Sun. (2015). Deep Residual Learning for Image Recognition.

---

### Official Review · Reviewer_4Ck3 · 2024-07-02

**Soundness:** 4
**Presentation:** 3
**Contribution:** 3
**Rating:** 5
**Confidence:** 3

**Summary:**

This paper proposed a new Natural Gradient Descent algorithm called iEF. The authors conduct a theoretical convergence analysis of iEF. The proposed PSMGD method achieves comparable or even superior performance over existing gradient descent methods.

**Strengths:**

1. The paper is fairly easy to read, and the ideas are presented well.

2. The inversely scaled projection issue of EF proposed in this paper is original.

3. The proposed iEF method achieves better performance compared with SGD, EF, and SF methods.

4. The theoretical analysis is comprehensive and the results are easy to understand.

**Weaknesses:**

1. I think some of the descriptions are not clear enough and difficult to understand

2. The experiment is not sufficient, and the results are not very impressive.

I explain the above weaknesses in the **Questions** part.

**Questions:**

1. I think the reason why the authors use Eq. (7) to solve the inversely-scaled projection issue raised in Lines 135-136 is not well explained. Why multiply "logits-level gradient norm" can resolve the inversely-scaled projection issue?

2. Though the author says that the proposed scaling vector can be considered as an efficient approximation to the GN algorithm in Section 5.2. This still can not explain why this scaling vector can resolve the inversely-scaled projection issue as the GN algorithm is not proposed for resolving the inversely-scaled projection issue. Additionally, I do not find the definition of the GN matrix and the claim in Lines 175-176 from the reference [26]. Can the authors tell me the detailed location?

3. In Remark 202-204. The authors say "This means that iEF method does not get stuck in local minima, unlike the gradient descent method.". I think most gradient descent methods, can achieve the global minima when the target model uses an m-strongly convex objective function.

4. The experimental result is not sufficient. The results for the AdamW and Adafctor methods are not complete. The proposed iEF method achieves comparable performance with AdamW. Some SOTA baselines such as CAME optimizer and Sophia optimizer are missing.

**Limitations:**

The paper discusses the limitations and there is no societal impact of the work performed.

---

> ### Author Rebuttal · Authors · 2024-08-07
>
> # Author Response to Reviewer 4Ck3
> Thank you for your review!
>
> > I think the reason why the authors use Eq. (7) to solve the inversely-scaled projection issue raised in Lines 135-136 is not well explained...GN algorithm is not proposed for resolving the inversely-scaled projection issue.
>
> We note that it could be better explained how iEF resolves the inversely-scaled projection issue of EF and we provide an alternative clearer explanation here.
>
> The inversely-scaled projection issue of EF is mainly caused by its per-sample loss reduction $\Delta \mathbf{l_\text{EF}} = -\eta\mathbf{1}$ (see equation below line 129), which enforces an equal loss reduction for all training samples without considering their individual level of convergence. This causes EF updates to overly focus on converged samples (with small $||\nabla_\theta l_n||$) while ignoring untrained samples (with large $||\nabla_\theta l_n||$), as is shown by the projection term $\kappa_\text{EF}$ in Eqn (6). To resolve this issue, we improve the per-sample loss reduction behaviour of the EF update to account for the individual convergence level of each sample. The per-sample loss reduction behaviour of iEF is designed to be the same as a GN update: $\Delta \mathbf{l_\text{iEF}} = -\eta||\nabla_{z_n}l_n||^2$ as is shown in Eqn (11). This allows iEF updates to be "curvature-aware", and focus on reducing the loss of untrained samples (with large $||\nabla_{z_n}l_n||$) while be conservative to converged samples (with small $||\nabla_{z_n}l_n||$). Note that $||\nabla_{z_n}l_n||$ is believed to be a good indicator of the convergence level of a sample because the objective function $l_n(z_n)$ is convex, and $||\nabla_{z_n}l_n||$ in general becomes smaller as the sample approaches its minimum.
>
> We feel it would be easier for readers to understand the motivation of iEF if we use Eqn (11) (the per-sample loss reduction) for motivation instead of Eqn (7) (the per-sample projection), and we are happy to make such edits in the final version of the paper.
>
> Regarding the GN algorithm, the reviewer is correct that it is not proposed to resolve the inversely-scaled projection issue. However, the GN algorithm performs well and is a widely-studied second-order optimiser, particularly for traditional ML setups (and was proposed much earlier (1809) than both NGD (1997) and EF (2007)). The iEF update is designed to match the loss reduction behaviour of the GN algorithm mainly to learn from "a successful teacher".
>
> > I do not find the definition of the GN matrix and the claim in Lines 175-176 from the reference [26]. Can the authors tell me the detailed location?
>
> Reference [1] below ([26] in main paper) states that the GN algorithm is equivalent to the $L^2$ NGD (which means it searches on the $L^2$ output space, i.e. $z$-space), which is mentioned in Section 2.6 of [1]. The GN matrix was introduced as the $L^2$ information matrix $G^{L^2}(\theta)$ in section 2.1 of [1] (between Eq. (2.6) and (2.7)).
>
> [1] Levon Nurbekyan, Wanzhou Lei, and Yunan Yang. (2023). “Efficient Natural Gradient Descent Methods for Large-Scale PDE-Based Optimization Problems”
>
> > In Remark 202-204. The authors say "This means that iEF method does not get stuck in local minima, unlike the gradient descent method.". I think most gradient descent methods, can achieve the global minima when the target model uses an m-strongly convex objective function.
>
> The "m-strongly convex objective function" in Assumption C.2 means: for the $n$-th sample, the function between the model output $z_n$ and per-sample loss $l_n$ is "m-strongly convex". As is stated in lines 652-653, under such an assumption, ***the per-sample loss w.r.t. model parameters can still be arbitrarily non-convex*** (depending on the exact structure of the model, i.e. the function between model parameter $\theta$ and model output $z_n$). Consequently, gradient descent methods, which operates on the non-convex model parameter space $\theta$, may not be able to converge to global minima under this assumption.
>
> > The experimental result is not sufficient. The results for the AdamW and Adafctor methods are not complete...Some SOTA baselines such as CAME optimizer and Sophia optimizer are missing.
>
> We conducted additional baseline experiments (Adafactor, CAME, Sophia for LoRA+GLUE, AdamW for PT+GLUE) for 5 selected GLUE tasks, and the validation performance (3 seeds) is reported in the following tables (following the style of Table 6 in the main paper). The following conclusions can be drawn from these experiments:
> 1. AdamW < Adafactor in PT, Adafactor < AdamW in LoRA for 4/5 tasks. That is why these results were left out in the main paper.
> 2. In LoRA, CAME > AdamW for 3/5 tasks, Sophia > AdamW for 2/5 tasks, iEF > CAME for 3/5 tasks, iEF > Sophia for 4/5 tasks. The addition of the two new baselines does not change the conclusion of the paper too much, but we are happy to include these new baseline results in the final paper.
>
> **Prompt Tuning**
>
> |Method|CoLA|SST-2|MRPC|QQP|RTE|
> |-|-|-|-|-|-|
> |Adafactor|$82.0\pm0.44$($53.8\pm1.01$)|$94.2\pm0.25$|$84.8\pm0.35$($88.0\pm0.73$)|$90.7\pm0.01$($87.7\pm0.01$)|$64.7\pm0.34$|
> |AdamW|$81.7\pm0.53$($55.7\pm1.25$)|$94.2\pm0.07$|$83.1\pm1.36$($87.8\pm0.32$)|$89.8\pm0.27$($86.6\pm0.32$)|$62.3\pm4.85$|
>
> **LoRA**
>
> |Method|CoLA|SST-2|MRPC|QQP|RTE|
> |-|-|-|-|-|-|
> |AdamW|$83.1\pm0.15$($58.7\pm0.55$)|$94.9\pm0.07$|$88.6\pm0.51$($91.9\pm0.26$)|$90.0\pm0.16$($86.8\pm0.06$)|$83.4\pm1.06$|
> |iEF|$83.4\pm0.24$($59.5\pm0.64$)|$94.9\pm0.21$|$88.5\pm0.88$($91.8\pm0.55$)|$89.9\pm0.99$($86.8\pm0.12$)|$81.7\pm0.55$|
> |Adafactor|$82.4\pm0.22$($57.8\pm0.52$)|$94.3\pm0.29$|$86.2\pm0.99$($90.4\pm0.57$)|$90.6\pm0.12$($87.5\pm0.17$)|$75.6\pm0.91$|
> |CAME|$83.3\pm0.25$($59.2\pm0.75$)|$94.8\pm0.23$|$89.1\pm0.79$($92.1\pm0.63$)|$90.4\pm0.06$($87.3\pm0.05$)|$59.3\pm3.98$|
> |Sophia|$83.2\pm0.58$($59.2\pm1.36$)|$94.8\pm0.23$|$88.4\pm0.37$($91.8\pm0.31$)|$90.1\pm0.16$($86.9\pm0.19$)|$75.6\pm0.42$|

---

> > ### Comment · Reviewer_4Ck3 · 2024-08-07
> >
> > **R1**: I think there still exists a gap between the inversely scaled projection issue and the used strategy. However, this explanation is somehow clearer and more related to the problem the author found.
> >
> > **R2**. Thanks for your reply. I have found the corresponding information.
> >
> > **R3**. I found that assumption C.2 is included in the appendix. I suggest the author claim this reply in the revision to avoid misunderstanding. When we try to provide a convergent result, we want to reach the global optimal or the stationary point w.r.t. to the model parameter. Is Theorem 5.4. related to this? I did not find the definition of $l_n(t)$. Is it equal to $l_n(\theta(t))$. So $l_n^*$ is related to the optimal model parameter $\theta^*$? Moreover, is there any connection between strongly convex objective function w.r.t model output and model parameter?
> >
> > **R4**. Though these tasks are parts of the GLUE tasks, still thanks for providing these additional results.

---

> ### Author Response · Authors · 2024-08-08
>
> # Further Response to Reviewer 4Ck3
> Thank you for your prompt follow-up!
>
> > R1: I think there still exists a gap between the inversely scaled projection issue and the used strategy. However, this explanation is somehow clearer and more related to the problem the author found.
>
> We are glad that the reviewer finds our explanation clearer. However, we are not sure exactly what you mean by "gap between the inversely scaled projection issue and the used strategy". Could you explain this in more detail please.
>
> > R3. I found that assumption C.2 is included in the appendix. I suggest the author claim this reply in the revision to avoid misunderstanding.
>
> If our understanding is correct, the reviewer is suggesting that we include assumption C.2 and the remark in lines 652-653 regarding "non-convex landscape on the parameter space" in the main paper. We agree with this point and we are happy to include this in the revision.
>
> > I did not find the definition of $l_n(t)$. Is it equal to $l_n(\theta(t))$.
>
> The reviewer is correct that $l_n(t) = l_n(\theta(t))$. $l_n(t)$ represents the $n$-th element of vector $\mathbf{l}(t)$ (i.e. the $n$-th per-sample loss at time $t$), which is first presented in Eqn (12) when describing the iEF update in the continuous time framework.
>
> > So $l^\star_n$ is related to the optimal model parameter $\theta^\star$?
>
> $l^\star_n$ (defined in lines 649-650) represents the lowest possible loss for the $n$-th sample, and is indeed related to the optimal model parameter $\theta^\star$. Near the global minimum, the optimal model parameter $\theta^\star$ is approached when every sample approaches its corresponding optimal loss, i.e. $\forall n, l_n \to l^\star_n$.
>
> > When we try to provide a convergent result, we want to reach the global optimal or the stationary point w.r.t. to the model parameter. Is Theorem 5.4. related to this?
>
> Yes, Theorem 5.4 shows that with the full-batch iEF method, for every sample in the training batch, the per-sample loss $l_n(t)$ will approach the respective optimal per-sample loss $l_n^\star$ at (at least) a linear rate. This then means the model approaches the optimal parameter $\theta^\star$ at a linear rate (as is explained in lines 657-659).
>
> > Moreover, is there any connection between strongly convex objective function w.r.t model output and model parameter?
>
> If our understanding is correct, the reviewer is asking whether the assumption of a strongly convex objective function w.r.t. model output $l_n(z_n)$ involves any further assumption on the parameterisation of the model $\theta$. The answer is that we do not make any further assumptions regarding the model parameterisation when making the "strongly convex objective function" assumption.

---

> > ### Comment · Reviewer_4Ck3 · 2024-08-08
> >
> > Thanks for your reply. As for my questions about the convergent result. Although this method does not need SC w,r,t $\theta$. I believe other assumptions, such as 5.1, can connect the properties in $z$-space to model parameter space. So finally the authors can draw a convergent result w.r.t. $\theta$. (Since SC can lead to PL. why the authors assume SC in Assumption C.2, but only use PL?). It should be quite hard to evaluate whether this method can achieve the global optimal numerically with an objective function, non-convex w.r.t $\theta$ with multiple local optimal and SC w.r.t $z$.
> >
> > Overall, I think my questions have been well-addressed. Since I am unfamiliar with some pieces of related work. I will keep my positive score.

---

> > > ### Author Response · Authors · 2024-08-08
> > >
> > > # Further Response to Reviewer 4Ck3
> > > Thank you for your prompt follow-up!
> > >
> > > > I believe other assumptions, such as 5.1, can connect the properties in $z$-space to model parameter space. So finally the authors can draw a convergent result w.r.t. $\theta$.
> > >
> > > We agree with this statement from the reviewer.
> > >
> > > > why the authors assume SC in Assumption C.2, but only use PL.
> > >
> > > The reason we focuses on "strongly convex objective function" is two-fold:
> > > 1. Convergence results for mean-square-error and general strongly convex objective function is provided in [1] for scalar-output model, and Theorem 5.4 with Assumption C.2 is a direct extension to their result.
> > > 2. Strongly convex objective function covers a wide range of important objective function in deep learning setups (in addition to CE+Softmax). We consider a convergence result for such a family of objective function is important.
> > >
> > > However, as is implied by the reviewer, it is indeed possible to extend our analysis to other objective functions that satisfies PL (in addition to "strongly convex functions") in the future work.
> > >
> > > > It should be quite hard to evaluate whether this method can achieve the global optimal numerically...
> > >
> > > We agree that the global convergence in practice requires further validation. However, we believe that our assumptions on model structure (Assumption 5.1 and 5.2) are practical for a highly over-parameterised model, and are fairly common for similar convergence analysis in the literature such as [1].
> > >
> > > [1] G. Zhang et al (2019). “Fast Convergence of Natural Gradient Descent for Over-Parameterized Neural Networks”.
> > >
> > > > Overall, I think my questions have been well-addressed. Since I am unfamiliar with some pieces of related work. I will keep my positive score.
> > >
> > > We are glad you feel that your questions on our paper have been well-addressed.
> > >
> > > Lastly, given the overall assessment score from the reviewer, we would like to ask if the reviewer has further concerns with the paper that need to be addressed.

---

### Official Review · Reviewer_13ZJ · 2024-07-06

**Soundness:** 3
**Presentation:** 4
**Contribution:** 3
**Rating:** 7
**Confidence:** 5

**Summary:**

The paper analyses the Empirical-Fisher-preconditioned gradient and highlights how its components on the per-sample gradients are biases towards well-trained samples, thus leading to potentially unstable training trajectories. The authors propose to solve this issue by scaling the per-sample components accordingly, which has negligible extra cost.
Theoretically, they prove convergence guarantees under the full-rank NTK assumption. Empirically, they show that iEF is better aligned with natural gradient than standard EF, and beats some baselines in optimization performance.

**Strengths:**

The paper is very well written and structured.
Notation and setting are gently introduced. The problem with EF preconditioning is clarified in great extent formally, and also helped with a nice visualization.

The proposed solution is well supported:

-the negligible extra cost is explained

-theoretical guarantees are carried out, under some (very common in these global convergence results) assumptions

-experiments are done by both measuring alignment (in a clever scalable way) with natural gradient, both

**Weaknesses:**

Regression or Classification? Section 3 begins saying that the paper focuses on classification, which at first allows the authors to define Fisher matrix discarding integrals in favour of a finite sum notation. Then the visual illustration (Sec 4.2 and App B) is on regression problem where, conveniently, NDG=iEF. It would be more fair to have a toy 2D example where NDG and iEF are not equal. Can you provide it?
Moreover, Assumptions C.2 and consequently Theorem 5.4 does *not* hold for classification setting (right?). If that's the case you should definitely make it clear.

And don't get me wrong, the fact that for scalar-output-regression your proposed iEF is exact natural gradient is great, definitely worth being highlighted. But the way these results are presented is misleading.

TYPOS: \
-Line 103 "Jacobian" has an extra "c" \
-Line 128 shouldn't "loss" be "loss change"? \
-Line 153 is "GDN" a typo for "NGD" or am I missing something? \
-Line 270 is "iEF" a typo for "EF"?

**Questions:**

In line 160 I don't totally get the meaning of "is now aligned with the convergence level of each sample", what do you formally mean with the word "aligned"? And in which sense does this "resolve" the inversely-scaled projection issue?

Lines 130-131: "has full row rank, and both $\lambda$ and $\eta$ are infinitesimally small", aren't this essentially Assumption 5.1 ($\lambda=0$ and $\eta\rightarrow 0$) and Assumption 5.2 (jacobian full-row-rank)?

Is it correct that the reason that limits iEF to scale to big models is the same reason that limits EF (i.e. the matrix inversion part)?

**Limitations:**

As mentioned before, there are some limitations with classification/regression, which are not well clarified.

---

> ### Author Rebuttal · Authors · 2024-08-06
>
> # Author Response to Reviewer 13ZJ
> Thank you for your review!
>
> >Regression or Classification? Section 3 begins saying that the paper focuses on classification...Then the visual illustration (Sec 4.2 and App B) is on regression problem...It would be more fair to have a toy 2D example where NDG and iEF are not equal...
>
> We realise a regression visualisation could be confusing as the paper mainly focuses on classification, but we still decided to use the regression setup (Sec 4.2 and App B) in Figure 1 for visualisation for the following reasons:
> 1. The least-squares regression problem is commonly used when analysing NGD and EF in the literature [1, 2]. Particularly, our visualisation follows a similar setup to [2], which is an important related work regarding the limitations of EF. Overall, the toy regression example allows us to be consistent with the literature.
> 2. The least-squares regression problem have several ***nice properties*** (consider only two datum). There exists ***a unique*** global minimum; the NG update can find the global minimum in one step, and the advantage over all other updates ***is self-evident***; the iEF update and NG update ***has the same form*** (as pointed out in the paper and by the reviewer); the distortion of the EF update ***is extreme***. Note that none of these ***properties*** hold for a classification setup, which may make the visualisation less straightforward to understand.
>
> However, as noted by the reviewer, given the scope of this paper, it is important to include a toy 2D visualisation for a classification setup and this is provided in ***Figure 1 of the global response document***. It again demonstrates the distortion effect of EF updates, and the similarity between the iEF and NG updates. Please refer to ***AE1 of Global Author Response*** for details. In the final version of the paper, we will clarify the justifcation of the regression visualisation, and we can include this new classification visualisation if space permits.
>
> [1] Valentin Thomas et al. (2019). “On the interplay between noise and curvature and its effect on optimization and generalization”
>
> [2] Frederik Kunstner, Lukas Balles, and Philipp Hennig. (2019). “Limitations of the Empirical Fisher Approximation for Natural Gradient Descent”.
>
> >Moreover, Assumptions C.2 and consequently Theorem 5.4 does not hold for classification setting (right?). If that's the case you should definitely make it clear.
>
> The global convergence analysis for classification setup (which is more important for the scope of this paper) is given in Theorem 5.3. We proved Theorem 5.4 mainly to extend the iEF global convergence guarantee to more general setups. The reviewer is correct that Assumption C.2 and Theorem 5.4 does not hold for classification, because the softmax + CE loss function typically used in classification problems is not strongly convex. We will emphasize this point in the final paper.
>
> > Typos...
>
> Thanks for pointing out these typos. We will fix them in the final version of the paper.
>
> > In line 160 I don't totally get the meaning of "is now aligned with the convergence level of each sample", what do you formally mean with the word "aligned"? And in which sense does this "resolve" the inversely-scaled projection issue?
>
> We note that how iEF resolves the inversely-scaled projection issue of EF could be better explained and we provide an alternative clearer explanation here.
>
> The inversely-scaled projection issue of EF is mainly caused by its per-sample loss reduction $\Delta \mathbf{l_\text{EF}} = -\eta\mathbf{1}$ (see equation below line 129), which blindly enforces an equal loss reduction for all training samples without considering their individual level of convergence. This causes EF updates to overly focus on converged samples (with small $||\nabla_\theta l_n||$) while ignoring untrained samples (with large $||\nabla_\theta l_n||$), as is shown by the projection term $\kappa_\text{EF}$ in Eqn (6). To resolve this issue, we improve the per-sample loss reduction behaviour of the EF update to account for the individual convergence level of each sample. The per-sample loss reduction behaviour of iEF is designed to be the same as a GN update: $\Delta \mathbf{l_\text{iEF}} = -\eta||\nabla_{z_n}l_n||^2$ as is shown in Eqn (11). This allows iEF updates to be "curvature-aware", and focus on reducing the loss of untrained samples (with large $||\nabla_{z_n}l_n||$) while be conservative to converged samples (with small $||\nabla_{z_n}l_n||$). Note that $||\nabla_{z_n}l_n||$ is believed to be a good indicator of the convergence level of a sample because the objective function $l_n(z_n)$ is convex, and $||\nabla_{z_n}l_n||$ in general becomes smaller as the sample approaches its minimum.
>
> We feel it would be easier for readers to understand the motivation of iEF if we use Eqn (11) (the per-sample loss reduction) for motivation instead of Eqn (7) (the per-sample projection), and we are happy to make such edits in the final paper.
>
> > Lines 130-131: "has full row rank, and both $\lambda$ and $\eta$ are infinitesimally small", aren't this essentially Assumption 5.1 and Assumption 5.2?
>
> The reviewer is correct. We will add a reference to Assumption 5.1/5.2 on line 130-131 in the final paper.
>
> > Is it correct that the reason that limits iEF to scale to big models is the same reason that limits EF (i.e. the matrix inversion part)?
>
> The reviewer is correct that the practical limitations of exact EF/iEF are identical (given their similar generation process in Algorithm 1/2). However, the main bottleneck of applying iEF/EF to big models is the memory complexity of storing the per-sample gradient $O(MP)$ ($M$ being the batch size and $P$ being the trainable parameter size), instead of the "matrix inversion" (for a Gram matrix of size $M\times M$), which is manageable as long as $M$ is not too large. A detailed discussion is provided in the "Time and Memory Complexity" paragraph of Appendix D.1.

---

> > ### Comment · Reviewer_13ZJ · 2024-08-08
> > **Keep score**
> >
> > Thanks for the clarifications and thanks for the toy classification visualization, great job! I keep my score and confirm my willing for acceptance.

---

> > > ### Author Response · Authors · 2024-08-08
> > > **Thank you**
> > >
> > > Thank you very much for your positive review of our paper!

---

### Official Review · Reviewer_Sh4Z · 2024-07-18

**Soundness:** 3
**Presentation:** 3
**Contribution:** 3
**Rating:** 5
**Confidence:** 2

**Summary:**

Many approaches have been proposed to approximate natural gradient descent, however most of them rely on estimating the fisher matrix with empirical fisher. The estimation is known to have the inversely-scaled projection issue, where the update is inversely proportional to per-sample gradient norm, as such samples that are not well-learnt would have a small update. In addition, as the covariance of the gradient is not invertible, a damping factor is required to ensure invertibility, and this parameter could be hard to tune in practice. To resolve this issue, improved empirical fisher (iEF) is proposed, which multiplies the per-sample gradient by the norm of its likelihood's gradient with respect to the logits, as such all samples would have same scale of updates. The authors then empirically shows that iEF is a better approximation to natural gradient descent (using a novel evaluation metric) compared with standard empirical fisher and shows better performance then AdamW on parameter efficient fine-tuning tasks.

**Strengths:**

-The proposed method is well motivated and supported by convergence guarantee.

- The proposed indicator for evaluating natural gradient approximation quality is very interesting and useful.

- The empirical experiments verify the usefulness of the proposed indicator and the proposed iEF method.

- The experiment settings considered are very up-to-date and practical: Parameter efficient fine-tuning for large model.

**Weaknesses:**

- The evaluation only contains fine-tuning experiments with parameter efficient fine-tuning techniques, it would be nice to have some more train-from-scratch. In addition, the workloads considered are of rather small scale in terms of tunable parameters, e.g. ResNet-18 has 11M parameters, while the largest tasks in the submission, fine-tuning T5 on GLUE, only contains 0.8M parameters.

- The experiments only consider a small number of training iterations (which is common for fine-tuning setting), but it would still be nice to see the behavior of the proposed method in longer training run.

- It would be better if the authors could cite [1] for Fig. 1.

- It would be nice to see experiments for iEF + KFAC (as is discussed in line 722).

[1] Limitations of the Empirical Fisher Approximation for Natural Gradient Descent

**Questions:**

- How does the method compare with Gauss-Newton? Can GN be considered as a baseline approach?

- Why does the approximation quality goes worth as optimization proceeds? (first three subfigure in Fig. 2)

- Why is the relationship between approximation quality and damping factor non monotonic?

- How does iEF compare with AdamW in terms of computational cost?

- In the appendix, line 960 says sgd's learning rate is searched between  {0.1, ... ,100}, 100 seems to be a pretty larger number? Is that a typo?

- I wonder if the authors could provide learning rate v.s. metrics plot to better convince readers that the learning rate is well tuned.

- The baseline AdamW uses weight decay but iEF does not incorporate weight decay, it would be nice to see results of AdamW without weight decay.

**Limitations:**

- The proposed method could potentially be more expensive than standard optimizers as Eq. 8 would require per-sample gradient.

---

> ### Author Rebuttal · Authors · 2024-08-07
>
> # Author Response to Reviewer Sh4Z
> Thank you for your review!
>
> >The evaluation only contains fine-tuning experiments with parameter efficient fine-tuning techniques, it would be nice to have some more train-from-scratch...small...tunable parameters...small number of training iterations.
>
> We agree that a train-from-scratch setup with a large number of training epochs and with a model with a large number of parameters can further support the claims of our paper. We have conducted an experiment for CIFAR10 + large MLP (~10M parameters), which further validates the claims in the main paper. Please refer to ***AE2 of the Global Author Response*** for detailed experiment setup and results.
>
> > It would be better if the authors could cite [1] for Fig. 1.
>
> We have cited [1] in Appendix B, which is referenced in the caption for Fig. 1. We will add a citation to [1] directly in the caption for Fig. 1 in the final version of the paper.
>
> [1] Frederik Kunstner, Lukas Balles, and Philipp Hennig. (2019). “Limitations of the Empirical Fisher Approximation for Natural Gradient Descent”.
>
> > It would be nice to see experiments for iEF + KFAC (as is discussed in line 722).
>
> We have conducted a preliminary evaluation (using our empirical evaluation framework) for block-diagonal versions of EF, iEF and SF on a selection of PEFT tasks. We found that KFAC with iEF achieves the most consistently good approximation quality to NG updates as compared to standard KFAC and KFAC with EF. Please refer to ***AE3 of Global Author Response*** for the detailed experiment setup and results.
>
> > How does the method compare with Gauss-Newton? Can GN be considered as a baseline approach?
>
> The GN algorithm is as expensive to implement as the NGD method, which cannot be easily implemented as a baseline optimiser for large setups. However, it is possible to extend the evaluation framework (by replacing the Fisher matrix $\mathbf{F}$ with the GN matrix $\hat{\mathbf{G}}$ in Eqn (16)) to evaluate the update approximation quality to an exact GN update.
>
> > Why does the approximation quality goes worth as optimization proceeds? (first three subfigure in Fig. 2)
>
> The datapoints reported in the first 3 subfigures of Fig 2 are relative indicator values w.r.t. SGD, i.e. $\frac{\gamma_\text{update}}{\gamma_\text{SGD}}$. Hence, the overall increasing trend only means that the relative improvement of the approximation quality to the NG update w.r.t. SGD diminishes as training progresses, and it does not necessarily mean that the approximation quality becomes worse. The cause requires further investigation, and is likely dependent on the model structure and task of the setup.
>
> > Why is the relationship between approximation quality and damping factor non monotonic?
>
> The relationship between damping and approximation quality for iEF is overall monotonic, but it is non-monotonic for both SF and EF approximations (see Figs 3/6/7/8/9). Although there is no guarantee that the relationship between damping and approximation quality should be monotonic, the fact that iEF has an overall monotonic relationship while SF and EF do not indicates that iEF is much more well-behaved and less sensitive to damping tuning, which is a practical advantage of iEF.
>
> > How does iEF compare with AdamW in terms of computational cost?...The proposed method could potentially be more expensive than standard optimizers as Eq. 8 would require per-sample gradient.
>
> The analysis of time and memory complexity of the iEF method is provided in Appendix D.1. The per-sample gradient theoretically can be obtained when computing the batch gradient, and incurs no additional computational cost. The main additional cost comes from the computation of the Gram matrix: $O(M^2P)$, $M$ being the batch size and $P$ being the parameter size, which is relatively small as compared to a back-propagation (assume $M$ is not too large). However, currently our implementation of per-sample gradient is achieved with backward hooks (in Pytorch), which roughly doubles the cost of the standard back-propagation process.
>
> However, we would like to emphasize that the exact iEF method implemented in our paper is mainly for theoretical accuracy during experimental comparison. Various mature approximation techniques could be used to accelerate practical iEF-based optimisers (as suggested in Appendix D.2).
>
> > In the appendix, line 960 says sgd's learning rate is searched between {0.1, ... ,100}, 100 seems to be a pretty larger number? Is that a typo? I wonder if the authors could provide learning rate v.s. metrics plot to better convince readers that the learning rate is well tuned.
>
> The SGD learning rate search range for LoRA + GLUE setups was indeed {0.1, 1, 10, 20, 50, 100} (identical range for PT). Such a large learning rate is searched mainly because in PT setups, which we carried out first, the gradient norm is significantly smaller than for other setups such as LoRA. Hence, a large learning rate is necessary for the model to converge effectively.
>
> We provide a table of the average validation accuracy (1 seed) vs. SGD learning rate below for the LoRA + GLUE setup to show that our learning rate is well tuned (we also searched an additional 0.01). The validation accruacy is averaged across 5 selected tasks (CoLA, SST-2, MRPC, QQP, RTE).
>
> |lr|Avg Val Acc.|
> |-|-|
> |0.01|80.6|
> |0.1|86.9|
> |1|64.1|
> |10|60.9|
>
> > The baseline AdamW uses weight decay but iEF does not incorporate weight decay, it would be nice to see results of AdamW without weight decay.
>
> We re-ran the AdamW + LoRA + GLUE experiments without weight decay. The averaged validation accuracy (3 seeds) for 5 selected tasks (CoLA, SST-2, MRPC, QQP, RTE) are as follows:
>
> |Method|Avg Val Acc.|
> |-|-|
> |AdamW|88.0|
> |AdamW w/o wd|87.8|
> |iEF|87.7|
>
> AdamW without weight decay overall shows only slightly worse generalisation, but indeed improves the relative performance of iEF. We are happy to include these results in the final paper.

---

> ### Author Response · Authors · 2024-08-13
>
> Dear Reviewer Sh4Z,
>
> For the reviewer's information, we have by now provided additional experimental results for another train-from-scratch setup: CIFAR10 + ResNet32 to further support the claims in the main paper. Details are provided in Section ***AE4: Train-from-scratch Experiment for CNN*** in the Official Comment to Global Author Response. Note that iEF achieved the best test accuracy of 85.6% for CIFAR10 in this setup, which makes it a more pratically useable setup than the MLP setup in ***AE2*** (best test accuracy of 58.6%). We believe this additional CNN-based experiment (***AE4***), together with the large MLP experiment (***AE2***), have better addressed your concerns regarding the lack of a large train-from-scratch experiment in our submission.
>
> Best regards,
>
> Paper 7035 Authors

---

### Author Rebuttal · Authors · 2024-08-07

# Global Author Response
Thank you all for your positive reviews!

We have attached to this global response a pdf document, which contains information for additional experiments (***AE***) (4 figures with captions) that address concerns regarding our paper. This document is referred to as the "Global Response Document" in our seperate author response to each reviewer. A summary of the contents of this pdf is provided as follows.

### AE1: Logistic Regression Classification
To address the suggestions of reviewer 13ZJ regarding the need for a classification based visualisation for the iEF method (in addition to the least-squares regression visualisation in Figure 1 of the main paper), we have provided a 2D visualisation for a toy logistic regression problem in Figure 1 of the global response document (following the style of Figure 1 in the main paper). Detailed problem setups are provided in the caption, and update generation of SGD/EF/iEF/NGD follows that of Appendix B in the main paper. The visualisation further validates the distortion of EF updates in logistic regression problem, and also demonstrates that iEF indeed resolves the inversely-scaled projection issue and achieves high resemblance to NG updates.
### AE2: Train-from-scratch Experiment
To address the suggestions of reviewer Sh4Z and Khnn regarding the need for an additional experiments on a classical train-from-scratch image classification task with a larger model and more training iterations, we have provided a full set of experimental results for a CIFAR10 + MLP setup (Figure 2, 3 in the global response document, corresponding to E1, E2, E3 in the main paper for PEFT setups). The 3-layer ReLU-activated MLP model has a parameter size of 10,510,346 (~10M), which takes in a flattened 3x32x32 image, and has two 2048 hidden layers (developed based on the setups described in [1, 2]).
1. An MLP model is used instead of a ResNet model due to the following reasons: **1)** It is straightforward to extend our current per-sample gradient implementation from LoRA setup to MLP model, both of which involve only Linear modules (in Pytorch); **2)** Reviewer Sh4Z and Khnn suggested using a ResNet of ~10M parameters. However, for CIFAR setups, 500 residual blocks are needed to reach this parameter size. While ResNet18 for ImageNet indeed has ~10M parameters, the training set of ImageNet is too large to complete during the limited rebuttal time [3]. Both options are difficult to run given the time constraints; **3)** We believe the current setup is sufficient to provide insight into the behaviours of EF/SF/iEF in a larger train-from-scratch setup.
2. During optimisation, no weight decay or dropout is applied. The Adam, SGD, EF, SF and iEF optimisers are used, and for all optimisers, 60 epochs are run with a batch size of 64. The learning rate 1e-4 of Adam is searched from {5e-5, 1e-4, 5e-4}. The learning rate 0.1 of the SGD is searched from {0.01, 0.1, 0.5}. The learning rate 50 of iEF is searched from {10, 50, 100}. The learning rate 0.1 of SF is searched from {0.01, 0.1, 0.5}. The learning rate 1e-4 of EF is searched from {1e-5, 1e-4, 1e-3, 0.01, 0.1}. Normalised update with a linear scheduler is used for EF and SF as in the paper. A constant learning rate is used for iEF. A multi-step scheduler (0.1 decay at fixed epoch number 15 [3]) is used for Adam and SGD. The training loss and validation accuracy curves are plotted in Fig. 2 of the global response document, error bar computed with 3-seeded runs. The final results of the experiments are as follows:

| Method | Val Acc. | Test Acc. |
| - | - | - |
| iEF     | $58.8\pm0.87$ | $58.6$ |
| Adam | $56.3\pm0.22$ | $56.6$ |
| SGD     | $54.3\pm0.66$ | $54.4$ |
| SF     | $54.8\pm0.08$ | $55.2$ |
| EF     | $28.2\pm1.43$ | $29.2$ |

Overall, we can observe that EF < SF $\approx$ SGD < Adam < iEF in terms of both generalisation and convergence, which is aligned with the claims in the main paper.

3. The approximation quality w.r.t. damping factor result is presented in Fig. 3 of the global response document, which shows consistent results with conclusions drawn in experiment E1, E3 of the main paper. Note that due to space limits, we are unable to provide a plot for experiment E1 (equivalent to Fig. 2 in the main paper)

[1] Behnam Neyshabur, Zhiyuan Li, Srinadh Bhojanapalli, Yann LeCun, Nathan Srebro. (2018). Towards Understanding the Role of Over-Parametrization in Generalization of Neural Networks.

[2] Mahesh Chandra Mukkamala, Matthias Hein. (2017). Variants of RMSProp and Adagrad with Logarithmic Regret Bounds.

[3] Kaiming H, Xiangyu Zhang, Shaoqing Ren, Jian Sun. (2015). Deep Residual Learning for Image Recognition.

### AE3: KFAC + iEF Experiment
As is requested by Reviewer Sh4Z, we have conducted a preliminary evaluation for the concept of combining KFAC with iEF (as discussed in Appendix D.2 of the main paper). The empirical evaluation framework proposed in the main paper is used to evaluate the approximation quality of 3 additional block-diagonal methods: KFAC, eKFAC and ieKFAC. KFAC stands for standard KFAC with SF (with 1 MC sample per-datum). eKFAC stands for KFAC with EF (see Eqn (43)) and ieKFAC stands for KFAC with iEF (see Eqn (44)). All methods use a damping factor of 1e-7 as in experiment E1 of the main paper. The evaluation approximation quality w.r.t. training progress for 3 selected tasks: QNLI+T5+LoRA, RTE+T5+LoRA, MRPC+T5+LoRA are shown in Figure 4 of the global response document. It is demonstrated that ieKFAC achieves the most consistent improvement of the approximation quality to NG updates, as compared to KFAC and eKFAC. This indicates the potential of the developement of ieKFAC in future work.

### Further Additional Experiments
There are other additional experiments that cannot fit into the one-page global response document. We will provided these results in our separate individual author responses.

---

> ### Author Response · Authors · 2024-08-13
> **Additional Train-from-Scratch Experiment (AE4) for CIFAR10 + ResNet32**
>
> # Additional Train-from-Scratch Experiment (AE4) for CIFAR10 + ResNet32
> We have now applied exact iEF/EF/SF on a commonly used CNN architecture: ResNet32 [1] on CIFAR10, which further validates the claims in the main paper for a practical train-from-scratch setup. Details are provided in the section below.
>
> ### AE4: Train-from-scratch Experiment for CNN
> We have now completed an implementation of the exact iEF/EF/SF method for CNN modules, and we have conducted experiments (E1, E2, E3 in the main paper) for CIFAR10 + ResNet32 [1] (without batch normalisation (BN)), as an addition to the CIFAR10 + MLP setup introduced in the Global Author Response in section ***AE2: Train-from-scratch Experiment***. The model architecture mostly follows the description in [1], which has a parameter size of 0.46M.
> 1. We did not include batch normalisation layers in ResNet32 mainly due to time constraints. However, we believe it is important for future work to extend experiments to models with normalisation layers. Note that it is not completely straightforward to integrate BN layers into iEF because the parameters for BN layers are very different by nature when compared to CNN/MLP weights (which can both be considered as linear transformations). We also note that the implementations of popular second-order methods such as K-FAC and its variants usually also ignore BN layers [2].
> 2. In our new experiments on ResNet32+CIFAR10, no weight decay or dropout is applied during optimisation. The Adam, SGD, EF, SF and iEF optimisers are used, and for all optimisers, 100 epochs are run with a batch size of 32. Normalised updates with a linear scheduler are used for EF and SF as in the paper, with the corresponding starting learning rate of 5e-5 and 0.1. The warmup-constant-decay scheduler is used for iEF, where the learning rate changes linearly in the first and final 30 epochs of training, and stays constant at 100 in the middle 40 epochs. A multi-step scheduler (0.1 decay at fixed epoch number 30) is used for Adam and SGD, with a starting learning rate of 5e-4 and 0.1 correspondingly. Also, SGD uses 1 epoch to warm up to the target learning rate. Grid search is applied to search for all learning rates. The final optimisation results of the experiments are as follows:
>
> |Method|Val Acc.|Test Acc.|
> |-|-|-|
> |iEF|$85.9\pm0.27$|$85.6$|
> |SGD|$85.2\pm0.85$|$85.4$|
> |Adam|$84.8\pm0.33$|$84.5$|
> |SF|$79.0\pm0.31$|$78.6$|
> |EF|$13.4\pm0.31$|$13.1$|
>
> Overall, we can observe that EF < SF < Adam < SGD < iEF in terms of generalisation and convergence, which is aligned with the claims in the main paper.
>
> 3. Evaluation with the empirical evaluation framework for approximation quality (corresponding to E1, E3 of main paper) further validates the conclusions in the main paper, where iEF consistently achieves the best approximation quality with a small damping factor, and is the most robust to the choice of damping factor. Interestingly, the corresponding curves have a very similar trend to that of CIFAR100 + ViT in the main paper (i.e. Figure 2 and Figure 8).
>
> Finally, we are happy to include this additional experiment in the final paper.
>
> [1] Kaiming H, Xiangyu Zhang, Shaoqing Ren, Jian Sun. (2015). Deep Residual Learning for Image Recognition.
>
> [2] James Martens, Roger Grosse. (2015). Optimizing Neural Networks with Kronecker-factored Approximate Curvature.

---

### Decision · Program_Chairs · 2024-09-25

**Decision:**

Accept (poster)

**Comment:**

The reviewers have expressed that the proposed indicator for evaluating natural gradient approximation quality is very interesting and useful. Moreover, the empirical experiments verify the usefulness of the proposed indicator and the proposed iEF method. Besides this, the experiment settings considered are very up-to-date and practical according to the reviewers and the paper is very well written and structured. The proposed solution is also well supported. The reviewers have also highlighted the originality of the proposed method. Furthermore, the theoretical analysis is comprehensive and the results are easy to understand. Given this, I believe this paper will constitute a nice contribution to the conference.